Manuscript prepared for Geosci. Model Dev.
with version 2014/07/29 7.12 Copernicus papers of the LaTeX class copernicus.cls.
Date: 12 September 2018

# A representation of the collisional ice break-up process in the two-moment microphysics scheme LIMA v1.0 of Meso-NH

Thomas Hoarau[1], Jean-Pierre Pinty[2], and Christelle Barthe[1]

[1]Laboratoire de l'Atmosphère et Cyclones, UMR 8105, CNRS/Météo-France/Université de La Réunion, St Denis, La Réunion, France
[2]Laboratoire d'Aérologie, University of Toulouse/CNRS/UPS, 14 avenue Edouard Belin, F-31400 Toulouse, France

*Correspondence to:* jean-pierre.pinty@aero.obs-mip.fr

**Abstract.** The paper describes a switchable parameterization of Collisional Ice Break-Up (CIBU), an ice multiplication process that fits in with the two-moment microphysical scheme LIMA (Liquid Ice Multiple Aerosols). The LIMA scheme with three ice types (pristine cloud ice crystals, snow-aggregates and graupel-hail) was developed in the cloud-resolving mesoscale model Meso-NH. Here

the CIBU parameterization assumes that collisional break-up is mostly efficient for the small and fragile snow-aggregate class of particles when they are hit by large, dense graupel particles. The increase of cloud ice number concentration depends on a prescribed number (or a random number) of fragments being produced per collision. This point is discussed and analytical expressions of the newly contributing CIBU terms in LIMA are given.

The scheme is run in the cloud resolving mesoscale model Meso-NH to simulate a first case of a three-dimensional deep convective event with heavy production of graupel. The consequence of dramatically changing the number of fragments produced per collision is investigated by examining the rainfall rates and the changes in small ice concentrations and mass mixing ratios. Many budgets of the ice phase are shown and the sensitivity of CIBU to the initial concentration of freezing nuclei

is explored.

The scheme is then tested for another deep convective case where additionally, the CAPE (Convective Available Potential Energy) is varied. The results confirm the strong impact of CIBU with up to a 1,000 fold increase in small ice concentrations, a reduction of the rainfall or precipitating area and an invigoration of the convection with higher cloud tops.

Finally it is concluded that the efficiency of the ice crystal fragmentation needs to be tuned carefully. The proposed parameterization of CIBU is easy to implement in any two-moment microphysics

scheme. It could be used in this form to simulate deep tropical cloud systems where anomalously high concentrations of small ice crystals are suspected.

## 1 Introduction

In a series of papers, Yano and Phillips (2011, 2016) and Yano et al. (2016) brought the Collisional Ice Break-Up (hereafter CIBU) process to the fore again as a possible secondary ice production mechanism in clouds. Using an analytical model, they showed that CIBU could lead to an explosive growth of small ice crystal concentrations. Afterwards Sullivan et al. (2017) tried to include CIBU in a six-hydrometeor-class parcel model, in which hydrometeors were assumed to be monodispersed, in an attempt to investigate the ice crystal number enhancement. However, intriguingly, and in contrast to the Hallett-Mossop ice multiplication mechanism[1] (hereafter H-M) (Hallett and Mossop, 1974), the vast majority of microphysics schemes do not include the CIBU process. Yet, the CIBU process is very likely to be active in inhomogeneous cloud regions where ice crystals of different sizes and types are locally mixed (Hobbs and Rangno, 1985; Rangno and Hobbs, 2001). For instance, collisions between large dense graupel growing by riming, and plane vapour-grown dendrites or irregular weakly rimed assemblages are the most conceivable scenario for generating multiple ice debris as envisioned by Hobbs and Farber (1972) and by Griggs and Choularton (1986). Therefore, a legitimate quest for a two-moment mixed-phase microphysics scheme, where number concentrations and mixing ratios of the ice crystals are predicted, is to find ways to include an ice-ice break-up mechanism and to characterize its importance, relative to other ice generating processes such as ice heterogeneous nucleation. Our aim to introduce CIBU in a microphysics scheme was initially motivated by the detection of unexplained high ice water content which sometimes largely exceeded the concentration of ice nucleating particles (Leroy et al., 2015; Field et al., 2017; Ladino et al., 2017).

As recalled by Yano and Phillips (2011), the first laboratory experiments dedicated to the study of ice collisions were conducted in the 1970s following investigations concerning the promising H-M process. In the pioneering work of Vardiman (1978) who highlighted the mechanical fracturing of natural ice crystals, the number of fragments was dependent on the shape of the initial colliding crystal and on the momentum change following the collision. According to a concluding remark by Vardiman (1978), this secondary production of ice could lead to concentrations as high as 1,000 times the natural concentrations of ice crystals in clouds that would be expected from heterogeneous nucleation on ice freezing nuclei. Another laboratory study by Takahashi et al. (1995) also revealed a huge production of ice splinters after collisions between rimed and deposition-grown graupels. However because as many as 400 fragments could be obtained, their experimental set-up was more appropriate to very large, artificially grown crystals and to large impact velocities.

---

[1]H-M is based on the explosive riming of "big" droplets on graupel particles in a narrow range of temperatures

For clarity, this study does not focus on cloud conditions that lead to explosive ice multiplication due to mechanical break-up in ice-ice collisions. Neither does it attempt to reformulate this process on the basis of collisional kinetic energy with many empirical parameters, as proposed by Phillips et al. (2017), or earlier by Hobbs and Farber (1972) in terms of their breaking energy, mostly applicable to bin microphysics schemes. Here, the goal is rather to implement an empirical but realistic parameterization of CIBU in the LIMA (Liquid, Ice, Multiple Aerosols) microphysics scheme (Vié et al., 2016) in conjunction with other microphysical processes (heterogeneous ice nucleation, droplet freezing, H-M process, etc.) to improve the representation of small ice crystal concentrations. In this study, our representation of CIBU is the formation of cloud ice crystals as the result of collisions between big graupel particles and small aggregates after which the graupel particles lose mass to the aggregates. This parameterization of CIBU relies on the laboratory observations by Vardiman (1978) to set limits on the number of fragments per collision. However, the large uncertainties attached to this parameter encouraged us to run exploratory experiments with several fixed values and also to model the number of fragments by means of a random process.

The LIMA scheme, inserted in the host model Meso-NH (Lafore et al., 1998), forms the framework of the present study. Several sensitivity experiments are performed to evaluate the importance of the CIBU process and the impact of the tuning (*i.e.* the number of fragments produced per collision). The efficiency of CIBU in dramatically increasing the concentration of small ice crystals can be scaled by the ice number concentration from nucleation. The case of a three-dimensional continental deep convective storm, the well-known STERAO (Stratospheric-Tropospheric Experiment: Radiation, Aerosols and Ozone) case simulated by Skamarock et al. (2000), provided a framework for several adjustments of the number of ice fragments. A series of experiments was then performed for the same case to see how much the CIBU process altered the precipitation and the persistence of convective plumes. The question of the number of ice nuclei necessary to initate CIBU (Field et al., 2017; Sullivan et al., 2018) was also addressed. A second case of a deep convective cloud (Weisman and Klemp, 1984) is run to confirm the impact of CIBU in a series of different CAPE environments. The simulations showed that the invigoration of convection when the CIBU efficiency was strong, led to larger cloud covers and an increase of the mean cloud top height. Finally, a conclusion is drawn on the importance of calibrating the parameterization of CIBU and the need to systematically include CIBU and other ice multiplication processes in bulk microphysics schemes.

## 2  Introduction of CIBU into the LIMA scheme

### 2.1  General considerations

In contrast to the work of Yano and Phillips (2011), where large and small graupel particles fuelled the CIBU process, we consider collisions involving two types of precipitating ice here: small ice particles growing by deposition and aggregation (aggregates including dendritic pristine ice crystals

with a size larger than ∼150 $\mu$m) and large graupel particles growing by riming. Collisions between graupel particles of different sizes are not considered because, according to Griggs and Choularton (1986), rime is very unlikely to fragment in natural clouds. For the proposed parameterization of CIBU, an impact velocity of the graupel particles that is well above 1 m s$^{-1}$ is imposed so as to stay in the break-up regime of the aggregates. This is achieved by selecting the size range of the aggregates and the graupel particles to enable CIBU.

A general form of the equation describing the CIBU process can be written

$$\frac{\partial n_i}{\partial t} = \alpha n_s n_g \tag{1}$$

where $n$ is the particle size distribution of the cloud ice (subscript "$i$"), the snow-aggregates ("$s$") and the graupel particles ("$g$"). The parameter $\alpha$ is the snow-aggregate-graupel collision kernel multiplied by $\mathcal{N}_{sg}$, the number of ice fragments produced per collision. An expression for $\alpha$, which does not include thermal and mechanical energy effects, is

$$\alpha = \mathcal{N}_{sg} V_{sg} \frac{\pi}{4} D_g^2 \tag{2}$$

where $V_{sg}$ is the impact velocity of a graupel particle of size $D_g$ at the surface of the aggregate.

In Eq. 2, it is assumed that the size of the aggregate is negligible compared to $D_g$. $V_{sg}$ is expressed as the difference in fall speed between the colliding graupel and the aggregate target so $V_{sg} = (\rho_{00}/\rho_a)^{0.4} \times (c_g D_g^{d_g} - c_s D_s^{d_s})$ using the generic formula of the particle fall speeds with the air density correction of Foote and du Toit (1969) due to the drag force exerted by the particles during their fall. The parameter $\rho_{00}$ is the reference air density $\rho_a$ at the reference pressure level.

As introduced above, and suggested in Yano and Phillips (2011), the impact velocity $V_{sg}$ should be large enough to enable CIBU. An easy way to achieve this is to restrict the size of the aggregates to the range [$D_{smin}$=0.2 mm, $D_{smax}$=1 mm] and to introduce a minimum size of $D_{gmin}$=2 mm for the graupel particles. The reasons for these choices are discussed below. The lower bound value of the aggregates, $D_{smin}$, is such that the collision efficiency with a graupel particle approaches unity. For $D_s < D_{smin}$, large crystals or aggregates stay outside the path of capture which explains the observation of bimodal ice spectra. Field (2000) reported minimum values of 150-200 µm for $D_{trough}$, a critical size separating cloud ice and aggregate regimes. The $D_{smin}$ value is also consistent with an upper bound of the cloud ice crystal size distribution resulting from the critical diameter of 125 µm to convert cloud ice to snow by deposition (see Harrington et al. (1995) for the original and analytical developments and Vié et al. (2016) for the implementation in LIMA). The choice of round numbers for $D_{smax}$ and $D_{gmin}$ is above all dictated by the empirical rule that $V_{sg}$ >1 m s$^{-1}$. With the setup in LIMA which is [$c_x$, $d_x$] = [5.1, 0.27] for "$x = s$" and [124, 0.66] for "$x = g$" in MKS units, we obtain $V_{sg}$>1.26 m s$^{-1}$ at ground level.

The number of ice fragments produced by a collision, $\mathcal{N}_{sg}$, is the critical parameter for ice multiplication. From scaling arguments Yano and Phillips (2011) recommended taking $\mathcal{N}_{sg} = 50$. Recently Yano and Phillips (2016) introduced a notion of random fluctuations into the production of

fragments which leads to a stochastic equation of the ice crystal concentration. The parameterization of $\mathcal{N}_{sg}$ as a function of collisional kinetic energy (Phillips et al., 2017) enables a treatment of the fragmentation that depends on the ice crystal type. All these results stem from Fig. 6 in Vardiman (1978) which suggests that $\mathcal{N}_{sg}$ is a function of momentum change, $\Delta M_g$, after the collision. As $\Delta M_g \sim 0.1$ g cms$^{-1}$ for $D_g$=2 mm, the corresponding $\mathcal{N}_{sg}$ lies between 10 (for collision with plane dendrites) and 40 (for rimed spatial crystals). These values are consistent with those found by Yano and Phillips (2011) for rimed assemblages. In conclusion, it is tempting to run both deterministic and stochastic simulations to test the sensitivity of the parameterization to $\mathcal{N}_{sg}$ in the range suggested by laboratory experiments. In the following $\mathcal{N}_{sg}$ is set successively to 0.1 (weak effect) implying one fragment per ten collisions, 1.0 (moderate effect) and to 10.0 or even 50.0 (strong effect). Additional experiments were performed by first generating a random variable $X$ uniformly distributed over [0.0, 1.0] and then applying an empirical formula, $\mathcal{N}_{sg} = 10^{2.0 \times X - 1.0}$, to generate values of $\mathcal{N}_{sg}$ in the interval [0.1, 10.0]. The randomization of $\mathcal{N}_{sg}$ reflects the fact that the number of fragments depends on the positioning of the impact, on the tip or on the body of the fragile particle, and also on the energy lost by the possible rotation of the residual particle.

## 2.2 Characteristics of the LIMA microphysics scheme

The LIMA microphysics scheme (Vié et al., 2016) includes a representation of the aerosols as a mixture of Cloud Condensation Nuclei (CCN) and Ice Freezing Nuclei (IFN) with an accurate budget equation (transport, activation or nucleation, and scavenging by rain) for each aerosol type. The CCN are selectively activated to produce cloud droplets which grow by condensation and coalescence to produce rain drops (Cohard and Pinty, 2000). The ice phase is more complex as we consider nucleation by deposition on insoluble IFN (black carbon and dust) and nucleation by immersion (glaciation of tagged droplets formed on partially soluble CCN containing an insoluble core). Homogeneous freezing of the droplets is possible when the temperature drops below -35° C. The Hallett-Mossop mechanism generates ice crystals during the riming of the graupel and the snow-aggregates. The H-M efficiency depends strongly on the temperature and on the size distribution of the droplets (Beheng, 1987). The initiation of the snow-aggregates category is the result of depositional growth of large pristine crystals beyond a critical size (Harrington et al., 1995). Aggregation and riming are computed explicitly. Heavily rimed particles (graupel) can experience a dry or wet growth mode. The freezing of raindrops by contact with small ice crystals leads to frozen drops which are merged with the graupel category. The melting of snow-aggregates leads to graupel and shed raindrops while the graupel particles melt directly into rain. Sedimentation is considered for all particle types. The snow-aggregates and graupel particles are characterized by their mixing ratios only. The LIMA scheme assumes a strict saturation of the water vapour over the cloud droplets while the small ice crystals are subject to super- or under-saturated conditions (no instantaneous equilibrium).

## 2.3 Representation of CIBU in the LIMA scheme

In a 2-moment bulk scheme, the zeroth order (total number concentration) and "b$^{\text{th}}$" order (mixing ratio)[2] moments of the size distributions are computed. From Eqs.1 and 2, the CIBU tendency of the number concentration of the cloud ice $N_i$ (here in $\# \, \text{kg}^{-1}$) can be written as:

$$\frac{\partial N_i}{\partial t} = \frac{\mathcal{N}_{sg}}{\rho_{dref}} \frac{\pi}{4} \left( \frac{\rho_{00}}{\rho_{dref}} \right)^{0.4} \int\limits_{D_{smin}}^{D_{smax}} n_s(D_s) \left\{ \int\limits_{D_{gmin}}^{\infty} D_g^2 (c_g D_g^{d_g} - c_s D_s^{d_s}) n_g(D_g) \mathrm{d}D_g \right\} \mathrm{d}D_s \qquad (3)$$

where $\rho_{dref}(z)$ is a reference density profile for dry air (Meso-NH is anelastic) and a further approximation $\rho_a = \rho_{dref}$ is applied.

In LIMA, the size distributions follow a generalized gamma law:

$$n(D)\mathrm{d}D = N \frac{\alpha}{\Gamma(\nu)} \lambda^{\alpha\nu} D^{\alpha\nu-1} e^{-(\lambda D)^\alpha} \mathrm{d}D$$

where $\alpha$ and $\nu$ are fixed shape parameters, $N$ is the total number concentration and $\lambda$ is the slope parameter. With the definition of the moments $M_x^{INC}(p; X)$ of the incomplete gamma law given in Appendix A, integration of Eq. 3 leads to:

$$\frac{\partial N_i}{\partial t} = \frac{\mathcal{N}_{sg}}{\rho_{dref}} \frac{\pi}{4} \left( \frac{\rho_{00}}{\rho_{dref}} \right)^{0.4} N_s N_g \times$$

$$\left\{ c_g \left( M_s^{INC}(0; D_{smin}) - M_s^{INC}(0; D_{smax}) \right) \left( M_g(2 + d_g) - M_g^{INC}(2 + d_g; D_{gmin}) \right) \right.$$

$$\left. - c_s \left( M_s^{INC}(d_s; D_{smin}) - M_s^{INC}(d_s; D_{smax}) \right) \left( M_g(2) - M_g^{INC}(2; D_{gmin}) \right) \right\} \qquad (4)$$

with $N_s = C_s \lambda_s^{x_s}$ and $N_g = C_g \lambda_g^{x_g}$. The set of parameters used in LIMA is $C_s = 5$, $C_g = 5. \times 10^5$, $x_s = 1$, $x_g = -0.5$. These values were chosen to generalize the classical Marshall-Palmer law, $n(D) = N_0 \exp(-\lambda D)$, a degenerate form of the generalized gamma law when $\alpha = \nu = 1$, leading to a total concentration $N = N_0 \lambda^{-1}$ with a fixed intercept parameter $N_0$.

Concerning the mixing ratios, the mass of the newly formed cloud ice fragments is simply taken as the product of the mean mass of the pristine ice crystals by the $N_i$ tendency (Eq. 3). The mass loss of the aggregates after collisional break-up is equal to the mass of the ice fragments. The mass of the graupel is unchanged. The mass transfer from aggregates to small ice crystals is constrained by the mass of individual aggregates that may break up completely. This limiting mixing ratio tendency is given by:

$$\frac{\partial r_i}{\partial t} = -\frac{\partial r_s}{\partial t} = \frac{a_s}{\rho_{dref}} \frac{\pi}{4} \left( \frac{\rho_{00}}{\rho_{dref}} \right)^{0.4} \int\limits_{D_{smin}}^{D_{smax}} D_s^{b_s} n_s(D_s) \left\{ \int\limits_{D_{gmin}}^{\infty} D_g^2 (c_g D_g^{d_g} - c_s D_s^{d_s}) n_g(D_g) \mathrm{d}D_g \right\} \mathrm{d}D_s.$$

---

[2]Ice mixing ratios are computed by integration over the size distribution of the mass of individual particles given by a mass-size relationship $m(D) = aD^b$, a power law with a non-integer exponent "b"

(5)

In the above expression the mass of an aggregate of size $D_s$ is given by $a_s D_s^{b_s}$ with $a_s$ set to 0.02 and $b_s$ to 1.9 in LIMA, meaning that aggregates are practically two-dimensional particles. After integration the mixing ratio tendencies are expressed as:

$$\frac{\partial r_i}{\partial t} = -\frac{\partial r_s}{\partial t} = \frac{a_s}{\rho_{dref}}\frac{\pi}{4}\left(\frac{\rho_{00}}{\rho_{dref}}\right)^{0.4} N_s N_g \times$$

$$\left\{ c_g\left(M_s^{INC}(b_s; D_{smin}) - M_s^{INC}(b_s; D_{smax})\right)\left(M_g(2+d_g) - M_g^{INC}(2+d_g; D_{gmin})\right)\right.$$

$$\left. - c_s\left(M_s^{INC}(b_s+d_s; D_{smin}) - M_s^{INC}(b_s+d_s; D_{smax})\right)\left(M_g(2) - M_g^{INC}(2; D_{gmin})\right)\right\} \quad (6)$$

This expression is independent of the number of fragments $\mathcal{N}_{sg}$.

## 3 Simulation of a 3-dimensional deep convective case

The test case is illustrated by idealized numerical simulations of the 10 July 1996 thunderstorm in the STERAO (Dye et al., 2000). This case is characterized by a multicellular storm which becomes supercellular after 2 hours. The simulations were initialized with the sounding over north eastern Colorado given in Skamarock et al. (2000) and convection was triggered by three 3K-buoyant bubbles aligned along the main diagonal of the X,Y plane along the wind axis. Meso-NH was run for 5 hours over a domain with $320 \times 320$ grid points and 1 km horizontal grid spacing. There were 50 unevenly spaced vertical levels up to a height of 23 km. With the exception of the wind components advected with a fourth-order scheme, all the fields, including microphysics, were transported by an accurate, conservative, positive-definite Piecewise Parabolic Method scheme (Colella and Woodward, 1984). There were no surface fluxes. The 3D turbulence scheme of Meso-NH was used. Open lateral boundary conditions were imposed. The upper level damping layer of upward moving gravity waves started above 12,500 m.

The aerosols were initialized as for the simulated squall-line case studied in Vié et al. (2016). A summary is given in Table 1 for the soluble Cloud Condensation Nuclei (CCN) and for the insoluble Ice Freezing Nuclei (IFN). Homogeneous vertical profiles are assumed for the aerosols. Although the LIMA scheme incorporates size distribution parameters and differentiates between the chemical compositions of the CCN and the IFN, the characteristics of the five aerosol modes are standard for the simulations shown here, except for the sensitivity of CIBU to the initial concentration of the IFN which is explored in Section 3.5.

## 3.1 Impact on precipitation

Figure 1 shows the accumulated precipitation at ground level after 4 hours of simulation for the four experiments corresponding to $\mathcal{N}_{sg}$=0.0, 0.1, 1.0 and 10.0. The highest amount of rainfall is obtained

when the CIBU process is ignored ($\mathcal{N}_{sg}$=0.0) in Fig. 1a. Then, by increasing the CIBU efficiency ten-folds from $\mathcal{N}_{sg}$=0.1, Fig. 1b-d clearly shows a steady reduction of precipitation and a fine scale modification of the precipitation pattern. Furthermore, Fig. 1d reveals that the spread of the precipitation field, caused by the motion of the multicellular storm, is significantly reduced when $\mathcal{N}_{sg}$=10.0. The results of Fig. 1 suggest empirically that a plausible range for $\mathcal{N}_{sg}$ is between 0.1 and 10.0 fragments per collision. A value lower than 0.1 leads to a negligible effect of CIBU in the simulation, while taking $\mathcal{N}_{sg}$>10.0 has an excessive impact on the storm rainfall (the "$\mathcal{N}_{sg}$=50.0" case is not shown). In addition, Fig 2 shows the results of a simulation, called "RANDOM" hereafter, where $\mathcal{N}_{sg} \in [0.1,\ 10]$ is generated by a random process as explained above. The perturbation caused by CIBU is also noticeable in this case, it remains weak for the precipitation field. These first 3D numerical experiments show that inclusion of CIBU can modify surface precipitation strongly when $\mathcal{N}_{sg} > 10.0$ fragments per aggregate-graupel collision. Taking $0.1 < \mathcal{N}_{sg} < 10.0$ and also considering $\mathcal{N}_{sg}$ as determined from a random process seems to be a more satisfactory approach. Admittedly, $\mathcal{N}_{sg} \sim 10$ is more an order of magnitude but our conclusion is to recommend an upper bound value of $\mathcal{N}_{sg}$ that is much lower than the former $N$=50 used by Yano and Phillips (2011) with their notation in the box model.

### 3.2   Changes in the microphysics

Essentially, intensifying the CIBU process by increasing $\mathcal{N}_{sg}$ leads to higher cloud ice crystal concentrations which deplete the supersaturation of water vapour that would otherwise contribute to the deposition growth of the snow-aggregates. However, a further effect is possible because the partial mass sink of the snow-aggregate particles also slows down the flux of graupel particles, which form essentially by heavy riming and conversion of the snow-aggregates. This point is now examined by considering the ice in the high levels of the STERAO cells. Figures 3–5 reproduce the 10 minute average of the mixing ratios $r_i$, $r_s$ and $r_g$ at 12 km from the 4 experiments having $\mathcal{N}_{sg}$=0.0, 0.1, 1.0 and 10.0 after 4 hours. The increase of the cloud ice mixing ratio with $\mathcal{N}_{sg}$ is clear in the area covered by the $0.2\,\mathrm{g\,kg^{-1}}$ isocontour in Fig. 3. Simultaneously, a slight decrease of $r_s$, indicating a slow erosion of the mass of the aggregates, is visible in Fig. 4. The effect on the graupel (Fig. 5) is even smaller but appears clearly for the case $\mathcal{N}_{sg}$=10.0, where less graupel is found. A last illustration is provided by Fig. 6, showing the number concentration of cloud ice $N_i$ at a higher altitude of 15 km. Again, the increase of $N_i$ follows $\mathcal{N}_{sg}$ with an explosive multiplication of $N_i$ when $\mathcal{N}_{sg}$=10.0 ($N_i$ is well above 1000 crystals $\mathrm{kg^{-1}}$ of dry air in this case). Figure 7 summarizes the behaviour of $r_i$, $r_s$, and $r_g$ at 12 km height, and of $N_i$ at 15 km height, for the "RANDOM" simulation. A comparison with Figs 3-6 shows that the results are those expected. The examination of the microphysics fields suggests that the "RANDOM" simulation corresponds to a mean CIBU intensity intermediate between $\mathcal{N}_{sg}$=1 and $\mathcal{N}_{sg}$=10.

The analysis of the STERAO simulations continues with an examination of the vertical profiles of microphysics budgets. The profiles are 10 minute averages of all cloudy columns that contain at least $10^{-3}\,\mathrm{g\,kg^{-1}}$ of condensate at any level. The column selection is updated at each time step because of the evolution and motion of the storm. Figure 8 shows the mixing ratio profiles for three cases: $\mathcal{N}_{sg} = 0.0$, "RANDOM" and $\mathcal{N}_{sg} = 10.0$. A key feature that shows up in Fig. 8a-c is the increase of the $r_i$ peak value at 11 km altitude. This change is accompanied by a reduction of $r_s$ (more visible between cases b) and c)) and by a reduction of $r_g$, which stands out at z=8,000 m. The decrease of $r_g$, even when graupels are passive colliders for CIBU, is the result of the decrease of $r_s$ in the growth chain of the precipitating ice. The low value of the mean $r_r$ profiles, compared to the mixing ratios of the ice phase above, is explained by the fact that rain is spread over fewer grid points than the ice in the anvil is (the mixing ratio profiles are averaged over the same number of columns).

### 3.3 Budget of ice mixing ratios

This step is devoted to the microphysics tendencies (using 10 minute average again with the nomenclature of the processes provided in Table 3) of the ice mixing ratios in Fig. 9-11 to assess the impact of the CIBU process. We do not discuss the case of the liquid phase here because the tendencies (not shown) are only marginally affected by the CIBU process.

As expected, many tendencies of $r_i$ (Fig. 9a-c) are affected by the CIBU process. The main processes standing out in Fig. 9a, when CIBU is not activated, are CEDS (deposition-sublimation), essentially a gain term, and AGGS (aggregation), the main loss of $r_i$ by aggregation with a rate of $0.5 \times 10^{-3}\,\mathrm{g\,kg^{-1}\,s^{-1}}$. The loss of $r_i$ by CFRZ (drop freezing by contact) makes a moderate contribution as some raindrops are present in the glaciated part of the storm. Above $z$=10,000 m, the net loss of $r_i$ (AGGS and SEDI, the cloud ice sedimentation) is balanced by the convective vertical transport (not shown). When $\mathcal{N}_{sg}$=RANDOM, the $r_i$ tendencies are amplified, even with a modest contribution of $\sim 0.2 \times 10^{-3}\,\mathrm{g\,kg^{-1}\,s^{-1}}$ for CIBU itself. The growth of AGGS, which doubles at 10 km height, is caused by CIBU and by an increase in the convection because SEDI (a loss at this height) is amplified in response to an increase of $r_i$ in the upper levels. The CFRZ contribution is also increased. The last case, with $\mathcal{N}_{sg}$=10 (Fig. 9c) confirms a further increase of the rates except for CFRZ, interpreted here as a lack of raindrops.

The budget of the snow-aggregates mixing ratio in Fig. 10 contains many processes of equivalent importance in the range $\pm 0.05 \times 10^{-3}\,\mathrm{g\,kg^{-1}\,s^{-1}}$ but SEDS (sedimentation of snow-aggregates) dominates at $z$=11,000 m and at $z = 7,000$ m. The inclusion of CIBU (Fig. 10b-c) mostly leads to an increase of AGGS, the other processes remaining almost the same. Finally many processes contribute to the evolution of the graupel mixing ratio profiles (Fig. 11). The strongest loss is in the GMLT term (melting of graupel) that converts graupel into rain (down to $-0.3 \times 10^{-3}\,\mathrm{g\,kg^{-1}\,s^{-1}}$) while CFRZ reaches $0.15 \times 10^{-3}\,\mathrm{g\,kg^{-1}\,s^{-1}}$. The sedimentation term SEDG (sedimentation of graupel) lies between $-0.3 \times 10^{-3}\,\mathrm{g\,kg^{-1}\,s^{-1}}$ at $z = 10,000$ m and $0.15 \times 10^{-3}\,\mathrm{g\,kg^{-1}\,s^{-1}}$ at 5,000 m. An-

other noticeable effect is the sign change of DEPG (growth of graupel by deposition, $\pm 0.07 \times 10^{-3}$ g kg$^{-1}$ s$^{-1}$) showing that the water vapour is supersaturated above $z$=7,000 m and undersaturated below $z$=7,000 m on average. The relative importance of these processes does not change very much when CIBU is increased but all tendencies weaken. To sum up, the impact of CIBU is modest for the microphysics mixing ratios. The increase of ice fragments in $r_i$ is approximately compensated by an increase of AGGS (see Fig. 9 and 10).

## 3.4 Budget of cloud ice concentration

This subsection examines the behaviour of the cloud ice number concentration as a function of the strength of the CIBU process after 4 hours of simulation. Figure 12 shows that the altitude of the $N_i$ peak value decreases when $\mathcal{N}_{sg}$ increases. In the absence of CIBU ($\mathcal{N}_{sg} = 0$), the source of $N_i$ is the heterogeneous nucleation processes on insoluble IFN and on coated IFN (nucleation by immersion) which are more efficient at low temperature. Nucleation on IFN provides a mean peak value $N_i = 400$ kg$^{-1}$ at $z = 11,500$ m. In contrast, the $\mathcal{N}_{sg} = 10$ case (here scaled by a factor 0.1 for ease of reading) keeps the trace of an explosive production of cloud ice concentration, $N_i = 7,250$ kg$^{-1}$, due to CIBU. The altitude of the maximum of $N_i$ in this case ($z = 10,000$ m) is consistent with the location of the maximum value of the $r_s \times r_g$ product (see Fig. 8). The "RANDOM" simulation produces $N_i = 1100$ kg$^{-1}$ at $z = 11,000$ m, a number concentration similar to that found for the $\mathcal{N}_{sg} = 2$ case. Table 2 reports the peak amplitude of the $N_i$ profiles as a function of $\mathcal{N}_{sg}$ but after 3 hours of simulation, when the CIBU rate is strongly dominant. Additional cases were run to cover $0.1 < \mathcal{N}_{sg} < 50$ with a logarithmic progression above $\mathcal{N}_{sg} = 1.0$. The CIBU enhancement factor, CIBU$_{\text{ef}}$, was computed as $N_i(\mathcal{N}_{sg})/N_i(\mathcal{N}_{sg} = 0) - 1$ since $N_i(\mathcal{N}_{sg} = 0)$ constitutes a baseline not affected by CIBU. The results presented in Table 2 show that the growth of $N_i$ is fast when $\mathcal{N}_{sg}$ reaches $\sim$5 (CIBU$_{\text{ef}}$ rises sharply from 135% to 913% when $\mathcal{N}_{sg}$ increases from 2 to 5). Taking $\mathcal{N}_{sg} = 50$ leads to an extremely high peak value of $N_i$.

The $N_i$ tendencies are the subject of Fig. 13. Many processes are involved during the temporal integration of $N_i$. The $\mathcal{N}_{sg} = 0$ case confirms the importance of the heterogeneous nucleation process by deposition (HIND, see Table 3) and, to a lesser degree, by immersion (HINC) at 8 km height. HIND peaks at three altitudes with two sources of IFN (Table 1). This case also reveals the importance of the HMG (Hallett-Mossop on Graupel, 1.3 kg$^{-1}$s$^{-1}$) and HMS (Hallett-Mossop on Snow, 0.85 kg$^{-1}$s$^{-1}$) processes. Here, we consider that H-M also operates for the snow-aggregates because this category of ice includes lightly rimed particles that can rime further to form graupel particles. These processes are first compensated by AGGS (capture of cloud ice by the aggregates). There is also a loss of cloud ice due to CFRZ and CEDS with the full sublimation of individual cloud ice crystals which replenish the IFN reservoir. The sedimentation profile transports ice from the cloud top (SEDI<0) to mid-level cloud (SEDI>0). Then, taking $\mathcal{N}_{sg}$ = RANDOM shows the domination of the CIBU process, which reaches 2.5 kg$^{-1}$s$^{-1}$ at 5 km height. The enhancement of

HIND at cloud top can also be noted. The CIBU source of ice crystals is balanced by an increase of AGGS and, above all, of CEDS (here CEDS represents the sublimation of the ice crystal concentration when the crystals are detrained in the low level of the cloud vicinity, such as below the anvil). Finally, the $\mathcal{N}_{sg} = 10$ case demonstrates the reality of the exponential-like growth of $N_i$ because the three main driving terms (CIBU, CEDS and AGGS) are growing at a similar rate, which is multiplied

by a factor of approximately 5.

### 3.5   Sensitivity to the initial concentration of freezing nuclei

The purpose of the last series of experiments was to look more closely at the sensitivity of the cloud ice concentration to $N_{IFN}$, the initial concentration of the IFN. Numerical simulations were run with $N_{IFN}$ decreasing ten-fold from 100 dm$^{-3}$ to 0.001 dm$^{-3}$ for each IFN mode (see Table 1).

Two different cases were considered. In the first case, CIBU was activated with the RANDOM set-up while, in the second case, CIBU effects were ignored. All the results are summarized in the plots of Fig. 14.

Figure 14a shows that $N_i$ concentrations did not change very much for a wide range of $N_{IFN}$ concentrations, which were varied ten-fold. This clearly illustrates the predominance of the CIBU

effect for current IFN concentrations, which disconnects $N_i$ concentrations from the underlying abundance of IFN particles. Likewise, the small hump superimposed on all profiles at 5,000 m height reveals a residual effect of the Hallett-Mossop process. Another remarkable feature is that a fairly low IFN concentration ($N_{IFN} = 0.001$ dm$^{-3}$) suffices to initiate the CIBU process and to reach $N_i \sim 500$ kg$^{-1}$. In contrast, and in the absence of CIBU (Fig. 14b), the $N_i$ profiles show a sensitivity to IFN

nucleation that is, indeed, difficult to interpret because of the non-monotonic trend of the $N_i$ profiles with respect to $N_{IFN}$. Some insight can be gained by checking the concentration of the nucleated IFN of the first IFN mode (dust particles). In Fig. 14c, the IFN profiles are rescaled (multiplication by an appropriate number of powers of ten) to be comparable. This is equivalent to computing an IFN nucleation efficiency. The important result here is that the number of nucleated IFN evolves in close

proportion to the initially available IFN concentrations, meaning that, as expected, the nucleating properties of the IFN do not depend on the IFN concentration. The last plot (Fig. 14d) reproduces the normalized differences of $N_i$ profiles between twin simulations performed with and without CIBU. Although simulations using the same initial concentration $N_{IFN}$ may diverge because of additional non-linear effects (vertical transport, enhanced or reduced cloud ice sink processes), the

figure gives an indication of the bulk sensitivity of CIBU to the IFN. The enhancement ratio due to CIBU remains low (less than 1 for $N_{IFN} \sim 100$ dm$^{-3}$) but can reach a factor of 20 at 9,000 m height in the case of moderate IFN concentration *i.e.* $N_{IFN} \sim 1$ dm$^{-3}$. The behaviour of LIMA can be explained in the sense that increasing $N_{IFN}$ too much leads to smaller pristine crystals that need a longer time to grow before being included to the next category of snow-aggregates because such

inclusion is size-dependent (see Harrington et al. (1995) and Vié et al. (2016)). On the other hand, a

low concentration of $N_{IFN}$ initiates fewer snow-aggregates and thus fewer graupel particles, so the whole CIBU efficiency is also reduced. Consequently, this study confirms the essential role of CIBU in compensating for IFN deficit when cloud ice concentrations are increasing.

## 4  Simulation of a 3-dimensional idealized supercell storm with varying atmospheric stability

The idealized sounding of Weisman and Klemp (1982, 1984) was appealing to use for this test case (referred to as WK) because the intensity of the CAPE can be easily modified by changing a reference water vapour mixing ratio. The environmental conditions of the simulations were close to those of the STERAO case with the same set-up for the physics and the aerosol characteristics. The simulation domain was 180×180 grid points at 1 km resolution and 70 levels with a mean vertical

grid spacing of 350 m. Convection was triggered by a domain-centered single 2K-buoyant air parcel of 10 km radius and 3 km height. The base of the upper level Rayleigh damper was set at 15 km above ground level.

Meso-NH was initialized with the analytic sounding of Weisman and Klemp (1984) with low 2-dimensional shear. The hodograph in Fig. 15 features a three-quarter-cycle with a constant wind of

6.4 m s$^{-1}$ (in modulus) above the height of 5 km. When running Meso-NH a constant translation speed ($U_{trans}$=5 m s$^{-1}$ and $V_{trans}$=1 m s$^{-1}$) was added to the wind to keep the convection well centred in the domain of simulation. As explained in Weisman and Klemp (1982), buoyancy was varied by altering the magnitude of the surface water vapour mixing ratio $q_{v0}$ keeping with the Weisman and Klemp (1984) notation. Three water vapour profiles were defined by taking $q_{v0} = 13.5$

g kg$^{-1}$, hereafter the "Low" CAPE case of 1970 J kg$^{-1}$; $q_{v0} = 14.5$ g kg$^{-1}$ as the "Mid" CAPE case of 2400 J kg$^{-1}$, and $q_{v0} = 15.5$ g kg$^{-1}$, the "High" CAPE case with 2740 J kg$^{-1}$. Four experiments of 4 hours each were performed for each CAPE case by using different magnitudes of $\mathcal{N}_{sg}$.

### 4.1  Sensitivity to mean ice concentrations

Figure 16 shows the mean concentrations of small ice crystals between 9.5 and 10.5 km levels

plotted on a log scale after 4 hours of simulation. In addition, two CTH (Cloud Top Height) contours delineate the 11 km (dotted line) and 13 km (solid line) levels. The $\mathcal{N}_{sg}$ =0, RANDOM, 10 and 50 cases, are explored for each sounding ("Low", "Mid" and "High" CAPE). In the absence of CIBU (first row in Fig. 16), the cloud ice concentrations $N_i$ are in the range of what was simulated for the STERAO case (see Figs. 6 and 7d). The $N_i$ peak values do not increase with the initial CAPE

(Figs 16a-b) but the area of CTH>11 km is larger in the "Mid CAPE" case. The "High CAPE" case is a little bit more difficult to analyse because of earlier development of the convection, spreading out ahead of the main system. This shows up in the "Low" and "Mid" CAPE cases but the $N_i$ peak values of the "High" CAPE case are in the same range as for the "Low" CAPE case, meaning that higher environmental instabilty is not decisive in fixing the $N_i$ peak values. In the $\mathcal{N}_{sg}$ =10 and 50

cases, we retrieve the dramatic increase of $N_i$ due to increasing CIBU efficiency. The enhancement is locally as high as 1,000 fold in the strongest case ($\mathcal{N}_{sg} = 50$). There are also other noteworthy features: an increase of the $N_i$ area coverage with $\mathcal{N}_{sg}$ (less visible in the "Low" CAPE case) and a higher CTH which exceeds 13 km for the "Mid" and "High" CAPE cases. All these observations strongly suggest that convection is invigorated when the CIBU effect is increased. In contrast, the

simulations run with $\mathcal{N}_{sg}$=RANDOM using values taken in the 0.1-10 range (see Section 2.1), show a moderate effect of CIBU. Locally, $N_i$ values reach $1 \times 10^4$ kg$^{-1}$, which is one hundred times lower than $N_i$ peak values in the $\mathcal{N}_{sg} = 50$ cases but approximately, ten times higher than in the "no CIBU" case ($\mathcal{N}_{sg} = 0$). Finally the simulation results suggest that the $\mathcal{N}_{sg}$ parameter could be constrained by satellite data because of the sensitivity of CIBU to the cloud ice coverage and the

cloud top height.

## 4.2   Sensitivity to precipitation

The 4-hour accumulated precipitation maps are presented in Fig. 17. On each row, precipitation increases from the "Low" to "High" CAPE cases. This is because the CAPE is enhanced by the addition of more water vapour. Looking at the sensitivity of the accumulated precipitation to $\mathcal{N}_{sg}$,

it is not as easy to draw a general conclusion on the decrease of the precipitation peak with $\mathcal{N}_{sg}$ as for the STERAO case (see section 3.1). The reason is the highly concentrated precipitation field, which leads to a sharp gradient around the location of the peak value. However, the decrease of the precipitation with $\mathcal{N}_{sg}$ is observed in the "Low" and "High" CAPE cases. In the "Mid" case, the precipitation peak value remains high when $\mathcal{N}_{sg} = 50$ but the area where the precipitation is

less than 10 mm shrinks continuously. The reduction of the area where the precipitation amount is greater than 10 mm when $\mathcal{N}_{sg}$ is increased, was found in all CAPE cases (not shown).

    In conclusion, the simulations illustrate the fact that the precipitation patterns are affected by the value of the $\mathcal{N}_{sg}$ parameter. When $\mathcal{N}_{sg}$ is increased from zero to 50, the precipitation is reduced, either for the peak value or, at least for the precipitating area. This is consistent with our previous

results concerning the STERAO case. The conversion efficiency of the small ice crystals to precipitating ice particles is lower when the cloud ice concentration is high because the deposition growth of individual small crystals is limited by the amount of supersaturated water vapour available.

## 4.3   Sensitivity to the ice thickness

This last analysis is concerned with the ice thicknesses (or ice water paths) computed as the integrals

along the vertical of $\rho_{dref} r_x$ where $r_x$ refers to the mixing ratio with $x \in i, s, g$ standing for the cloud ice, the snow-aggregates and the graupel-hail, respectively. Fig. 18 displays the total ice thickness, a sum of three terms, in mm (coloured area) with the superimposed cloud ice thickness (THIC), contoured at 1 mm. A remarkable feature is that the total ice thickness seems almost insensitive to the CIBU process for a given CAPE case: there is no great modification in the plots when moving

from $\mathcal{N}_{sg} = 0$ to $\mathcal{N}_{sg} = 50$. This is in contrast with the 1 mm contour of cloud ice thickness, the enclosed area of which increases with $\mathcal{N}_{sg}$ as shown in Fig. 18. A rise in the maximum value of THIC was also expected for increasing values of $\mathcal{N}_{sg}$. However, the increase of $\text{THIC}_{\text{max}}$ with the CAPE is much more moderate between the "Low" and "High" cases because a higher CAPE regime with higher humidity tends to favour the horizontal spread of the cloud ice mass.

## 5 Summary and perspectives

The aim of this work was to study a parameterization of the Collisional Ice Break-Up for the bulk 2-moment microphysics scheme LIMA running in a cloud resolving mesoscale model (Meso-NH in our case). While the process is suspected to occur in real clouds, it is not included in current bulk microphysics schemes. Because of uncertainties to physically describe the ice break-up process, the 445  present parameterization has been kept as simple as possible. It considers only collisions between small aggregates and large dense graupel particles. The number of ice fragments that results from a single collision, $\mathcal{N}_{sg}$, is a key parameter, which is estimated from only a very small number of past experiments (Vardiman, 1978). This study suggests an upper bound on $\mathcal{N}_{sg}$ because of the sensitivity of $\mathcal{N}_{sg}$ to the simulated precipitation. We found that taking $\mathcal{N}_{sg} > 10$ significantly reduced surface 450  precipitation. This is problematic because most of the cloud schemes (running without the CIBU process) are carefully verified for quantitative precipitation forecasts in operational applications. Furthermore, we suggest that $\mathcal{N}_{sg}$ could be considered as the realization of a random process that reduces the impact of CIBU on the precipitation and also that delicate radiating crystals undergoing fragmentation lead to a variety of crystals with a missing arm or to many irregular fragments as 455  illustrated and discussed by Hobbs and Farber (1972). As a result, it has been shown, that running LIMA with $\mathcal{N}_{sg} > 10$ for the STERAO and WK deep convection cases taken from Skamarock et al. (2000) and Weisman and Klemp (1982, 1984) respectively, alters surface precipitation because the conversion of cloud ice crystals to precipitating ice is slowed down. In any case, the increase of the number concentration of the small ice crystals due to the application of CIBU is clearly substantial 460  (up to 1,000 fold in the WK simulations with $\mathcal{N}_{sg} = 50$).

The microphysics perturbation due to the activation of CIBU has been studied in detail for the STERAO case by looking at the profiles of the mixing ratios, ice concentrations and corresponding budget terms. In particular, the CIBU effect on the pristine ice and aggregate mixing ratios is compensated by an enhancement of the capture of the small crystals by the aggregates. The sensitivity 465  of the ice concentration to $\mathcal{N}_{sg}$ is demonstrated with a mean multiplication factor as high as 25 for $\mathcal{N}_{sg} = 10$. The last study on the sensitivity of the simulations to the initial IFN concentration showed that CIBU was mostly efficient for current IFN concentrations of $\sim 1 \text{ dm}^{-3}$. Furthermore, the CIBU process was still active for very low IFN concentrations, down to $0.001 \text{ dm}^{-3}$, which were sufficient to initiate the ice phase.

The effects of CIBU have been confirmed by a second series of WK simulations. The enhancement of the cloud ice concentration is very high when $\mathcal{N}_{sg} > 10$ and a loss of surface precipitation is found in terms of the peak value and the reduction of the precipitating areas. Higher ice concentrations lead to a larger coverage of ice clouds and higher cloud tops for the most vigorous convective cells. In constrast, the total ice thickness is almost insensitive to CIBU. An increase of cloud ice mass with

$\mathcal{N}_{sg}$ is balanced by a slight decrease of the precipitating ice (aggregates and graupels).

   The proposed parameterization is very easy to implement. It would be useful to evaluate it in other microphysics schemes where the conversion of the cloud ice and the growth of precipitating ice (aggregates and rimed particles) are treated differently. Adjustments to the scheme can be revised as soon as laboratory experiments are available to enable more precise fixing of the sizes and the shapes

of the crystals that break following collisions, and also to examine any possible thermal effect and to estimate the variety of fragment numbers more accurately. Another way to determine the acceptable range of values for $\mathcal{N}_{sg}$ is to work with satellite data, as the WK experiments demonstrated an enhancement of the cloud top ice cover with $\mathcal{N}_{sg}$ (and possibly the cloud top height).

   With new imagers, counters and improvements in data analysis (Ladino et al., 2017), more and

more evidence is being presented that ice multiplication is an essential process in natural deep convective clouds. However, the explanation of anomalously high ice crystal concentrations is still difficult to link to a precise process (Rangno and Hobbs, 2001; Field et al., 2017). Therefore the next step in the LIMA scheme will be to introduce the shattering of raindrops during freezing as proposed by Lawson et al. (2015) in order to complete the LIMA scheme, since the different ingredients of

raindrops and small ice crystals offer another pathway for ice multiplication. One task will then be to study whether all the known sources of small ice crystals, nucleation and secondary ice production, are able to work together in microphysics schemes to reproduce the very high values of ice concentrations sometimes observed. Quantitative cloud data gathered in the tropics during the HAIC/HIWC (High Altitude Ice Crystals/ High Ice Water Content) field project (Leroy et al., 2015; Ladino et al.,

2017) could provide a starting point for the evaluation of the capability of high resolution cloud simulations to reproduce events where high cloud ice contents have been recorded.

## 6 Code availability

The Meso-NH code is publicly available at http://mesonh.aero.obs-mip.fr/mesonh51. Here the model development and the simulations were carried out with version "MASDEV5-1 BUG2". The modifi-

cations made to the LIMA scheme (v1.0) are available upon request from Jean-Pierre Pinty and in the Supplement related to this article, available at http://doi.org/10.5281/zenodo.1078527.

**Appendix A: Moments of the gamma and incomplete gamma functions**

The $p^{th}$ moment of the generalized gamma function (see definition in the text) is

$$M(p) = \int_0^\infty D^p n(D) \mathrm{d}D = \frac{\Gamma(\nu + p/\alpha)}{\Gamma(\nu)} \frac{1}{\lambda^p} \tag{A1}$$

where the gamma function is defined as:

$$\Gamma(x) = \int_0^\infty t^{x-1} e^{-t} \mathrm{d}t. \tag{A2}$$

The $p^{th}$ moment of the incomplete gamma function is written

$$M^{INC}(p; X) = \int_0^X D^p n(D) \mathrm{d}D. \tag{A3}$$

The algorithm of the "GAMMA_INC$(p; X)$" function (Press et al., 1992) is useful to tabulate
$M^{IN}(p; X) \times \Gamma(p)$ in addition to the "GAMMA" function algorithm of Press et al. (1992). A change
of variable is necessary to take the generalized form of the gamma size distributions into account.
As a result, $M^{INC}(p; X)$ is written:

$$M^{INC}(p; X) = M(p) \times \text{GAMMA\_INC}(\nu + p/\alpha; (\lambda X)^\alpha) \tag{A4}$$

with $M(p)$ given by Eq. A1.

*Acknowledgements.* J.-P. Pinty wishes to thank V. Phillips for discussions about his original work on the topic.
This work was done during the PhD of T. Hoarau who is financially supported by Reunion Island Regional
Council and the European Union Council. T. Hoarau thanks the University of La Réunion for supporting a short
stay at the Laboratoire d'Aérologie. Mrs Susan Becker and Callum Thompson corrected the English of the
manuscript. Preliminary computations were performed on the 36 node home-made cluster of Lab. Aérologie.
J.-P. Pinty acknowledges CALMIP (CALcul MIdi-Pyrénés) of the University of Toulouse for access to the "eos"
supercomputer where useful additional simulations were performed. T. Hoarau and C. Barthe acknowledge the
GENCI ressources for access to the "Occigen" supercomputer. This work was supported by the French national
programme LEFE/INSU through the LIMA-TROPIC project. The authors thank the reviewers and the topical
editor for their pertinent comments and meticulous review which greatly improved previous manuscripts.

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

| CCN | Aitken mode | Accumulation mode | Coarse mode |
|---|---|---|---|
| $N$ (cm$^{-3}$) | 300 | 140 | 50 |
| $d_X$ (µm) | 0.23 | 0.8 | 2.0 |
| $\sigma_X$ | 2.0 | 1.5 | 1.6 |

| IFN | Dust mode | BC+Organics mode |
|---|---|---|
| $N$ (dm$^{-3}$) | 10 | 10 |
| $d_X$ (µm) | 0.8 | 0.2 |
| $\sigma_X$ | 2.0 | 1.6 |

**Table 1.** Background CCN and IFN configuration for the STERAO idealized case simulations.

| $\mathcal{N}_{sg}$ (no unit) | 0.0 | 0.1 | 1.0 | 2.0 | 5.0 | 10.0 | 20.0 | 50 |
|---|---|---|---|---|---|---|---|---|
| $N_i$ ($\#\mathrm{kg}^{-1}$) | 790 | 940 | 1,160 | 1,860 | 8,000 | 25,670 | 62,010 | 112,740 |
| $\mathrm{CIBU}_{\mathrm{ef}}$ (%) | 0 | 19 | 47 | 135 | 913 | 3149 | 7749 | 14171 |

**Table 2.** After 3 hours of simulation, maximum value of the cloud ice number concentration $N_{i_{max}}$ as a function of the number of fragments produced per snow/aggregate-graupel collision $\mathcal{N}_{sg}$. The last row is the CIBU enhancement factor $\mathrm{CIBU}_{\mathrm{ef}}$ in percent (see text).

| Process Acronym | Description |
|---|---|
| ACC | Raindrop accretion on snow to produce graupel |
| AGGS | Snow growth by capture of cloud ice |
| BERFI | Growth of cloud ice by Bergeron-Findeisen process |
| CEDS | Deposition/sublimation of water vapour on cloud ice |
| CFRZ | Raindrop Freezing by contact with cloud ice |
| CIBU | Snow break-up by collision with graupel |
| CMEL | Conversion Melting of snow into graupel |
| CNVI | Decreasing snow converted back to cloud ice |
| CNVS | Growing cloud ice converted into snow |
| DEPG | Water vapour deposition on graupel |
| DEPS | Water vapour deposition on snow |
| DRYG | Graupel dry growth (water can freeze fully) |
| HINC | Heterogeneous nucleation by immersion |
| HIND | Heterogeneous nucleation by deposition |
| HONC | Homogeneous freezing of the cloud droplets |
| HONH | Haze homogeneous freezing |
| HMG | Droplet riming and Hallett-Mossop process on graupel |
| HMS | Droplet riming and Hallett-Mossop process on snow |
| IMLT | Melting of cloud ice |
| RIM | Riming of cloud droplets on snow to produce graupel |
| SEDI | Sedimentation of cloud ice, snow or graupel |
| WETG | Graupel wet growth (water is partially frozen) |

**Table 3.** Nomenclature of the microphysics processes of the budget profiles.

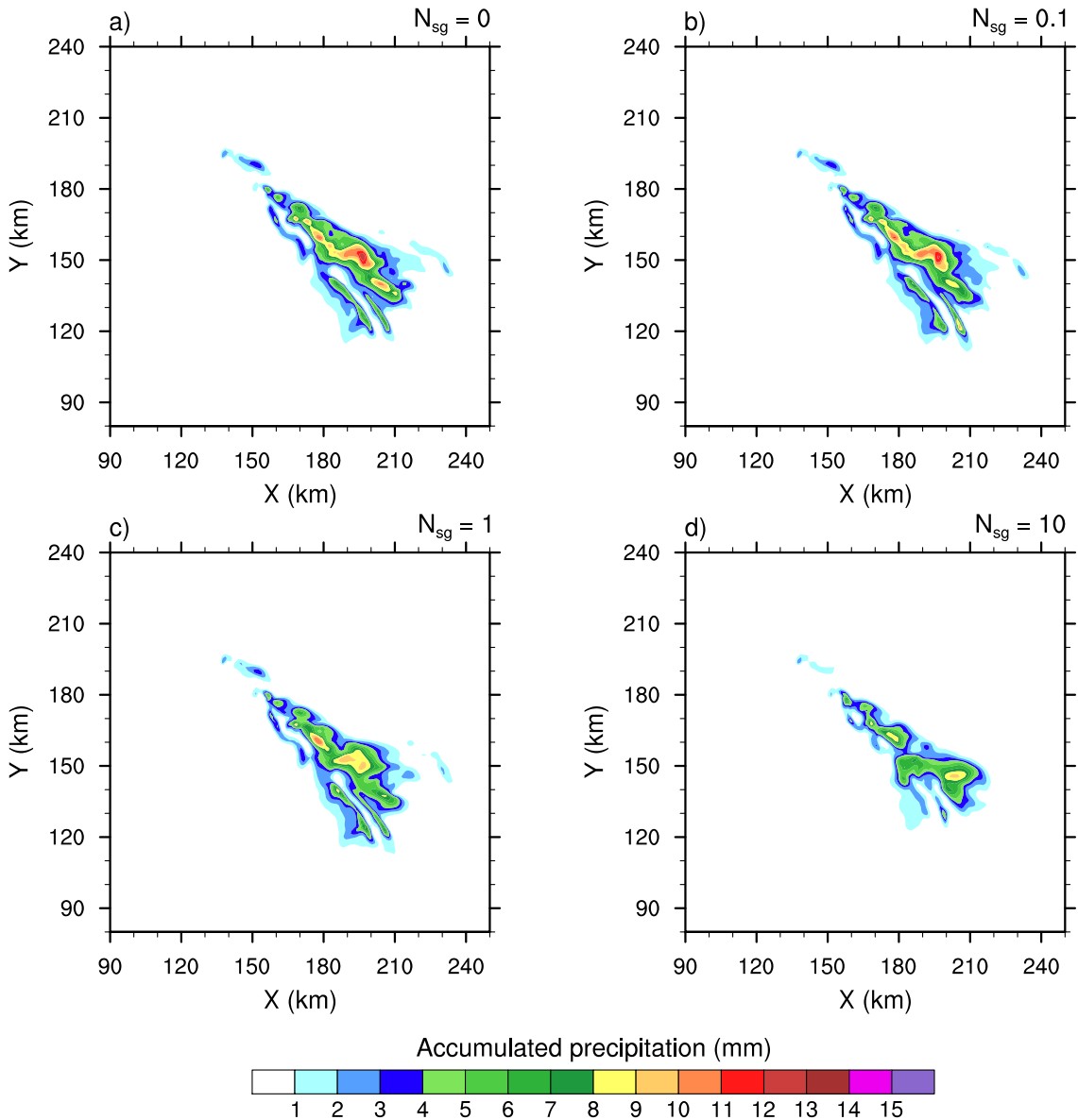

**Figure 1.** 4-h accumulated precipitation of the STERAO simulations where a) to d) refers to cases with $\mathcal{N}_{sg}$=0.0, 0.1, 1.0 and 10.0 ice fragments per collision, respectively. The plots are for a fraction of the computational domain.

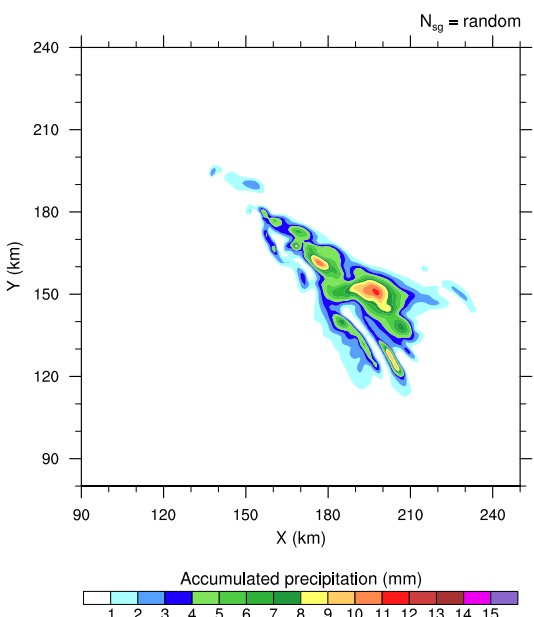

**Figure 2.** As in Fig. 1, but for the "RANDOM" simulation.

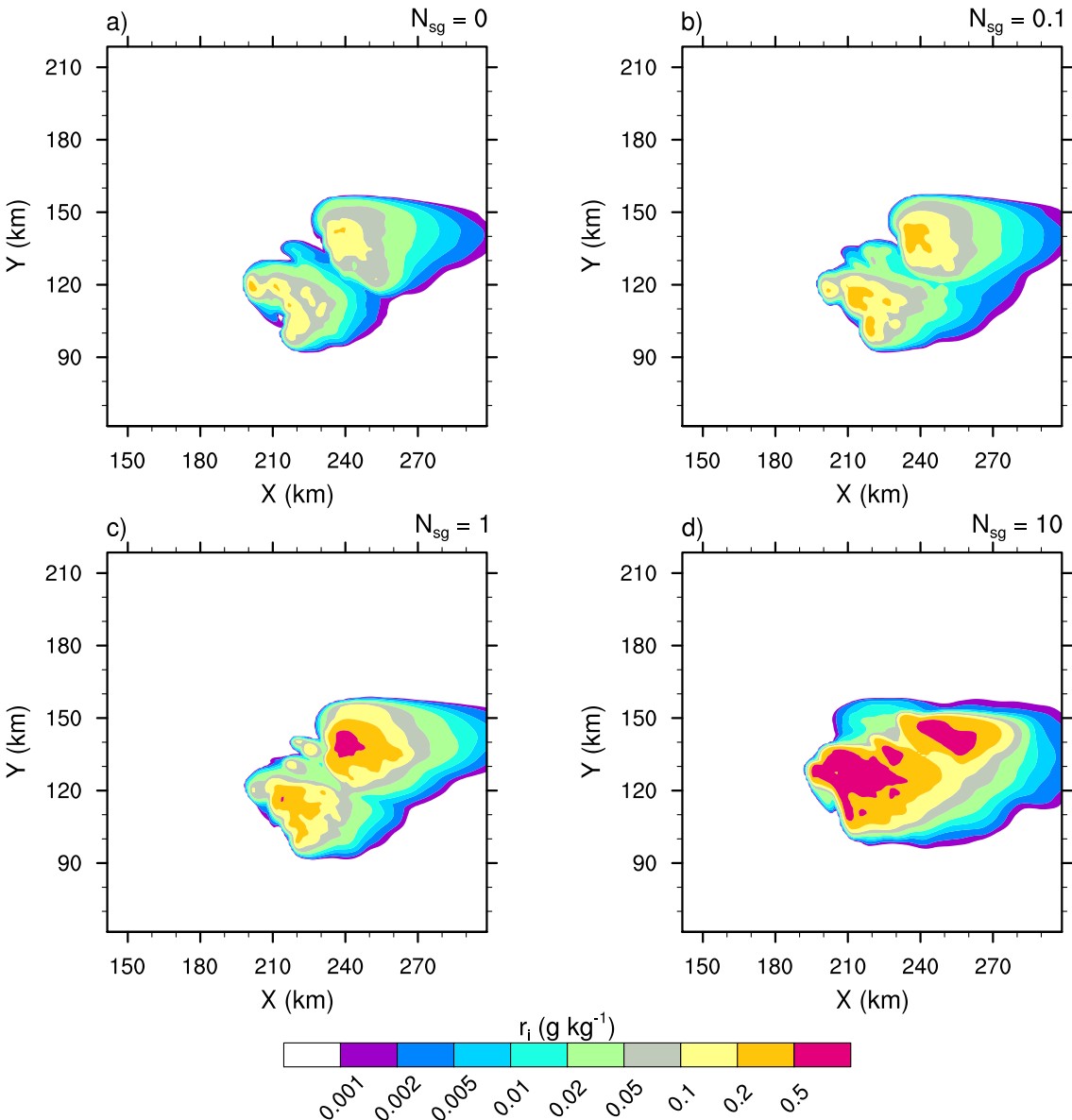

**Figure 3.** Mixing ratios of the cloud ice ($r_i$ in log scale) of the STERAO simulations at 12 km height, where a) to d) refer to cases with $\mathcal{N}_{sg}$=0.0, 0.1, 1.0 and 10.0 ice fragments per collision, respectively. The plots are for a fraction of the computational domain.

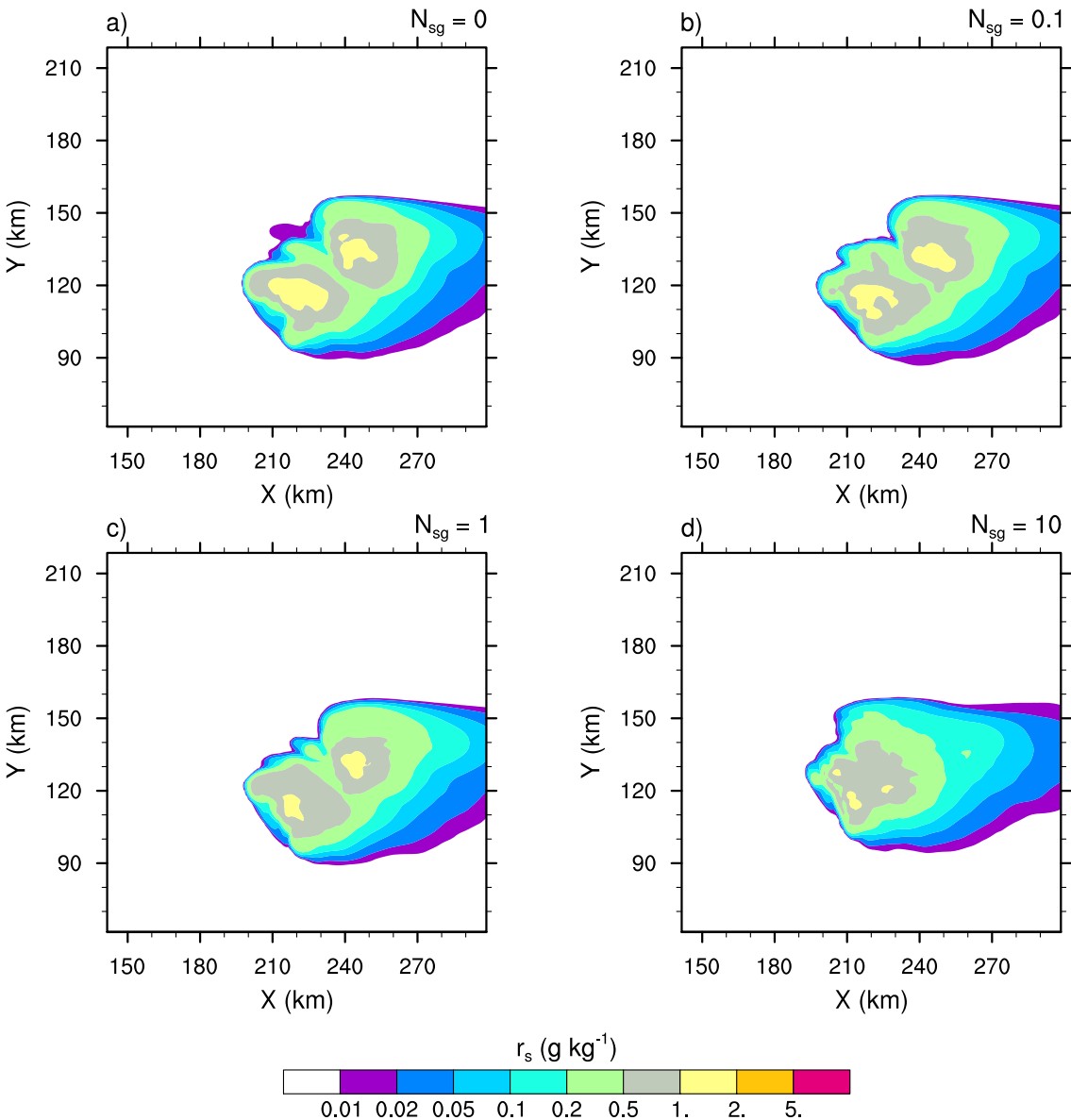

**Figure 4.** As in Fig. 3 but for the mixing ratios of snow-aggregates ($r_s$).

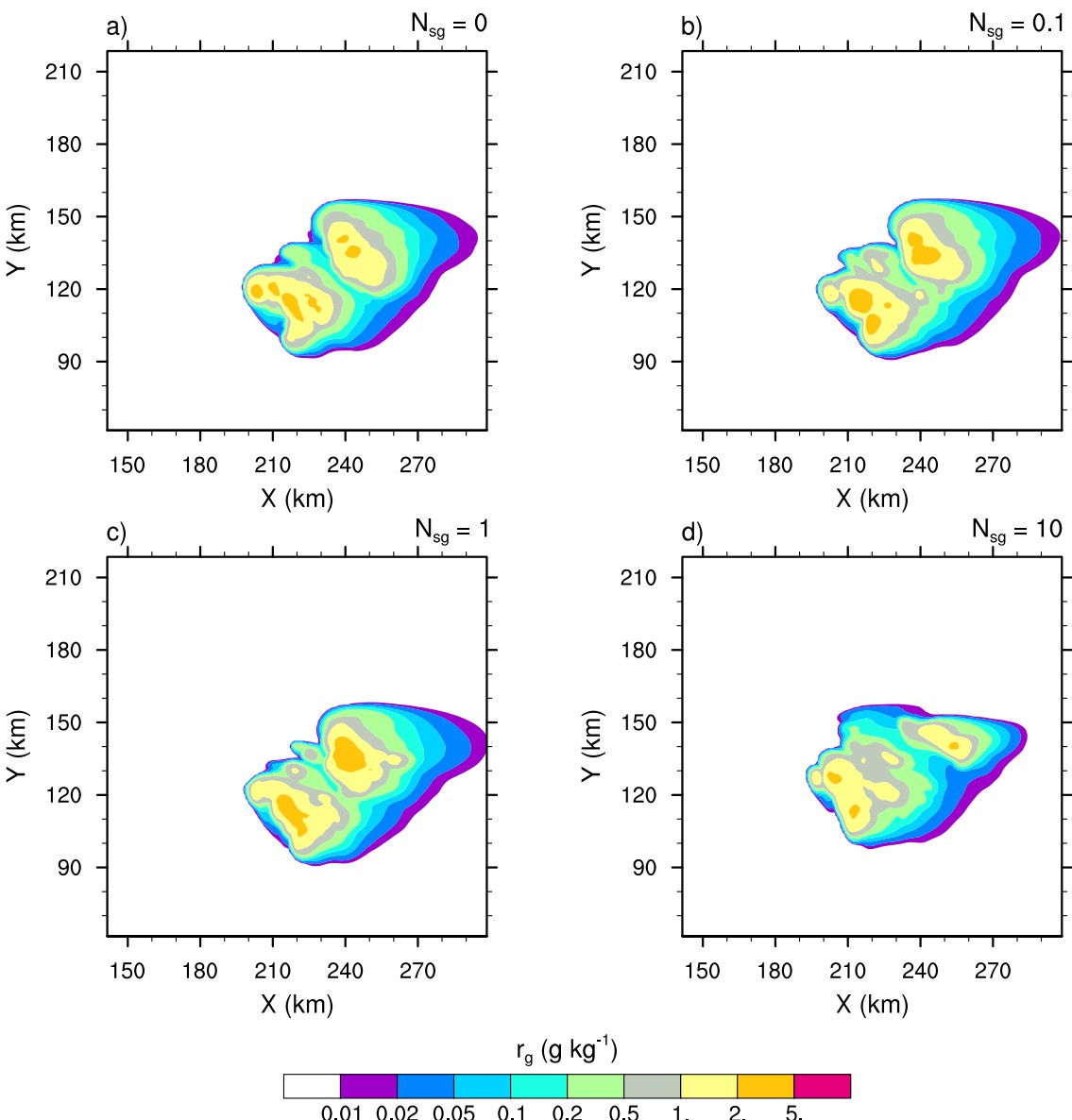

**Figure 5.** As in Fig. 3 but for the mixing ratios of graupel ($r_g$).

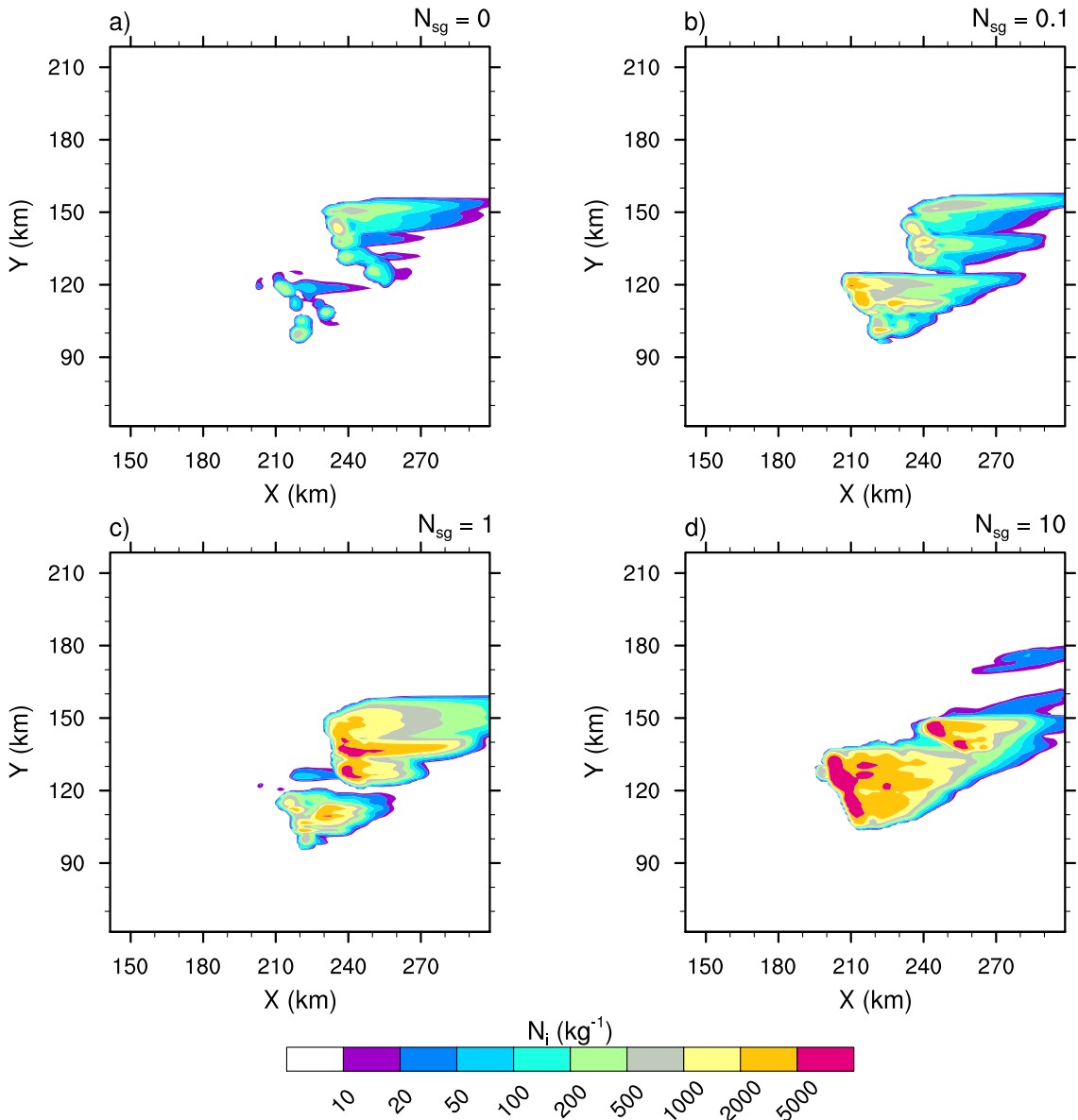

**Figure 6.** Number concentration of the cloud ice ($N_i$ in log scale) of the STERAO simulations at 15 km height, where a) to d) refer to cases with $\mathcal{N}_{sg}$=0.0, 0.1, 1.0 and 10.0 ice fragments per collision, respectively. The plots are for a fraction of the computational domain.

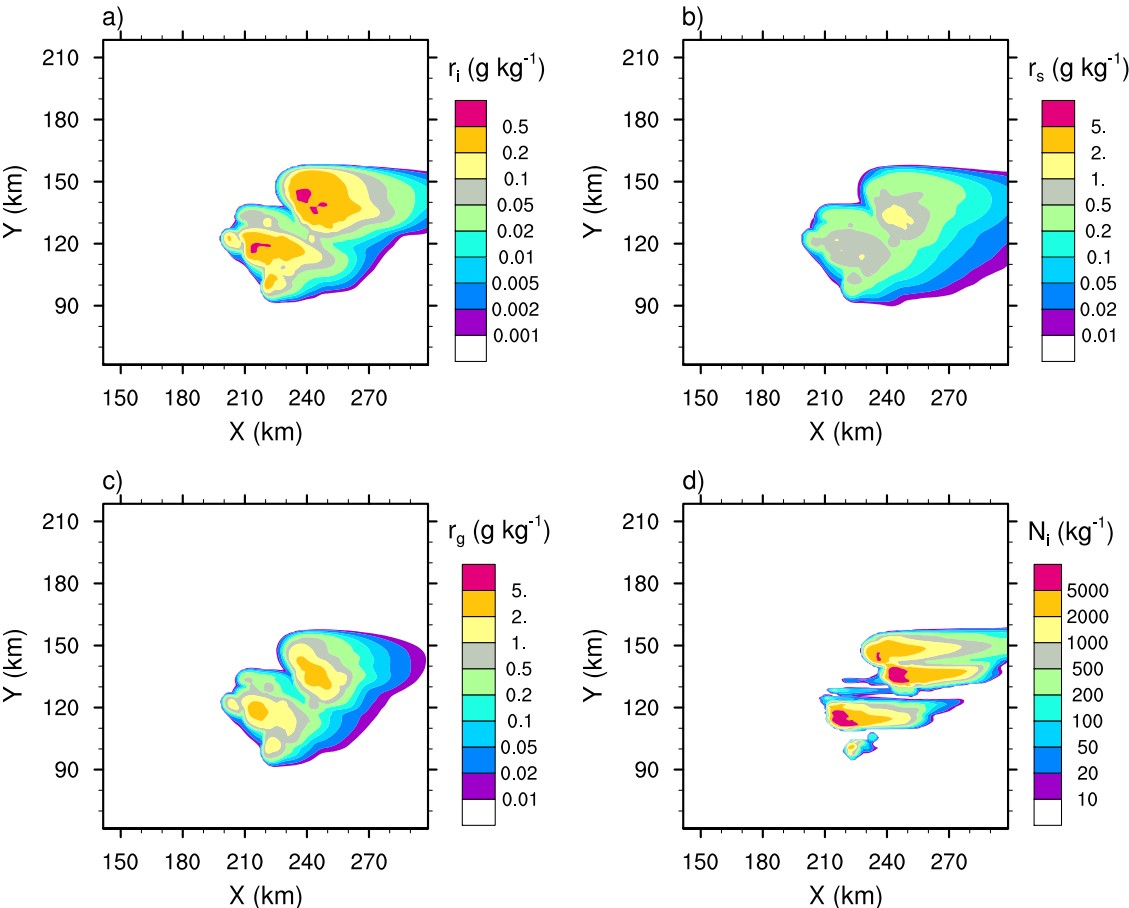

**Figure 7.** "RANDOM" case of the STERAO simulations showing the mixing ratios of a) the cloud ice ($r_i$), b) the snow-aggregates ($r_s$), and c) the graupel ($r_g$) at 12 km height. Plot d) refers to the number concentration of the cloud ice crystals ($N_i$) at 15 km height. The plots are for a fraction of the computational domain.

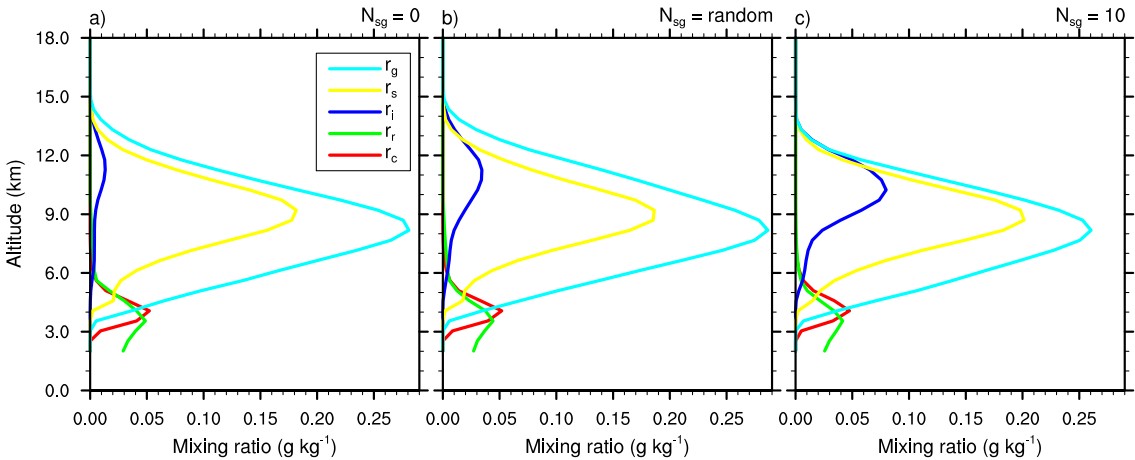

**Figure 8.** Mean profiles of condensate mixing ratios $r_c$, $r_r$, $r_i$, $r_s$ and $r_g$ ; in g kg$^{-1}$) of the STERAO simulations corresponding to a) the $\mathcal{N}_{sg}$=0.0 case, b) the "RANDOM" case and c) the case with $\mathcal{N}_{sg}$ = 10.0.

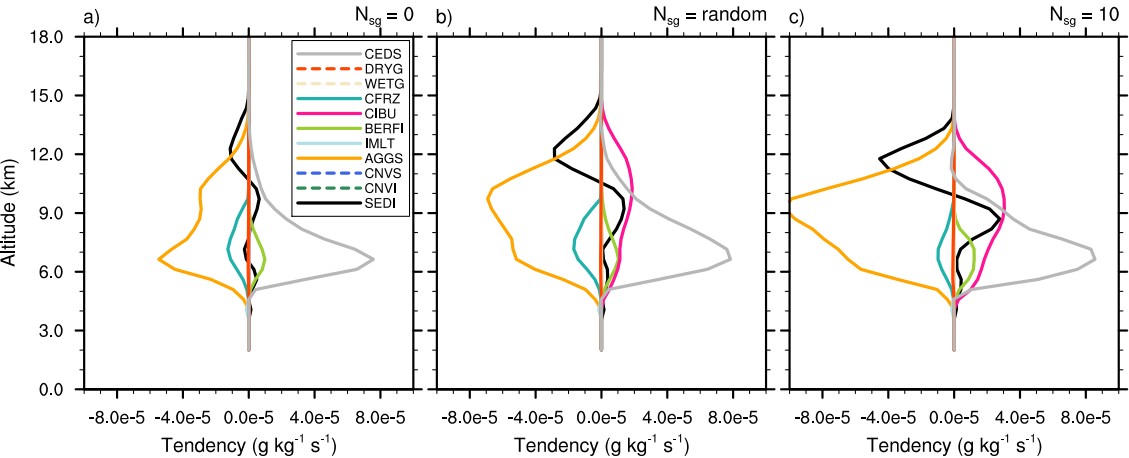

**Figure 9.** Mean microphysics profiles of cloud ice mixing ratio tendencies of the STERAO simulations corresponding to a) the $\mathcal{N}_{sg} = 0.0$ (no CIBU) case, b) the "RANDOM" case and c) the case with $\mathcal{N}_{sg} = 10.0$. The dashed lines are associated with processes having no significant impact on these budgets.

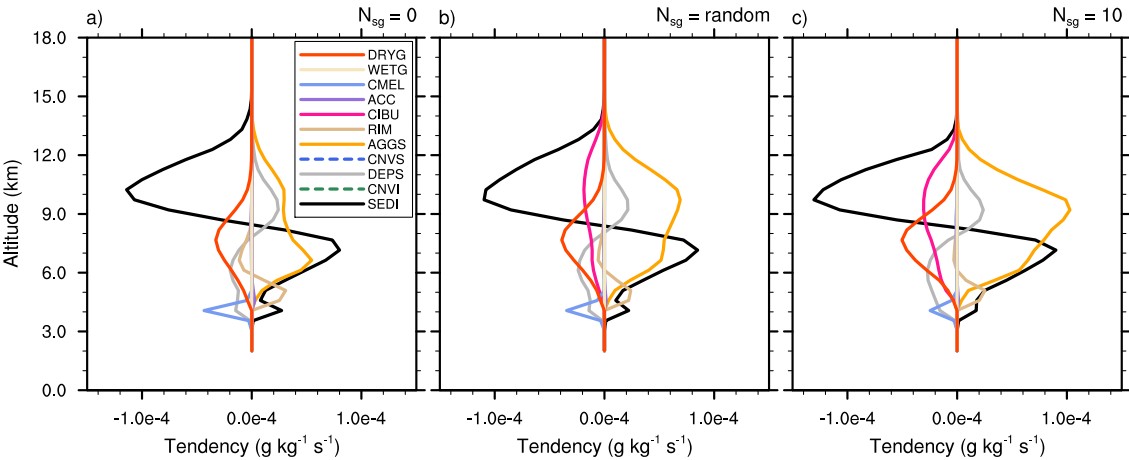

**Figure 10.** As in Fig. 9 but for snow-aggregates.

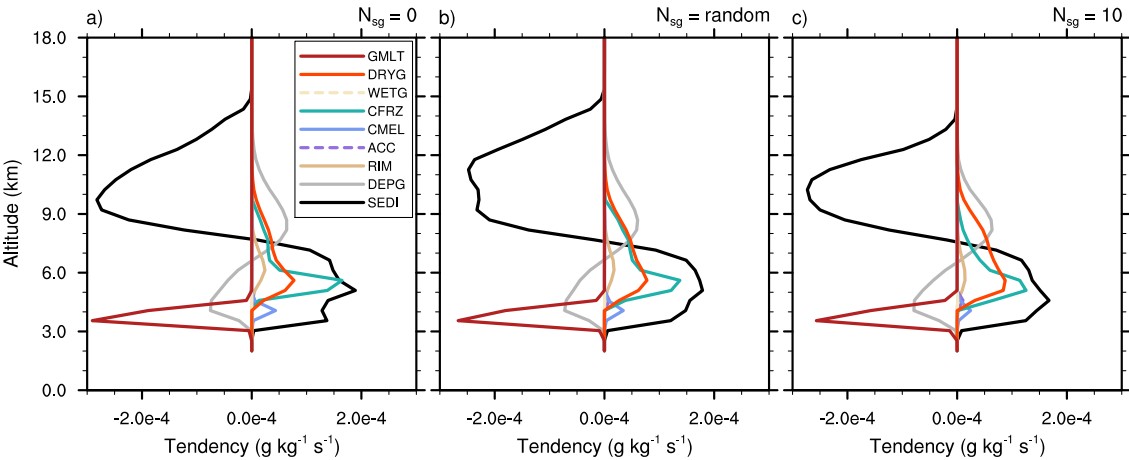

**Figure 11.** As in Fig. 9 but for graupel.

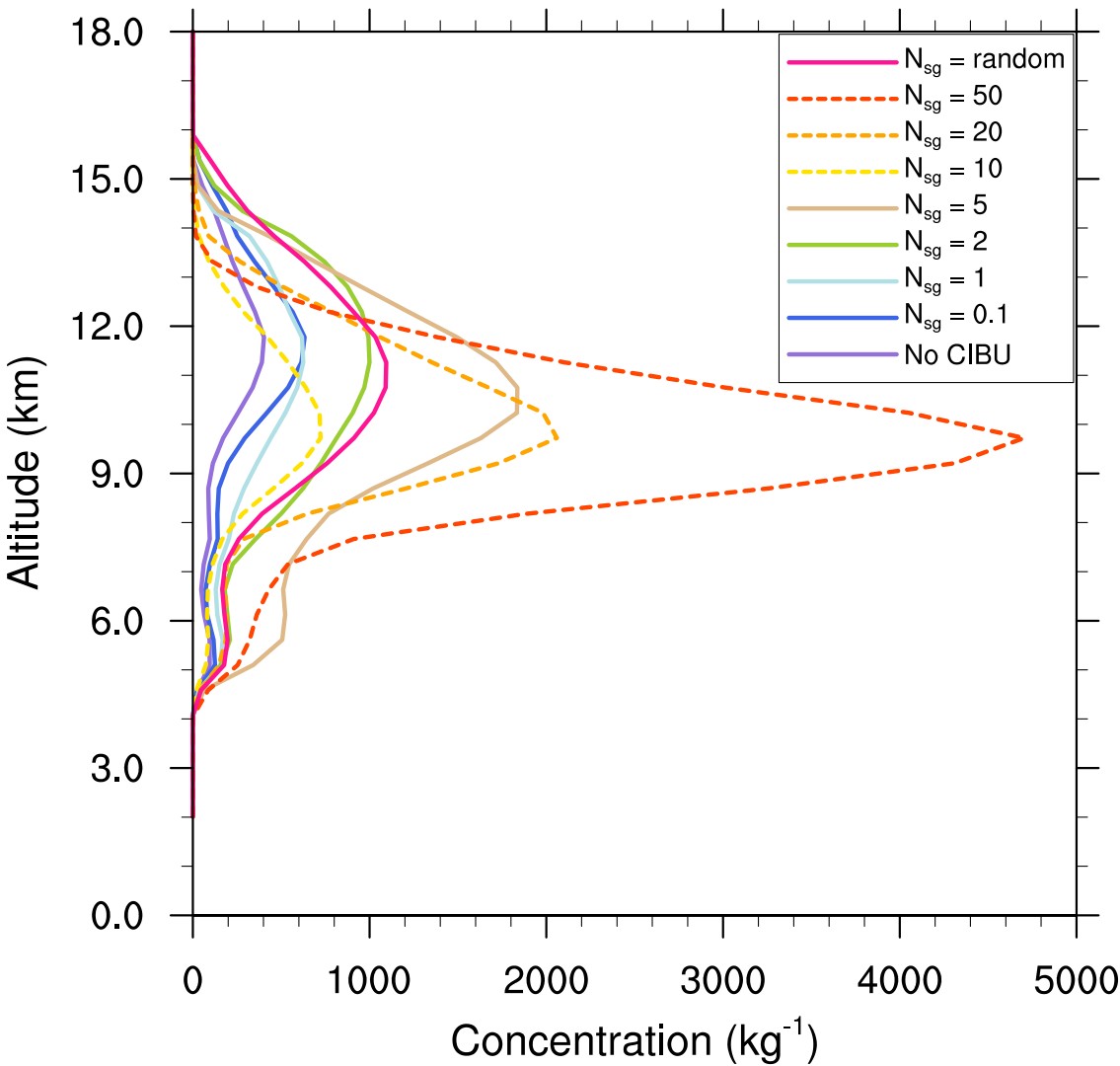

**Figure 12.** Mean profiles of the cloud ice crystal concentrations $N_i$ (g kg$^{-1}$) of the STERAO simulations corresponding to different values of $\mathcal{N}_{sg}$ (see the legend for details). The profiles drawn with a dashed line have been divided by 10 to fit into the plot.

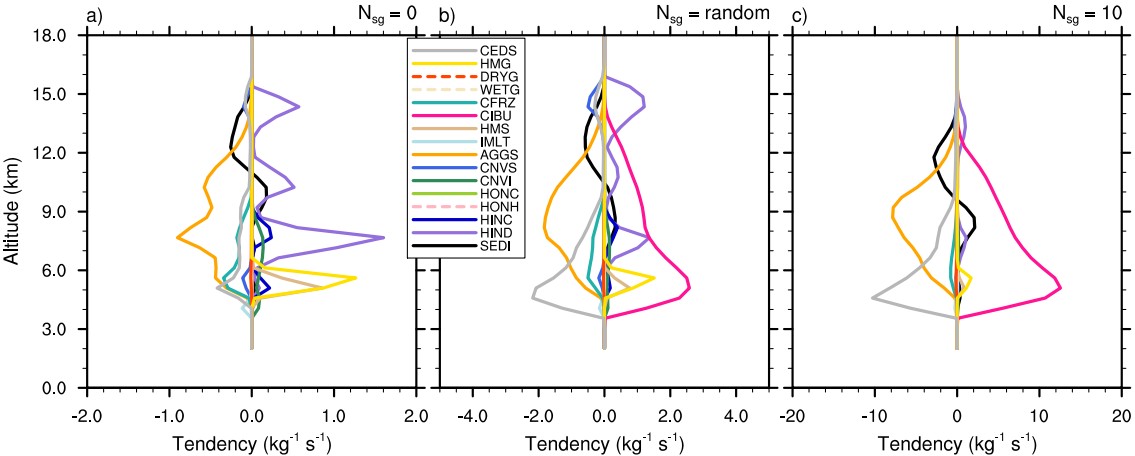

**Figure 13.** Mean microphysics profiles of the cloud ice crystal concentration tendencies of the STERAO simulations corresponding to a) the $\mathcal{N}_{sg} = 0.0$ (no CIBU) case, b) the "RANDOM" case and c) the case with $\mathcal{N}_{sg} = 10.0$ (Note that the horizontal scale increases from a) to c)). The dashed lines of the list box are associated with processes having no significant impact on these budgets.

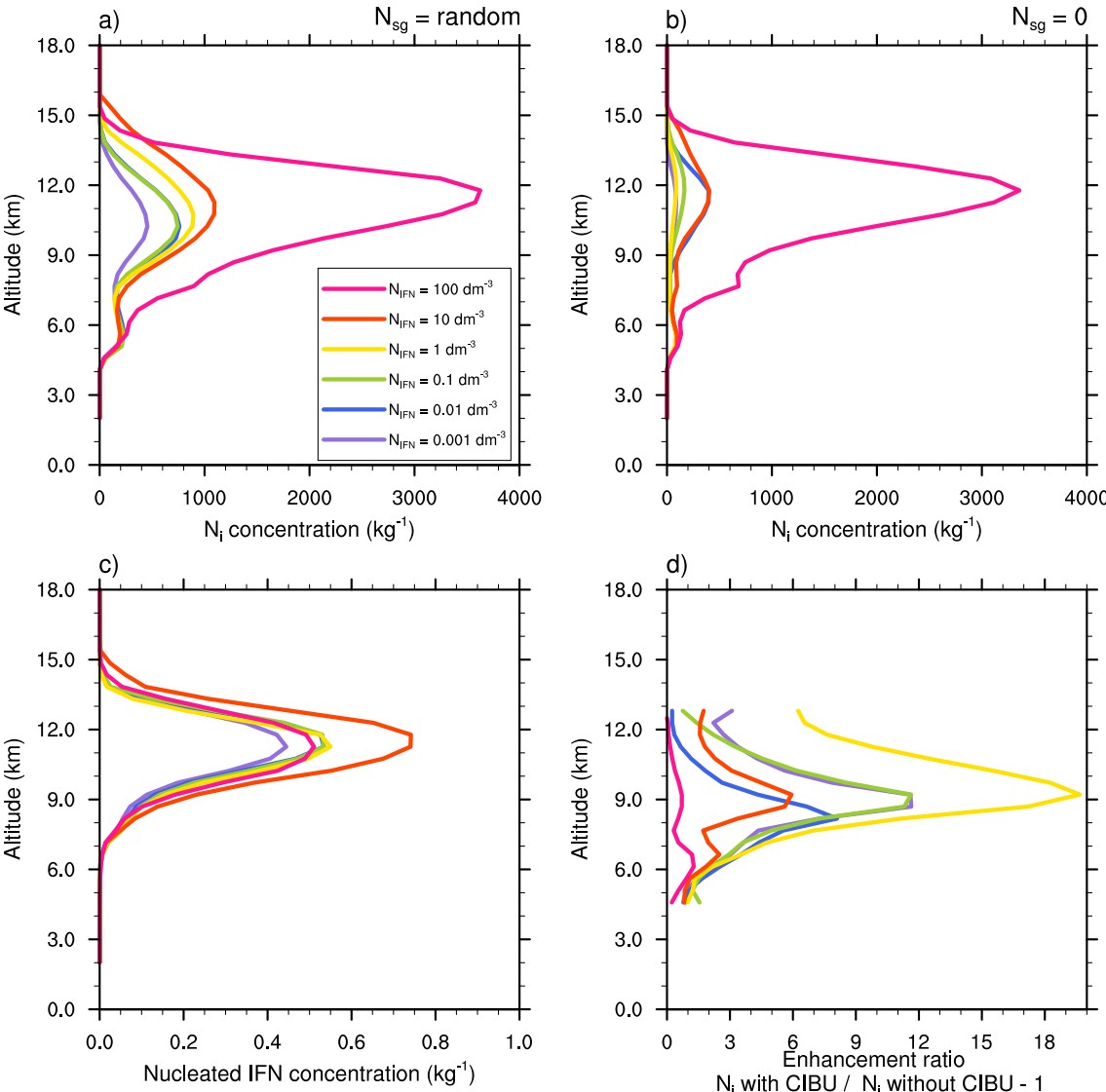

**Figure 14.** Mean profiles of cloud ice crystal concentration for initial IFN concentrations from $100\ dm^{-3}$ to $0.001\ dm^{-3}$ of the STERAO simulations corresponding to a) the CIBU simulation and "RANDOM" case and b) the non-CIBU simulation. The mean profiles of the nucleated IFN concentrations are plotted in c) after rescaling to fit the [0.0-1.0] range. The rough estimate of CIBU enhancement factor of $N_i$ is plotted in d) as a function of the initial IFN concentrations.

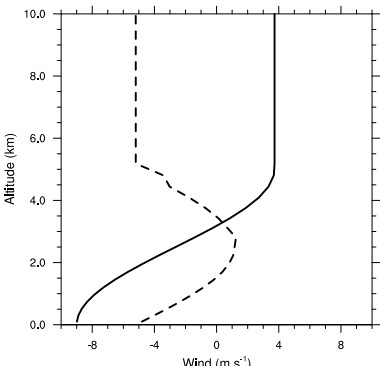

**Figure 15.** Vertical profile of the horizontal wind components of the WK84 simulations. The solid line with a constant shear $(2.5 \times 10^{-2}~\text{s}^{-1})$ refers to $U$, the $x$-component of the wind and the dashed line with a jet-like structure, refers to $V$, the $y$-component of the wind. $U$ and $V$ are constant above 5 km height.

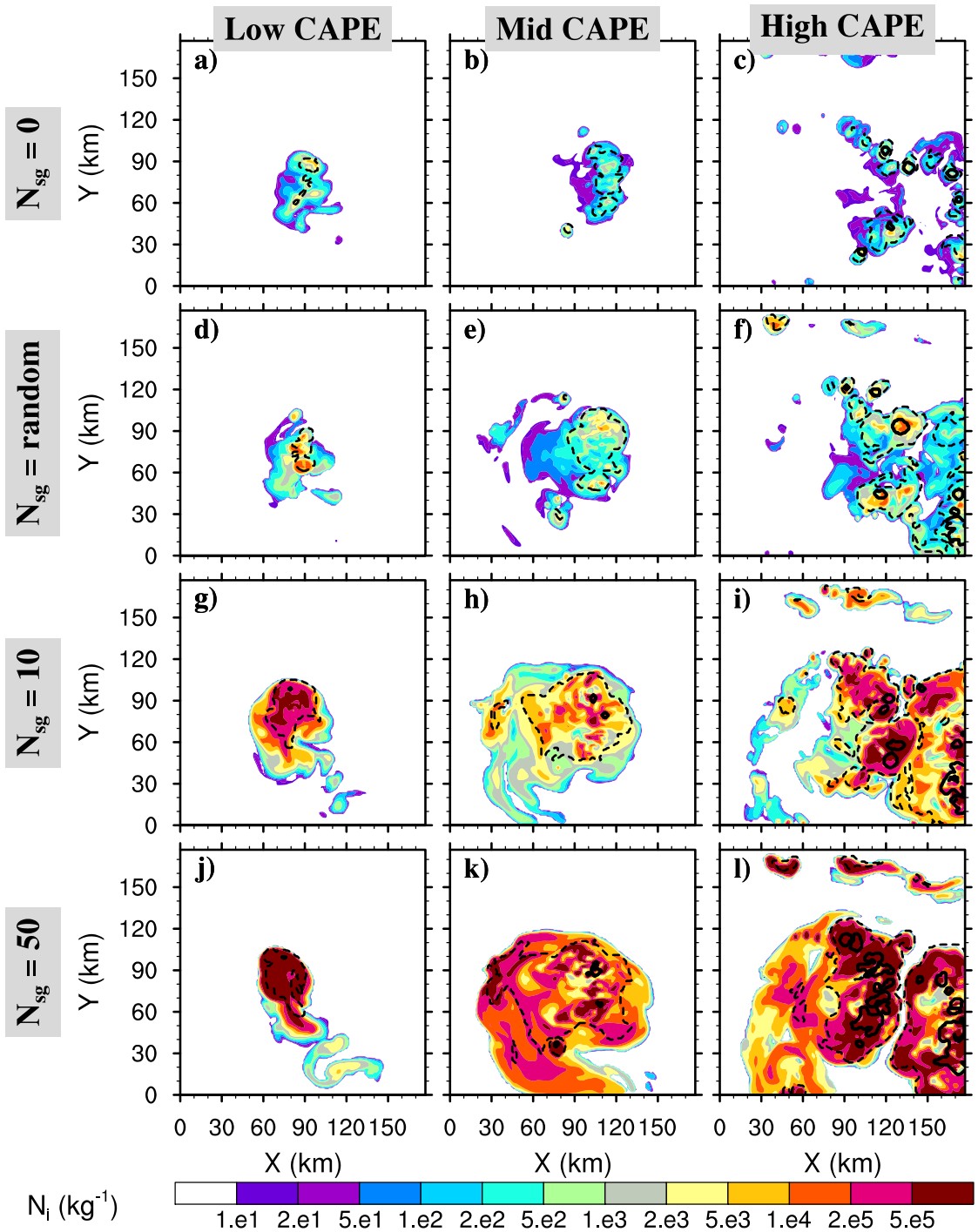

**Figure 16.** Small ice concentration $N_i$ average between 9.5 and 10.5 km height after 4 hours of the WK84 simulations, where a) to c) refer to no CIBU cases ($\mathcal{N}_{sg}$=0.0), d) to f) to cases with random CIBU (0.1<$\mathcal{N}_{sg}$<10) and g) to i) to cases with a high CIBU effect ($\mathcal{N}_{sg}$=10.0), and j) to l) to cases with an intense CIBU effect ($\mathcal{N}_{sg}$=50.0). The contours are the cloud top heights with dotted lines for 11 km and solid lines for 13 km.

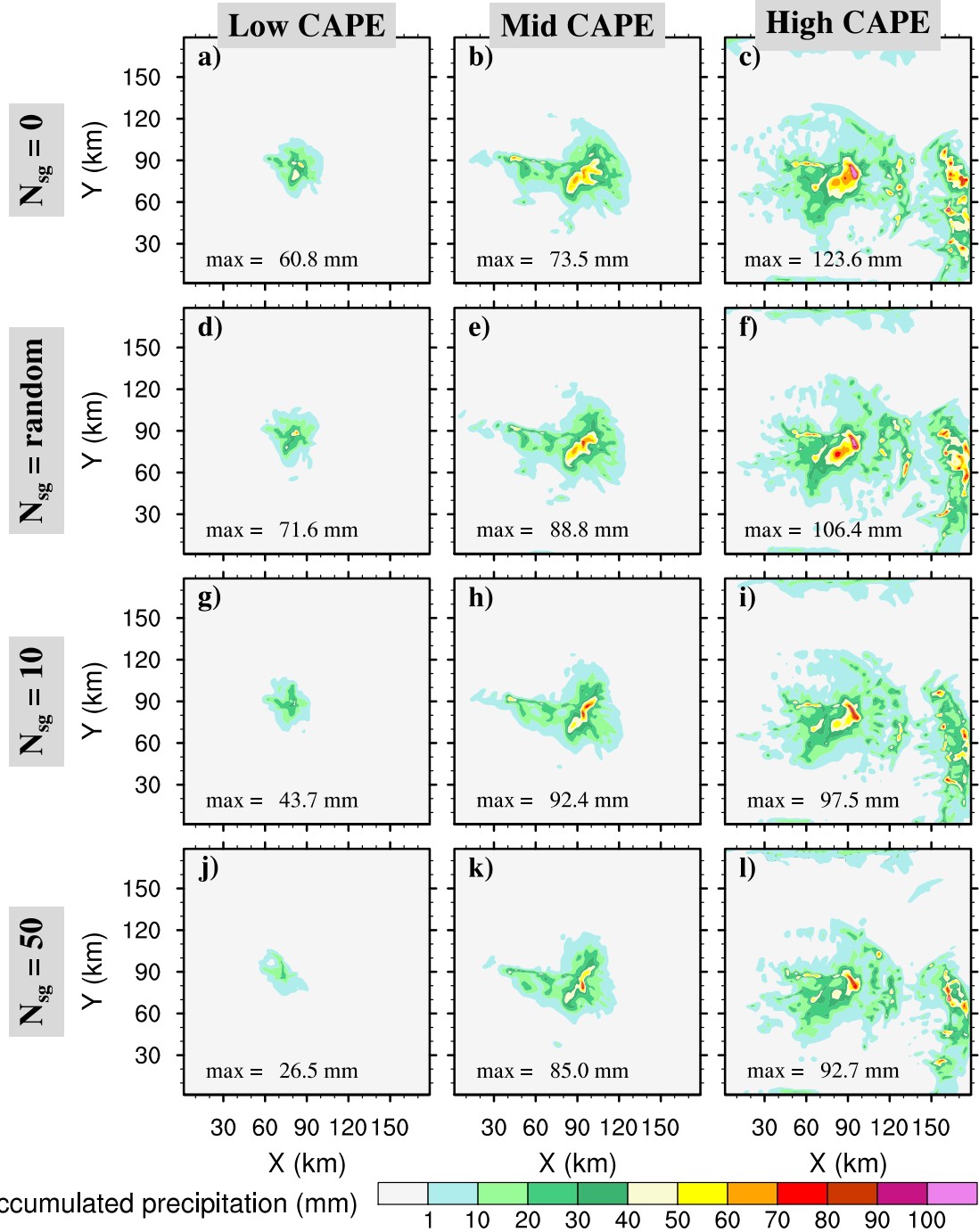

**Figure 17.** As in Fig. 16, but for the 4-h accumulated precipitation of the WK84 simulations. The peak value (max in mm) corresponds to the peak value of precipitation of the main convective clouds in the centre of the simulation domain.

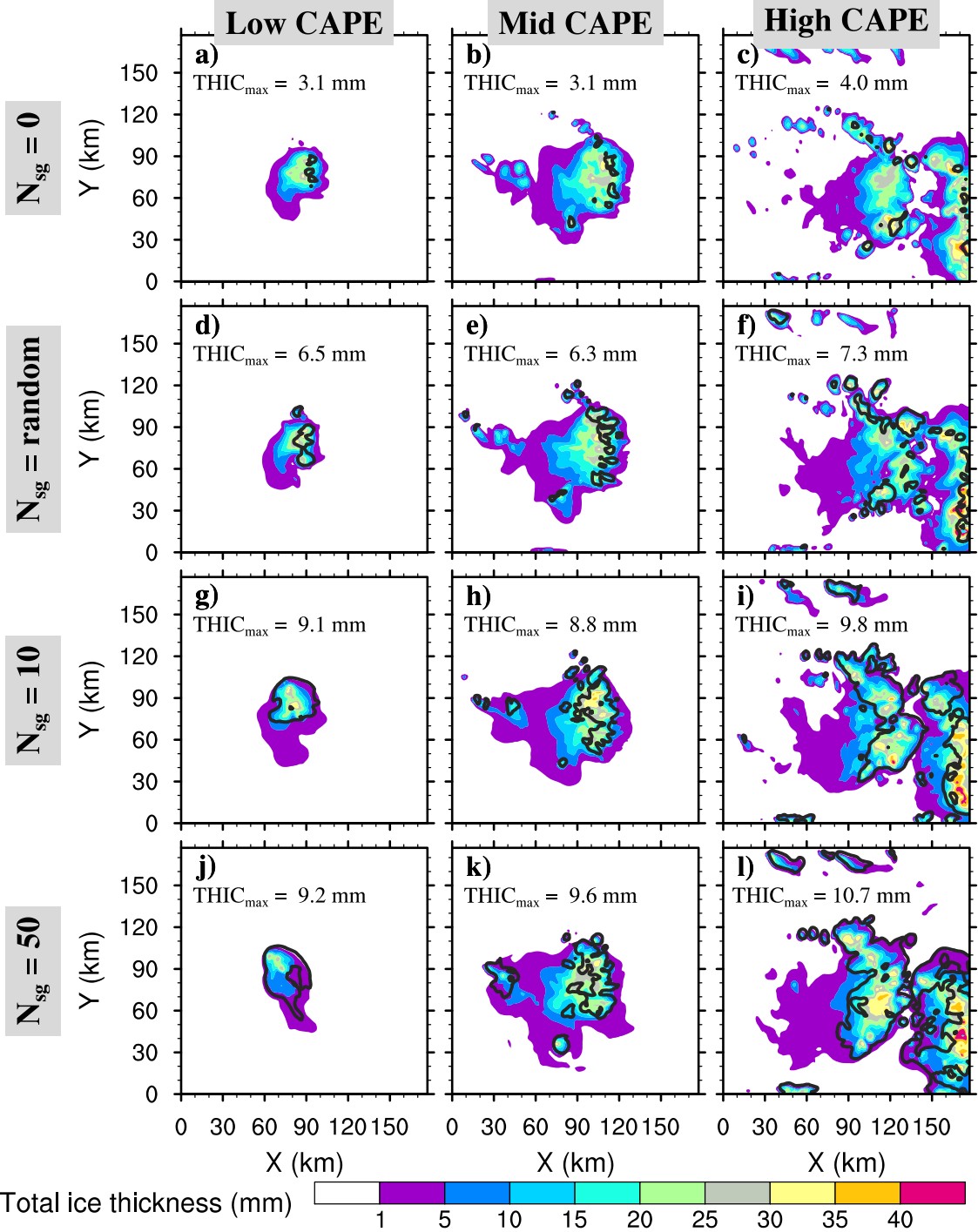

**Figure 18.** As in fig. 16, but for the total ice thickness in mm after 4 hours of the WK84 simulations. The contours are the small ice thickness component (THIC) taken at 1 mm. The peak value of THIC (THIC$_{max}$ is given in mm).