# Peer review of "A representation of the collisional ice break-up process in the two-moment microphysics scheme LIMA v1.0 of Meso-NH"

_Geoscientific Model Development, 2017_

## Referee Comment (RC1) · Anonymous Referee #1 · 12 Feb 2018

**Review of "A representation of the collisional ice break-up process in the two-moment microphysics scheme LIMA v1.0 of Meso-NH"**

**Major comments**

This work implements a parameterization of collisional ice breakup (CIBU) into the LIMA mesoscale model. The simulations are well-planned and some of the results are interesting, but the manuscript needs significant work. First, there is no discussion at all of the LIMA scheme into which the CIBU parameterization has been implemented. Is this a bin or bulk scheme; what are the different classes of ice hydrometer; and what are there threshold sizes? The scheme needs to be explained for the reader to understand the results. Then I understand that the location and synoptic environment of the STERAO case study are available in Skamarock et al. 2000, but these are crucial to this study and an overview should be given here as well.

Then the parameterization itself is not especially sophisticated. Even the limited laboratory measurements of collisional ice breakup suggest that there are strong temperature dependences of the fragment number. And bigger snow-aggregates break up into more pieces, no? In which case, there should be some kind of aggregate size dependence in the fragment number.

I am particularly concerned by some of the altitude / temperature dependence in the results. For example, ice mixing ratio from this CIBU process is peaking at 12 km, certainly corresponding to cirrus formation and quite cold temperatures. But these secondary ice processes have been discussed for mixed-phase conditions at much lower altitudes and warmer temperatures. The discrepancy in nucleating particles and ice crystal concentrations is at these lower altitudes, so what exactly is the CIBU parameterization intended to explain?

This leads in to my final point, which is that no comparisons to data are made. Are there precipitation or ICNC data from the STERAO case? If so, some attempt should be made to assess whether the new parameterization is yielding more or less accurate precipitation rates or crystal numbers. This will justify a number of currently unsubstantiated statements throughout that certain results are "plausible" or "excessive" or "satisfactory" (Lines 191 to 193, 200, 276).

**Specific comments**

Then a number of details need clarification:

Line 27 – "The CIBU process was not perceived as a particularly important feature in cloud physics." Here it is unclear to me in what context CIBU has been perceived as unimportant. In general, in cloud microphysics schemes? If so, please state that explicitly.

Lines 30 to 31 – "CIBU process is *very likely* to be active when cloud conditions are deemed favourable." I do not think that the two proceeding citations validate this statement. Some additional discussion, and perhaps other citations, is needed of what these favourable conditions are.

Lines 57 to 59 – This sentence could use rewording, for example "An empirical but realistic CIBU parameterization is implemented in the well-suited LIMA scheme and interacts with other microphysical processes (heterogeneous ice nucleation, H-M process, etc.) to determine the concentration of small ice crystals."

Line 61 – What does "erosion" mean here? Reduction of number?

Line 69 – "nucleation process yield" It would be clearer to say "scaled by the ice number concentration from nucleation".

Lines 73 to 74 – Sullivan et al. 2018 doi 10.5194/acp-18-1593-2018 would be another appropriate reference.

Line 81 – What does "covering" mean here? Including? Can you give an estimate of the average size of the large graupel particles? Or the lower threshold size for this categorization? This especially needed to assess the appropriateness of the assumption in line 94.

Lines 85 to 86 – Again it is unclear what this means: "particle sizes are taken to stay within a range of substantial occurrence of CIBU." Please make it more specific.

Line 92, Equation 2 – Please define Π.

Line 104 – Please cite the source from which you get your ice collisional efficiencies.

Line 106 – What is $D_{trough}$? It does not seem necessary to add a variable name.

Line 110 – Two parameters, i.e. both $D_{s,max}$ and $D_{g,min}$, cannot be dictated by a single equation.

Line 112 – "Least favourable situation" is unclear here. "Least favourable" for a large contribution from CIBU to ICNC? Why would you be considering this "at ground level" where temperatures will generally not permit ice formation in any case?

Lines 144 to 146, Equation 4 – My recommendation would be to move all of this to Appendix A. Otherwise, a large number of undefined variables appear all of sudden.

Lines 153 to 154 – What is the "local mean mass of the pristine ice crystals"? On what does this depend? What is "ice debris"?

Line 172 – What does "along the main diagonal" mean? The location of the 10 July 1996 thunderstorm needs to be included.

Line 176 – The acronym PPM needs to be expanded.

Line 182 – If the aerosol concentrations "have no importance for the simulations", perhaps Table 1 can be omitted.

Line 188 – This is a nice result, but it would be clearer to show difference fields in Figure 1b-d.

Figures 3, 4, and 5 – Again this is your call, but I think it would be easier to see the impact with difference fields of mixing ratio (taken from the base case).

Figure 7 – Here, I think you really need to show difference fields. Otherwise, you force the reader to flip back and forth with previous figures to make the comparison.

Section 3.1 – To me, it would make more sense to begin with the changes to ice metrics and microphysics because these should be directly impacted and to follow with precipitation because this link is indirect.

Lines 234 to 236 – You need to mention that the acronyms are given in Table 3 here.

Line 242 – 0.2 **x** 10$^{-3}$

Figure 9 – Why is nucleation - HINC, HIND, and HONC – not included in this Figure? These seem to be the tendencies one would most like to compare with CIBU.

Figures 9 to 11 – Are these domain-averaged? Or shown for a single grid cell?

Line 273 – $N_i$ ($N_{sg}$ = 0) The parentheses are important.

Around Line 277 – There needs to be discussion about why CIBU ice mixing ratio peaks at higher altitudes than does the CIBU ice number concentrations. Are the snow-aggregates at higher altitudes bigger? Otherwise, it is not clear to me what is going on here.

Line 305 – This behavior is not difficult to interpret. It results from the tradeoff between homogeneous and heterogeneous ice nucleation. Until there is quite a large IFN concentration, additional particles will suppress homogeneous nucleation and reduce ICNC.

Figure 14 – It is harder to interpret your results when you switch between $L^{-1}$ and $kg^{-1}$. In particular, I am confused by some enhancement values in panel d. For example the peak $N_i$ for $N_{IFN}$ = 1 $L^{-1}$ is 1000 $kg^{-1}$ which is more or less 1:1, no? Why does the enhancement in yellow go up to 18? What am I missing?

Line 328 – "shocks" is generally used for electrostatic phenomena. "Collisions" is better.

---

## Referee Comment (RC2) · Anonymous Referee #2 · 18 Mar 2018

Review of "A representation of the collisional ice break up process in the two moment microphysics scheme LIMA v1.0 of Meso-NH"

This paper describes a new implementation of a collisional ice break parameterization in a two moment microphysics scheme. This particular secondary ice formation mechanism is very poorly understood, and modelling studies are necessary to ascertain whether it can have an impact on mixed phase cloud microphysics. The subject of the paper is thus quite suitable for GMD. However, the analysis is too limited and the results are unclear. The language is very hard to follow, which makes the results harder to understand and review. The writers are strongly urged to make the best effort possible at improving the readability of the manuscript by revising the language.

The major shortcoming of the paper, which is recurrent throughout all of the analysis carried out, is the lack of conducting proper diagnostics to establish that the results are robust. There is very little description of the test case used, which is not acceptable given that the results are very specific to the details of the experimental setup. The writers are urged to dedicate a full section at describing the experimental setup so the reader can have an idea of how susceptible the simulation is to the microphysical changes incurred. My impression is that the simulation is very dynamically forced so one does not expect changes in the dynamics that would feed back into the microphysics which would make comparison of the microphysical fingerprints difficult. The writers must show that this is the case, or if it is not, then conduct the appropriate analysis on the dynamic-microphysical feedbacks.

In the spirit of the preceding critique, there is very little discussion on how the collisional breakup mechanism can alter microphysics-dynamics interactions. The precipitation results for example are presented as if changes in precipitation do not alter the dynamics of the storm. The authors do calculate the tendencies for the ice budget, but very little discussion is carried out. The tendencies of vapor depositional growth, riming, sedimentation etc. are all being altered but not enough detail is given as to how. Instead there is only a very brief overview (e.g. Sec. 3.3).

Unfortunately, the manuscript in its current form is not suitable for publication in GMD. Despite the uniqueness of the study and its importance, the manuscript fails at placing the collisional breakup mechanism in the context of a cloud resolving model. My major concerns are further detailed in the specific comments that follow.

Specific comments

Abstract, L16-19: This statement is contradictory to the preceding one. If it is concluded that the CIBU scheme needs better observational constrains, then why is it ready to be used in its current form to simulated REAL deep tropical clouds?

Introduction: A discussion, with the relevant references, is needed to motivate the collisional break up process. Specifically, the writers should cite cases in which excessive ice crystal numbers cannot be explained by the Hallet-Mossop mechanism. The authors should also refer to other possible secondary ice formation mechanisms like drop shattering. In its current form, the introduction does not motivate the need to carry out numerical experiments of the collisional break up process.

Sec. 2.1: Please justify the choice of a temperature independent $N_{sg}$ here. For example, Sullivan et al. (2017) use an $N_{sg}$ that is temperature dependent.

Sec. 2.2: A better description of the two moment scheme is needed. The equations can go into an appendix and more qualitative discussion of how the scheme defines the ice categories and how those would relate to the CIBA would be very beneficial here.

Sec. 3. L172-176: As mentioned above, many details of the test case are missing. "Several hours" is too general of a timeframe, please specify the actual time of simulation. There are no details of the boundary conditions. What is the spacing between the vertical levels?

Sec.3 L179-183: What about sensitivity to CCN? Please be clear here. Do you mean to say that you do not change the CCN concentration? As it is written, it sounds like you are saying that there is no sensitivity to the CCN concentration.

Sec. 3.1. L191-192: Why do the results suggest this empirically? Are the precipitation profiles being compared to some expectation which is satisfied in the specified range of $N_{sg}$? What is "unrealistic" about the simulation results for $N_{sg} > 10.0$?

Sec. 3.1. L199-202: This statement is unjustified. As emphasized in the preceding comment, realism of a specific $N_{sg}$ range has not been established, therefore the writers' conclusion on the choice of $N_0$ by Yano and Phillips (2011) being unrealistic is not justified. Also there aren't enough details about the cited study to make a meaningful comparison here.

Sec. 3.2. L206-208: This would not be counteracting effect. There is a reduction in the snow category as well as a reduction in the graupel category.

Sec. 3.2. L229-230: This statement needs justification. There should have been more analysis of why the precipitation changes in the different simulations in Sec. 3.1.

Sec. 3.3: This section is struggling to properly describe what is going as a result of the lack of explanation of the two moment microphysics scheme. Please define AGGS, CFRZ, and SEDI. These are physical processes, why not just use their names (e.g. deposition-sublimation)? Overall, its ok that this section is descriptive but it needs to be expanded to properly discuss the impact on each ice microphysical process.

Sec. 3.4. L274-276: An increase of 135% to 913% when $N_{sg}$ increases from 2 to 5 deserves a lot greater attention. The authors should conduct more analysis here to find out why this is the case. Saying its "exponential" is not enough. The result is also not tied to what is happening to the ice mass. There needs to be a more comprehensive analysis of what is happening to the ice budget as a whole.

Sec. 3.4. L276: Another reference to realism without justification.

Sec. 3.4. L280: Why is HIND more efficient here? Is it because the air becomes sub-saturated with respect to liquid water? Why about homogenous ice nucleation? What are HMG and HMS?

Sec. 3.4. L289: "Equilibrated" is not the right word here. I think you mean "balanced".

Sec. 3.4. L290: I gather here that there is that the authors have some understanding of why $N_i$ grows exponentially. This can address my earlier comment if the authors can clarify what they mean here. Why do all of these process rates grow in this fashion?

Sec. 3.5. L304-305: "Difficult to interpret" is not a satisfactory conclusion here. If the reader is going to be convinced of the very important argument being made in this section, a better effort needs to be made at understanding how the baseline simulations' Ni change with different ice nucleating particle concentrations. I'm also quite concerned that homogenous ice nucleation hasn't been addressed at all.

Sec. 3.5. L308-310: This statement is unclear. The authors say the nucleated IFN evolve in close proportion to initially available IFN but then the authors are also saying that the IFN do not depend on the IFN concentrations as expected?

Sec. 3.5. L314-322: The conclusions here are struggling to be properly understood and interpreted due to the fact that not enough information about LIMA or the baseline simulation have been give.

Sec. 4. L329-330: I can't agree with this statement. As I've already noted, no justification to this has been given.

Sec. 4. L359-360: A quantitative conclusion about the sensitivity of the simulations to different realizations of CIBU (due to changes in observationally constrained parameters) hasn't really been reached.

References

Sullivan, S. C., Hoose, C., and Nenes, A.: Investigating the contribution of secondary ice production to in-cloud ice crystal numbers, J. Geophys. Res., 122, doi:10.1002/2017JD026 546, 2017.

Yano, J.-I. and Phillips, V. T.: Ice–Ice Collisions: An Ice Multiplication Process in Atmospheric Clouds, J. Atmos. Sci., 68, 322–333, doi:10.1175/2010JAS3607.1, 2011.

---

## Author Comment (AC1) · 13 Apr 2018

**Responses to Referee #01**

« A representation of the collisional ice break-up process in the two-moment scheme LIMA v1.0 of Meso-NH » by Hoarau et al.

**Major comments**

This work implements a parameterization of collisional ice breakup (CIBU) into the LIMA mesoscale model. The simulations are well planned and some of the results are interesting, but the manuscript needs significant work. First, there is no discussion at all of the LIMA scheme into which the CIBU parameterization has been implemented. Is this a bin or bulk scheme; what are the different classes of ice hydrometer; and what are there threshold sizes? The scheme needs to be explained for the reader to understand the results. Then I understand that the location and synoptic environment of the STERAO case study are available in Skamarock et al. 2000, but these are crucial to this study and an overview should be given here as well.

Concerning the host microphysics scheme, it is true that we provide no extensive description of the LIMA scheme as we refer to Vié et al. (2016). We wished to describe our implementation of CIBU in a brief paper. However, it was clear enough that LIMA was a 2-moment bulk scheme. It was also our idea without a new lab. dataset, to include CIBU as simply as possible in a bulk scheme to see some consequences on the precipitation and the growth of the ice phase (the small crystals) depending on break-up efficiency i.e., the number of fragments produced per collision.

 $\rightarrow$  We added a few sentences in the last paragraph of the introduction to recall the processes to generate ice crystals in the bulk scheme LIMA.

In contrast to previous modelling studies (analytical solution in Yano and Phillips (2011, 2016) and the parcel model of Sullivan et al. (2017)), our purpose here was to suggest a way to include CIBU in a standard bulk scheme and so to encourage other similar microphysics scheme to account for this process in our state of knowledge of this phenomenon.

The choice of the STERAO case is purely illustrative as we could run any academic or real meteorological case.

Then the parameterization itself is not especially sophisticated. Even the limited laboratory measurements of collisional ice breakup suggest that there are strong temperature dependences of the fragment number. And bigger snowaggregates break up into more pieces, no? In which case, there should be some kind of aggregate size dependence in the fragment number.

Based on the few available data (Vardiman (1978), Takahashi et al. (1995)), it was hard to suggest a much complex parameterization. Precisely here we

worked on the critical parameter  $\alpha$ , the number of fragments per collision defined in Eq. 2, which multiplies the importance of CIBU. Then we found that limiting  $\alpha$  is necessary both to enhance the concentration of the small ice crystals and to alter not too much the precipitation at the ground. We don't consider any temperature effect, not mentioned in Vardiman (1978). Temperature plays a crucial role in ice nucleation, with assistance of ice forming nuclei (IFN), in the Hallett-Mossop process of droplet riming and possibly in the raindrop shattering by freezing (but, the parameterization of this process by Lawson et al. (2015) didn't include a temperature effect).

In the case of CIBU, it is clear at first sight that it is the possibility of collisions between dense graupel and fragile aggregates that governs this type of ice multiplication process. Without new laboratory experiments, one can only speculate on the true dependence of the temperature and the size of the aggregates. As we integrate the collision kernel over the size distributions (Eq. 3) of the graupel and the aggregates, we include somehow a size effect. Note also that intuitively the number of fragments should depend more on the radial location of the impact of the colliding graupel on the aggregates. This means that only a bulk approach, here the evaluation of a mean  $\alpha$  coefficient, is helpful in this situation as first we are more interested by the consequences to include or not a CIBU-like effect in a bulk microphysics scheme.

I am particularly concerned by some of the altitude / temperature dependence in the results. For example, ice mixing ratio from this CIBU process is peaking at 12 km, certainly corresponding to cirrus formation and quite cold temperatures. But these secondary ice processes have been discussed for mixed-phase conditions at much lower altitudes and warmer temperatures. The discrepancy in nucleating particles and ice crystal concentrations is at these lower altitudes, so what exactly is the CIBU parameterization intended to explain?

The enhancement of the small ice crystal mixing ratio (Fig 3) at the 12 km level is not very surprising because the upward transport in the STERAO convective cells is very efficient (the vertical velocity reaches 40 m/s see Barth et al., 2007). We feel that this is a good point when besides we notice no dramatic change in the aggregate and graupel mixing ratios (Figs 4-5). Of course the CIBU process needs the simultaneous presence of aggregates and graupel which are peaking close to 9 km height (Fig. 8). As CIBU is independent of the temperature in our case, we don't favour the ice multiplication through CIBU at very cold temperature. It is true however that it is a possible way to check the CIBU efficiency by examining the persistence of detrained cirrus clouds from convective areas.

We see no conflict between ice nucleation and CIBU in the glaciated regions of the convective cells. Our representation of the nucleation is adapted from Phillips's empirical scheme of 2008 with a careful budget of the IFN as we consider the available and the nucleated IFN of several origins (here a dust mode and a BC mode, see Vié et al., 2016). So ice nucleation is governed by the temperature and the abundance of IFN while, independently, CIBU is the result of the simultaneous presence of aggregates and graupel particles. It is true also that ice crystals coming from ice nucleation are transported too at higher levels to populate cold regions well above 10 km high. So CIBU is an alternative to ice nucleation to increase the small ice crystal concentrations when IFN are limited. There is no malice behind that.

This leads in to my final point, which is that no comparisons to data are made. Are there precipitation or ICNC data from the STERAO case? If so, some attempt should be made to assess whether the new parameterization is yielding more or less accurate precipitation rates or crystal numbers. This will justify a number of currently unsubstantiated statements throughout that certain results are "plausible" or "excessive" or "satisfactory" (Lines 191 to 193, 200, 276)

We acknowledge that no comparisons to data are made also because there is no case study yet showing unambiguously that CIBU was strongly operating. The few ICE-T cases reported by Lawson et al. (2015) concluded on the importance of raindrop shattering because of the presence of spicules on frozen drops seen on CPI images. Clearly, missing arms on aggregates are probably more difficult to detect in the same way. Only Hobbs and Farber (1972) reported evidence for CIBU with a formvar replicator. Our feeling is that ice multiplication does exist in clouds (Leroy et al., 2015; Ladino et al., 2017) but without CPI images it is difficult to assess that it is solely the result of collisional ice break-up or raindrop shattering by freezing. As a result, work is underway to include this last process to complete the panoply of extra ice crystal sources in the clouds simulated by the LIMA scheme in Meso-NH.

To conclude and also to account for remarks of the  $2^{nd}$  reviewer, we justify our parameterization of CIBU (for 2-moment bulk microphysics scheme) by the need to introduce new mechanisms to explain "anomalous" high ice water concentrations but under the constraint of minimizing perturbations to the production of precipitating hydrometeors. This is the starting point of our study to check the value of the critical parameter  $\alpha$ . We agree to remove most of the unsubstantiated statements in the revised version of the paper.

Additional references:

Barth, M. C., Kim, S.-W., Wang, C., Pickering, K. E., Ott, L. E., Stenchikov, G., Leriche, M., Cautenet, S., Pinty, J.-P., Barthe, Ch., Mari, C., Helsdon, J. H., Farley, R. D., Fridlind, A. M., Ackerman, A. S., Spiridonov, V., and Telenta, B. 2007: Cloud-scale model intercomparison of chemical constituent transport in deep convection, Atmos. Chem. Phys., 7, 4709-4731, https://doi.org/10.5194/acp-7-4709-2007.

Lawson, R.P., S. Woods, and H. Morrison, 2015: The microphysics of ice and precipitation development in tropical cumulus clouds. J. Atmos. Sci., 72, 2429–2445, https://doi.org/10.1175/JAS-D-14-0274.1

Leroy, D., Fontaine, E., Schwarzenboeck, A., Strapp, J. et al., "HAIC/HIWC Field Campaign - Specific Findings on PSD Microphysics in High IWC Regions from In Situ Measurements: Median Mass Diameters, Particle Size Distribution Characteristics and Ice Crystal Shapes," SAE Technical Paper 2015-01-2087, 2015, https://doi.org/10.4271/2015-01-2087.

Phillips V.T., P.J. DeMott and C. Andronache, 2008. An empirical parameterization of heterogeneous ice nucleation for multiple chemical species of aerosol. Journal of the Atmospheric Sciences 65(9): 2757–2783.

**Specific comments**

Then a number of details need clarification:

Line 27 – "The CIBU process was not perceived as a particularly important feature in cloud physics." Here it is unclear to me in what context CIBU has been perceived as unimportant. In general, in cloud microphysics schemes? If so, please state that explicitly.

We simply meant that the CIBU process is never taken into account explicitly in a microphysics scheme (bulk or bin) probably because its importance is overlooked in cloud physics. This observation justifies our present modelling study in GMD.

Correction: "...the CIBU process was overlooked in cloud physics. So to our knowledge a contribution of CIBU is never accounted for in the vast majority of the currently used microphysics schemes."

Lines 30 to 31 – "CIBU process is very likely to be active when cloud conditions are deemed favourable." I do not think that the two proceeding citations validate this statement. Some additional discussion, and perhaps other citations, is needed of what these favourable conditions are.

The referee is right, the sentence is awkward. So we suggest replacing it by: "... the CIBU process is very likely to be active in case of inhomogeneous cloud regions where ice crystals of different sizes and types are locally mixed." Then we introduce CIBU as the result of collisions between hydrometeors of different types.

Lines 57 to 59 – This sentence could use rewording, for example "An empirical but realistic CIBU parameterization is implemented in the well-suited LIMA scheme and interacts with other microphysical processes (heterogeneous ice nucleation, H-M process, etc.) to determine the concentration of small ice crystals."

We follow the suggestion to write:

"Here, the goal is rather to implement an empirical but realistic parameterization of CIBU in the well-suited LIMA scheme to cooperate with other microphysical processes (heterogeneous ice nucleation, droplet freezing, H-M process, etc.) to determine the concentration of small ice crystals."

*Line* 61 – What does "erosion" mean here? Reduction of number?

Here "erosion" means the mass loss of ice of the aggregates. This word is used sometimes in this context.

*Line* 69 – "nucleation process yield" It would be clearer to say "scaled by the ice number concentration from nucleation".

We agree, change made.

*Lines* 73 to 74 – *Sullivan et al.* 2018 doi 10.5194/acp-18-1593-2018 would be another appropriate reference.

We agree to add this new reference.

Line 81 – What does "covering" mean here? Including? Can you give an estimate of the average size of the large graupel particles? Or the lower threshold size for this categorization? This especially needed to assess the appropriateness of the assumption in line 94.

Initially we used "covering" because the "snow-lightly rimed" category of ice hydrometeor (aggregates) is wide enough to collect big pristine crystals (D>150  $\mu$ m) coming from water vapour grown pristine ice crystals and assemblages as a result of ice aggregation with light rime eventually. The sentence is rewritten as:

"... here we consider collisions involving two types of precipitating ice: small aggregates gathering pristine ice crystals larger than 150  $\mu$ m and large graupel particles."

In CIBU we integrate over the particle size distribution (PSD) of the graupel for sizes larger than  $D_{gmin}=2$  mm while we are doing the same for the PSD of the snow-aggregates but for 0.2 mm  $< D_s < 1$  mm, so we reasonably assume that  $D_g > D_s$  most of the time because the particle size is raised to power 2.

Lines 85 to 86 – Again it is unclear what this means: "particle sizes are taken to stay within a range of substantial occurrence of CIBU." Please make it more specific.

We meant that a way to impose an impact velocity of the graupel larger than 1 m/s is to integrate over the PSD but with an appropriate range of size. We felt that the choice of  $D_{smin}$ ,  $D_{smax}$  and  $D_{gmin}$  is a good compromise.

We modify the whole sentence in the following way:

"For the sake of simplicity and because the impact velocity of the graupel particles should be well above  $1 \text{ m s}^{-1}$  to remain in the break-up regime of the aggregates, the particle sizes are selected to enable a substantial occurrence of CIBU."

*Line* 92, *Equation* 2 - Please *define*  $\Pi$ .

Sorry for the typo, one should read  $\pi$  instead.

*Line* 104 – *Please cite the source from which you get your ice collisional efficiencies.*

We take the collision efficiency equal to one for the sake of simplicity and because we assume that for  $D_{smin} < D_s < D_{smax}$ , there is no lateral deflection of an aggregate (trajectory) when hit by a larger graupel particle. We offer no other explanation (see also Chapter 14 of Pruppacher and Klett, 1997). Note however that ice-ice collection processes are more dependent on the sticking efficiency which is temperature dependent in LIMA as revised in Ferrier et al. (1995), see also Phillips et al. (2015).

*Line* 106 – *What is Dtrough? It does not seem necessary to add a variable name.*

 $D_{trough}$  is the name given by Field (2000) in his Fig. 5 to separate the small pristine ice regime from the "modal" snow-aggregates.

*Line* 110 – *Two parameters, i.e. both Ds,max and Dg,min, cannot be dictated by a single equation.*

That's true but we had to make a choice because we are describing a bulk parameterization which is indeed sensitive to the contrasted properties of the aggregates and the graupel. Furthermore as it is clear that CIBU is not a threshold process (as it is the case for the autoconversion of the droplets for instance) there is an acceptable uncertainty for the choice of these parameters provided that the impact velocity is larger than  $1 \text{ m s}^{-1}$ .

A more elaborated choice for  $D_{smax}$  and  $D_{gmin}$  values could be based on the graupel-aggregate collision kinetic energy CKE per surface area of the aggregates (Phillips et al., 2015) but there is no clear indication of what reference to take to scale this parameter. In our case with  $D_{smax}=1$  mm and  $D_{gmin}=2$  mm, one gets CKE/ $(\pi/4D_{smin}^2)=0.038$  Kg s-2.

Line 112 – "Least favourable situation" is unclear here. "Least favourable" for a large contribution from CIBU to ICNC? Why would you be considering this "at ground level" where temperatures will generally not permit ice formation in any case?

The least favourable condition in this case is when an aggregate of size  $D_{smax}$  is hit by a small graupel of size  $D_{gmin}$  leading to the minimal impact velocity  $V_{sg}$ . We replace "the least favourable situation gives  $V_{sg}=1.26$  m s-1" by "one gets  $V_{sg}>1.26$  m s-1". We refer to the ground level because  $V_{sg}$  is always larger aloft.

Lines 144 to 146, Equation 4 – My recommendation would be to move all of this to Appendix A. Otherwise, a large number of undefined variables appear all of sudden.

We don't agree to move Eq. 4 (and Eq. 6) to the appendix A. The moments of the complete and incomplete gamma function are easy to identify. We suggest to modify line 142: "With the definitions of the moments  $M^{INC}_{x}(p,X)$  of the incomplete gamma law given in Appendix A, ..."

*Lines* 153 to 154 – What is the "local mean mass of the pristine ice crystals"? On what does this depend? What is "ice debris"?

We suggest to remove the word "local" and to replace "ice debris" by "ice fragments" for a better understanding.

*Line* 172 – *What does "along the main diagonal" mean? The location of the 10 July 1996 thunderstorm needs to be included.*

The convective bubbles are arranged according to Skamarock et al. (2000) in order to maintain the multicellular convection (that becomes supercellular at the end) as long as possible in the computation domain. The chosen STERAO case is a very classical one to test parameterizations in the context of continental high CAPE (Convective Available Potential Energy). The true location of the storm is of secondary importance. We modify the text in the following way: "The simulations were initialized with the sounding of northeastern Colorado given in ..." and "... along the main diagonal of the horizontal X, Y plan in the wind axis.".

*Line* 176 – *The acronym PPM needs to be expanded.*

PPM is Piecewise Parabolic Method a finite volume transport scheme. Done.

*Line* 182 – *If the aerosol concentrations "have no importance for the simulations", perhaps Table 1 can be omitted.*

Table 1 is necessary for those who wish to redo the simulation. We reword the sentence: " ..., the characteristics of the five aerosol modes are standard for the simulations shown here ..."

*Line* 188 – *This is a nice result, but it would be clearer to show difference fields in Figure 1b-d.*

We don't agree because differences of precipitation fields are more confusing to comment with positive and negative isocontours. We think that using the same color scale as it is in Fig. 1, is more demonstrative to underline the decrease of the precipitation when  $N_{sg}$  increases.

Figures 3, 4, and 5 – Again this is your call, but I think it would be easier to see the impact with difference fields of mixing ratio (taken from the base case).

We give the same response to the preceding question because we tried to plot difference fields but with less clarity.

Figure 7 – Here, I think you really need to show difference fields. Otherwise, you force the reader to flip back and forth with previous figures to make the comparison.

Well that's true but in a final publication, the figures are inserted in text body.

Section 3.1 - To me, it would make more sense to begin with the changes to ice metrics and microphysics because these should be directly impacted and to follow with precipitation because this link is indirect.

Unsurprisingly we knew that the critical parameter  $N_{sg}$  was monitoring the increase of the ice concentration  $N_i$  as much as wanted. So then a strong issue was to avoid too much perturbation to the simulated precipitation at the ground level when CIBU was activated. We add this constraint because microphysics schemes that don't include CIBU, are now running quantitative precipitation forecasts. For this reason we put in the foremost of Section 3.1 the limitation of *Nsg* in the revised version of the manuscript.

*Lines* 234 to 236 – You need to mention that the acronyms fare given in Table 3 *here.*

We agree and we add " ... (10 minute average again and the nomenclature of the processes provided in Table 3) ... "

*Line* 242 – 0.2 x 10-3

Corrected here and elsewhere.

Figure 9 – Why is nucleation - HINC, HIND, and HONC – not included in this Figure? These seem to be the tendencies one would most like to compare with CIBU.

Nucleation is an essential contributor to the ice concentration but not to the ice mixing ratio because the early ice crystals are very small until they grow by water vapour deposition.

Figures 9 to 11 – Are these domain-averaged? Or shown for a single grid cell?

We explain (Lines 224-226) how we select the cloudy columns to generate the profiles of Figs 9-11. We average over all the three main cells.

*Line*  $273 - Ni (N_{sg} = 0)$  *The parentheses are important.*

Sorry for the mislocation of the closing parenthesis. Corrected.

Around Line 277 – There needs to be discussion about why CIBU ice mixing ratio peaks at higher altitudes than does the CIBU ice number concentrations. Are the snow-aggregates at higher altitudes bigger? Otherwise, it is not clear to me what is going on here.

If we compare the profiles in Fig. 8 (mixing ratios) and in Fig. 12 (concentrations), we can see that the "cloud ice" peaks are located at the same height (12 km " $N_{sg}$ =0" case, 11 km " $N_{sg}$ =random" case and 10 km " $N_{sg}$ =10" case). So the question is more to understand why the profiles of the CIBU contribution seem out of phase when looking at the mixing ratio tendency  $\partial r_i/\partial t|_{CIBU}$  in Fig. 9 and at the number concentration tendency  $\partial N_i/\partial t|_{CIBU}$  in Fig. 13 (both are red coral curves). As written Line 153,  $\partial r_i/\partial t|_{CIBU}$  is taken as the minimum between the limiting value  $\partial r_i/\partial t$  given by Eq. 5 and  $\partial r_i/\partial t$  estimated as  $(r_i/N_i) \ge \partial N_i/\partial t|_{CIBU}$  where  $r_i$  and  $N_i$  are local characteristics of the cloud ice field (it is implicitly suggested here that the ice fragments produced by CIBU follow the local size distribution of the small ice crystals). So essentially because  $r_i$  is very low below 6 km, even where  $\partial N_i/\partial t|_{CIBU}$  is high,  $\partial r_i/\partial t|_{CIBU}$  remains low. Above 9 km, both  $r_i$  and  $N_i$  are reaching higher values so  $\partial r_i/\partial t|_{CIBU}$  is increasing.

Concerning the snow-aggregates, we don't consider the total concentration  $N_s$  as a state variable in LIMA. These particles are characterized by a single moment, the mixing ration  $r_s$ , while  $N_s$  is parameterized as  $C\lambda^x$  as recalled at Line 148.

Line 305 – This behaviour is not difficult to interpret. It results from the tradeoff between homogeneous and heterogeneous ice nucleation. Until there is quite a large IFN concentration, additional particles will suppress homogeneous nucleation and reduce ICNC.

We modify the end of the sentence to make it clearer (line 306): "... because of the non-monotonic trend of the  $N_i$  profiles with respect to  $N_{IFN}$ ." and we add a sentence "Here this is equivalent to computing an IFN nucleation efficiency" to introduce Fig. 14c at Line 308.

We don't see why homogeneous and heterogeneous ice nucleation should cooperate. They are independent processes. However the proportion of nucleated IFN doesn't change very much when  $N_{\rm IFN}$  spans over 6 decades.

Figure 14 - It is harder to interpret your results when you switch between L-1 and kg-1. In particular, I am confused by some enhancement values in panel d. For example the peak Ni for NIFN = 1 L-1 is 1000 kg-1 which is more or less 1:1, no? Why does the enhancement in yellow go up to 18? What am I missing?

We provide the simulation results in  $\# \text{ kg}^{-1}$  while the forcing conditions of the initial IFN concentrations are given in  $\# \text{ dm}^{-3}$  unit which is more intuitive. Sorry for the "L-1" unit in the title box of Fig 14a.

In panel d) the CIBU enhancement ratio shows a maximum for  $N_{IFN} = 1 \text{ dm}^{-3}$  (yellow curve) at an altitude of 9 km so in this case the simulation with CIBU is leading to an ice concentration  $N_i$  which is nearly 20 times larger than  $N_i$  of a similar simulation but run without CIBU. Of course the profiles in panel d) rely on the profiles shown in panel a), here giving  $N_i \approx 900 \text{ kg}^{-1}$ , and in panel b) with  $N_i$  less than 100 kg-1 but hard to see ! Note that this is only a snapshot and that ice crystals are also produced by Hallett-Mossop process and removed by aggregation and so on.

*Line* 328 – "shocks" is generally used for electrostatic phenomena. "Collisions" is better.

We replaced word "shocks" by "collisions" at the same place.

Additional references:

Ferrier, B. S., W.-K. Tao and J. Simpson, 1995: A double-moment multiple phase four-class bulk ice scheme. Part II: simulations of convective storms in

different large-scale environments and comparison with other bulk parameterizations. J. Atmos. Sci., 52, 1001-1033.

Phillips VTJ, Formenton M, Bansemer A, Kudzotsa I, Lienert B. 2015. A parameterization of sticking efficiency for collisions of snow and graupel with ice crystals: Theory and comparison with observations. J. Atmos. Sci. 72: 4885–4902.

Pruppacher, H. R., and J. D. Klett, 1997: Microphysics of Clouds and Precipitation, 2nd rev. edition, Kluwer Academic Publishers, 954 pp.

---

## Author Comment (AC2) · 13 Apr 2018

**Responses to Referee #02**

« A representation of the collisional ice break-up process in the two-moment
scheme LIMA v1.0 of Meso-NH » by Hoarau et al.

**Major comments**

*This paper describes a new implementation of a collisional ice break
parameterization in a two moment microphysics scheme. This particular
secondary ice formation mechanism is very poorly understood, and modelling
studies are necessary to ascertain whether it can have an impact on mixed phase
cloud microphysics. The subject of the paper is thus quite suitable for GMD.
However, the analysis is too limited and the results are unclear. The language is
very hard to follow, which makes the results harder to understand and review.
The writers are strongly urged to make the best effort possible at improving the
readability of the manuscript by revising the language.*

*The major shortcoming of the paper, which is recurrent throughout all of the
analysis carried out, is the lack of conducting proper diagnostics to establish
that the results are robust. There is very little description of the test case used,
which is not acceptable given that the results are very specific to the details of
the experimental setup. The writers are urged to dedicate a full section at
describing the experimental setup so the reader can have an idea of how
susceptible the simulation is to the microphysical changes incurred. My
impression is that the simulation is very dynamically forced so one does not
expect changes in the dynamics that would feed back into the microphysics
which would make comparison of the microphysical fingerprints difficult. The
writers must show that this is the case, or if it is not, then conduct the
appropriate analysis on the dynamic-microphysical feedbacks.*

The study relies on two main ideas. First, a collisional ice break-up (CIBU)
mechanism should occur in natural clouds where shocks between ice crystals are
very frequent (sometimes leading to cloud electrification) while this process is
not considered in microphysics schemes. Second, including CIBU should
increase the number concentration of the small ice crystals by two to four order
of magnitude (Ladino et al, 2017 for tropical clouds) but this effect should not
be too much detrimental to the genesis and to the amount of precipitation at
ground level. So the tuning of the CIBU scheme, here finding empirically
appropriate values to the $N_{sg}$ parameter as in other studies, is therefore necessary
but is a difficult task: taking $N_{sg} > 10$ starts to dramatically reduce the
precipitation but $N_{sg} < 0.1$ leads to no noticeable effects to the ice crystal
concentration in clouds.
The authors support the idea that after Vardiman (1978), it is urging to renew
the lab experiments to investigate CIBU and to fix up the limit of the number of

fragments after collisions. However, in the meanwhile, it is pertinent to examine the possible consequences of CIBU in current microphysics schemes and to propose a simple parameterization of CIBU based on state variables of a 2-moment microphysics scheme (Eq. 3 of the manuscript).

To summarize, we show that the CIBU process increases the number concentrations of the small ice crystals while an excessive CIBU forcing can dramatically reduce the precipitation. As we believe that cloud models have now greatly improved to reach the level of quantitative precipitation forecast, the inclusion of new secondary ice formation processes like CIBU and raindrop shattering is not harmless. To the authors knowledge, all the recent studies concerning CIBU were done out of the context of three dimensional cloud simulation so ignoring side effects of CIBU, as the goal was to show an expected (explosive) increase of the number concentration of the small ice crystal concentration.

In the introduction and in the revised text, we insist more on the aspect that adding CIBU (and the raindrop shattering in a future work), must be carefully conducted to not alter a fundamental outcome of microphysics scheme that is the production of precipitation. However, we feel that the non-linearity of the microphysics schemes is such that the number concentration of the small ice crystals can be considerably increased through unsuspected cloud physics processes without significant changes in the production rate of the hydrometeors. This is what this manuscript would like to demonstrate.

The supporting case study STERAO is a standard case of isolated deep continental convection over the US Great Plains that lasted several hours. As recommended we provide more information about the model set up for STERAO. However we don't understand the remarks concerning the microphysics-dynamics feedbacks. The microphysics-dynamics interactions do exist naturally in the model, with and without CIBU. The results show that even "very dynamically forced", CIBU modifies the precipitation at the ground level if we compare with a reference simulation where CIBU is ignored. This does not mean that CIBU should act strongly and directly on the dynamics. On the contrary, our results show that CIBU is moderately but systematically reducing the production efficiency of the hydrometeors, snow-aggregate and graupel particles on the rebound, so that all simulations show qualitatively a similar evolution of the storm.

*In the spirit of the preceding critique, there is very little discussion on how the collisional breakup mechanism can alter microphysics-dynamics interactions. The precipitation results for example are presented as if changes in precipitation do not alter the dynamics of the storm. The authors do calculate the tendencies for the ice budget, but very little discussion is carried out. The tendencies of vapor depositional growth, riming, sedimentation etc. are all*

*being altered but not enough detail is given as to how. Instead there is only a very brief overview (e.g. Sec. 3.3).*

We understand the critique but our goal was also to write a short paper. We don't discuss on the alteration of the storm dynamics but we show that increasing CIBU efficiency i.e., taking $N_{sg}$>10 is detrimental to the production of the precipitation. This is basically a major result of the STERAO case simulation of strong convection. We admit that we could have discussed this point more in depth. However it would be very strange that a process like CIBU which is ignored in most of the microphysics scheme, would change so much the dynamics of a storm when activated. We agree to extend the discussion around the ice budgets even if our primary goal was to shorten the manuscript.

*Unfortunately, the manuscript in its current form is not suitable for publication in GMD. Despite the uniqueness of the study and its importance, the manuscript fails at placing the collisional breakup mechanism in the context of a cloud resolving model. My major concerns are further detailed in the specific comments that follow.*

We hope that the amended version of the manuscript is more convincing. Our purpose was to draw attention to the overall integrity of a microphysics scheme. This means that adding "beneficial" processes, like collisional breakup or raindrop shattering, to increase the concentration of the small ice crystals, must be examined in the light of three dimensional simulations to evaluate all the consequences on the evolution of a precipitating system.

**Specific comments**

*Abstract, L16-19: This statement is contradictory to the preceding one. If it is concluded that the CIBU scheme needs better observational constrains, then why is it ready to be used in its current form to simulated REAL deep tropical clouds?*

We admit that the end of the abstract is awkward, so it has been revised.
We are convinced that CIBU does occur in clouds and that the strength of the process is high enough to be responsible for a secondary production of ice besides Hallett-Mossop mechanism and the rain drop shattering. But indeed, it is clear that very few lab data exist to elaborate a solid parameterization of CIBU (temperature dependence?). What is suggested here is to examine the consequences of a parameterization based only on a simplified form of the collection kernel between snow-aggregates and graupel particles times $N_{sg}$. So we agree without difficulty that our representation of CIBU could evolve with more data from lab experiments. In the meantime we urge the cloud physics

community to include a very simple representation of CIBU like the one proposed here in the microphysics schemes.

Consequently we add L12
"… an upper bound of the CIBU effect by examining the rainfall rates"
We modified the last sentence of the abstract to remove any ambiguity.
"However the proposed parameterization which is easy to implement in any two-moment microphysics schemes, could be used in this primary form to simulate deep tropical cloud systems where anomalously high concentrations of small ice crystals are preferentially suspected to occur."

*Introduction: A discussion, with the relevant references, is needed to motivate the collisional break up process. Specifically, the writers should cite cases in which excessive ice crystal numbers cannot be explained by the Hallet-Mossop mechanism. The authors should also refer to other possible secondary ice formation mechanisms like drop shattering. In its current form, the introduction does not motivate the need to carry out numerical experiments of the collisional break up process.*

We agree but the focus here is on the representation of the collisional ice breakup as a candidate for secondary ice production in clouds. The whole story of airborne observations of high ice crystal concentrations was redrawn in Field et al. (2017). We have no contribution here on observational evidence of collisional ice breakup so we would like to be brief on this kind of discussion. We have already a footnote (number 2) that refers to Table 1 of Field et al. (2017) where the secondary ice production mechanisms are listed with references.
However we added a new sentence at the end of the first paragraph (L39)
"… are predicted. At first, our wish to introduce CIBU in a microphysics scheme is essentially motivated by the detection of unexplained high ice water contents that sometimes largely exceed the concentration of ice nucleating particles (Leroy et al., 2015; Field et al., 2017; Ladino et al., 2017)

*Sec. 2.1: Please justify the choice of a temperature independent Nsg here. For example, Sullivan et al. (2017) use an Nsg that is temperature dependent.*

Sullivan et al. (2017) who didn't justify their choice either, introduced a temperature enhancement factor based on Fig. 4 of Takahashi et al. (1995). Compared to Vardiman (1977), we feel that Takahashi's experimental setup is not adapted for CIBU because the splinters are produced by cm size ice spheres to simulate graupel particles rubbing against each other (and not breaking up as expected for fragile aggregates). Furthermore we feel intuitively that the production of up to 400 ice fragments per collision at ~254 K is exaggerated.

We concluded that a limit of 10 fragments per collision is enough to increase by one magnitude the ice number concentration of the STERAO (continental) case. Yano and Phillips (2011) retained a production of 50 fragments per collision with no temperature dependence.

*Sec. 2.2: A better description of the two moment scheme is needed. The equations can go into an appendix and more qualitative discussion of how the scheme defines the ice categories and how those would relate to the CIBA would be very beneficial here.*

We would like to keep Sec. 2.2 as it is, but becoming Sec. 2.3 with title "**2.3 Representation of CIBU in the LIMA scheme**". We suggest reorienting the content of Sec. 2.2

"2.2 Characteristics of the LIMA microphysics scheme

The microphysics LIMA scheme (Vié et al., 2016) includes a representation of the aerosols as a mixture of Cloud Condensation Nuclei (CCN) and Ice Freezing Nuclei (IFN) with an accurate budget equation (transport, activation or nucleation, scavenging by rain) for each aerosol type. The CCN are selectively activated to produce the cloud droplets which grow by condensation and coalescence to produce the rain drops (Cohard and Pinty, 2000). The ice phase is more complex as we consider the nucleation by deposition on the IFN and the nucleation by immersion (glaciation of tagged droplets formed on partially soluble CCN). The homogeneous freezing of the droplets is possible when the temperature drops below -35° C. The Hallett-Mossop mechanism generates ice crystals during the riming of the graupel and the snow-aggregates. The H-M efficiency depends sharply on the temperature and on the size distribution of the droplets (Beheng, 1987). The initiation of the snow-aggregates category is the result of the depositional growth of large pristine crystals beyond a critical size (Harrington et al., 1995). Aggregation and riming are computed explicitly. Heavily rimed particles (graupel) can experience a dry or wet growth mode. The freezing of the raindrops by contact with the small ice crystals is leading to the frozen drops merged with the graupel category. The melting of the snow-aggregates leads to graupel and shedded raindrops while the graupel particles directly melt into rain. The sedimentation of all particle types is considered. The snow-aggregates and graupel particles are characterized by their mixing ratios only.

The LIMA scheme assumes a strict saturation of the water vapour over the cloud droplets while the small ice crystals are subject to super or undersaturated conditions (no instantaneous equilibrium)."

*Sec. 3. L172-176: As mentioned above, many details of the test case are missing. "Several hours" is too general of a timeframe, please specify the actual time of*

*simulation. There are no details of the boundary conditions. What is the spacing between the vertical levels?*

We added the information in the revised text.

"The test case is illustrated by idealized numerical simulations of the 10 July 1996 thunderstorm in the Stratospheric-Tropospheric Experiment: Radiation, Aerosols, and Ozone (STERAO) experiment (Dye et al., 2000). This case is characterized by a multicellular storm that becomes supercellular after 2 hours. The simulations were initialized with the sounding given in Skamarock et al. (2000) and convection was triggered by three 3K-buoyant bubbles aligned along the main diagonal of the X,Y plan in the wind axis. Meso-NH was run for 5 hours over a domain of 320 x 320 gridpoints with 1 km-horizontal grid spacing. There were 50 unevenly spaced vertical levels up to 23 km height. With the exception of the wind component, all the fields including microphysics, were transported by an accurate, conservative, positive-definite PPM scheme (Colella and Woodward, 1984). There are no surface fluxes but the 3D turbulence scheme of MesoNH is activated. Open lateral boundary conditions are imposed. The upper level damping layer of the upward moving gravity waves starts above 12500 m."

*Sec.3 L179-183: What about sensitivity to CCN? Please be clear here. Do you mean to say that you do not change the CCN concentration? As it is written, it sounds like you are saying that there is no sensitivity to the CCN concentration.*

We are running the simulations with the same CCN set up. Of course the LIMA microphysics scheme is sensitive to different CCN characteristics (CCN size distribution or mode and chemistry) and the scheme is able to deal with an external mixture of several CCN modes. Here CCN activation is not the purpose of the study but we need to give CCN characteristics to run the LIMA scheme. We just mention that the focus is not on the sensitivity to the CCN. The same for the IFN except for the last study in Sec 3.5 where we are seeking for a minimal IFN concentration to enable the CIBU process.
We modify the last sentence of the paragraph:
"… chemical composition of the CCN and of the IFN, the characteristics of the five aerosol modes are standard for the simulations shown here except for …"

*Sec. 3.1. L191-192: Why do the results suggest this empirically? Are the precipitation profiles being compared to some expectation which is satisfied in the specified range of Nsg? What is "unrealistic" about the simulation results for Nsg > 10.0?*

The numerical simulations can only provide an indication of the plausible range of values for $N_{sg}$ and so the interpretation of model results in terms of threshold

is by essence "empirical". We clearly claim that $N_{sg}$ must be larger that 0.1 to perturb a standard simulation run without CIBU while taking $N_{sg}>10$ is leading to an excessive perturbation of the precipitation field. A simulation performed with $N_{sg}=50$ reduced the accumulated precipitation by a factor 2.

Our strong argument is that it is not acceptable that the CIBU process which is ignored in the great majority of microphysics schemes, should take so much importance to accurately predict the precipitation. The central question of the study is therefore to find a compromise for $N_{sg}$ in order to increase the cloud ice concentration but with the less possible disturbance to the amount of precipitation at ground level.

We revised the text:

"A value lower than 0.1 leads to a negligible effect of CIBU in the simulation, while taking $N_{sg}>10.0$ has an excessive impact on the storm precipitation (the $N_{sg}=50$ case is not shown)."

and

L199: "… considering the strong adverse effect …"

L200: "… satisfactory approach. Admittedly, the limit $N_{sg} \sim 10.0$ is more an order of magnitude but our conclusion is to recommend …"

*Sec. 3.1. L199-202: This statement is unjustified. As emphasized in the preceding comment, realism of a specific Nsg range has not been established, therefore the writers' conclusion on the choice of N0 by Yano and Phillips (2011) being unrealistic is not justified. Also there aren't enough details about the cited study to make a meaningful comparison here.*

The paper of Yano and Phillips (2011) was trying to demonstrate that CIBU could lead to an explosive ice multiplication regime most of the time (see the discussion in their section 3 about the normalized ice multiplication efficiency). Clearly speaking the concern there was to reproduce very high concentrations of ice in idealized simulations and so ignoring any possible side effects such as perturbations brought to the production of ice hydrometeors. In addition, looking at Fig. 6 of Vardiman (1978) results, one can see that the plausible values of the fragment numbers are well below 50. Only Takahashi et al (1995) could detect hundreds of splinters in their lab experiment but we believe that the experimental set up was not truly representative of ice crystal collisions in cloud conditions (rubbing of cm size ice spheres against mechanical fragmentation of aggregates and dendrites by shocks as in Vardiman (1978)). We admit however that an upper boundary of $N_{sg}=10$ is more an order of magnitude than a true threshold so we revise our wording in the text (see above).

The arguments to randomize $N_{sg}$ were given in L128-131. Besides, the idea was also to check if rare events with $N_{sg} \gg 1$ could be sufficient to enhance the concentration of the small ice crystals but without adverse effect that is a significant decrease of the accumulated rain at ground level.

*Sec. 3.2. L206-208: This would not be counteracting effect. There is a reduction in the snow category as well as a reduction in the graupel category.*

This is true. Sorry for the coarse mistake. The sentence is "However a further effect is …"

*Sec. 3.2. L229-230: This statement needs justification. There should have been more analysis of why the precipitation changes in the different simulations in Sec. 3.1.*

In deep convective storms like the STERAO case, rain comes from the melting of the graupel particles and of the snow-aggregates, but with less importance. We showed in Fig. 8 that the extent of the mean profiles of the $r_g$ and $r_s$ mixing ratios is reduced when $N_{sg}$ is increased. Consequently the same occurs for the $r_r$ mean profile peaking at 0.05 gkg$^{-1}$ for $N_{sg}$=0 compared to 0.04 gkg$^{-1}$ for $N_{sg}$=10. We admit that the difference is not easy to detect but anyway less graupel implies less rain below the freezing level.
Modifications in the text:
"This change is accompanied by a reduction of $r_s$ (more visible between cases b) and c)) and by a reduction of $r_g$ which clearly stands out at z=8,000 m. The final result is a decrease of the rain mixing ratio $r_r$, because rain is mostly fed by the melting of the graupel particles."

*Sec. 3.3: This section is struggling to properly describe what is going as a result of the lack of explanation of the two moment microphysics scheme. Please define AGGS, CFRZ, and SEDI. These are physical processes, why not just use their names (e.g. deposition-sublimation)?*
*Overall, its ok that this section is descriptive but it needs to be expanded to properly discuss the impact on each ice microphysical process.*

We apologize to keep the acronyms. Reference to Table 3 is made earlier. This is corrected in the revised version.

*Sec. 3.4. L274-276: An increase of 135% to 913% when Nsg increases from 2 to 5 deserves a lot greater attention. The authors should conduct more analysis here to find out why this is the case. Saying its "exponential" is not enough. The result is also not tied to what is happening to the ice mass. There needs to be a more comprehensive analysis of what is happening to the ice budget as a whole.*

We drew attention to the transition of $N_{sg}$ from 2.0 to 5.0 (log scale) because the peak value of $N_i$ is growing fast in this narrow range, no more. We don't think this corresponds to an underlying physical process in this range of $N_{sg}$. However,

a saturation effect is slightly noticeable in the evolution of $CIBU_{ef}$ for $N_{sg} > 10$. We agree to smooth the text in order to avoid words like "exponential" and "unrealistic":

"The results clearly show that the growth of $N_i$ is fast when $N_{sg}$ reaches …" and "… leads to a tremendously high $N_i$ peak value."

*Sec. 3.4. L276: Another reference to realism without justification.*

This is true, see above for the correction.

*Sec. 3.4. L280: Why is HIND more efficient here? Is it because the air becomes sub-saturated with respect to liquid water? Why about homogenous ice nucleation? What are HMG and HMS?*

Without CIBU ($N_{sg}=0$), the heterogeneous nucleation process, HIND, is essential to feed the ice crystal concentration. However we can't say that HIND is less efficient when CIBU is activated (the horizontal scale is changing from 13a to 13c). The peaks of the HIND profiles are due to the different nucleating properties of the IFN (Ice Freezing Nuclei), dust black carbon and organics, as compiled by Phillips et al. (2008).
The homogeneous nucleation of the ice starts at a height level colder than -36°C. This process HONH is not very active in this case study. The definition of the acronyms is given in Table 3. HMS(HMG) is the Hallett-Mossop process attached to the riming of the droplets on the "snow-aggregates" ("graupel particles"). Both are treated the same way.

*Sec. 3.4. L289: "Equilibrated" is not the right word here. I think you mean "balanced".*

Yes, "balanced" is much better.

*Sec. 3.4. L290: I gather here that there is that the authors have some understanding of why Ni grows exponentially. This can address my earlier comment if the authors can clarify what they mean here. Why do all of these process rates grow in this fashion?*

Well if CIBU is run with $N_{sg}=10$, then the most important processes to shape $N_i$ are CIBU (source), AGGS and CEDS (both sinks). The sedimentation of $N_i$ (SEDI profile) is also noticeable on the plots of Fig. 13. In passing, it is important to stress that the CEDS process acting on $N_i$ corresponds to the full sublimation of the cloud ice crystals (a local loss of $N_i$ concentration) when these are detrained in unsaturated areas of the cloud surroundings. In a 2-moment scheme mixing ratios and number concentrations must be consistent so

$N_i$ is set to zero whenever the mixing ratio $r_i$ becomes negative (in LIMA, we check for a strict conservation of the mass of condensate and water vapour). However it is clear also that ice sublimation is marginal for the ice mixing ratio in the low levels of the clouds because, as expected, the water vapour deposition dominates the growth of the ice inside the convective cells (see the profiles of mixing ratio $r_i$ in Fig. 9).

So increasing $N_i$ through the CIBU source of ice crystal is compensated by an increase of AGGS (more available crystals are captured by the snow-aggregates) and by an increase of CEDS (more ice crystals are detrained). The other processes revealed in Fig 13a ($N_{sg}$=0) are not changing very much but their importance is reduced because of changing the plotting scale when moving from Fig. 13a to Fig. 13c.

Modifications in the text:

"The CIBU source of ice crystals is balanced by an increase of AGGS and, above all, of CEDS (here CEDS represents the sublimation of the ice crystal concentration when detrained in the low level of the cloud vicinity, below the anvil for instance). Finally, the "$N_{sg}$=10" case demonstrates the reality of the exponential-like growth of $N_i$ because the three main driving terms CIBU, CEDS and AGGS are growing at a similar rate that is multiplied by a factor 5, approximately."

*Sec. 3.5. L304-305: "Difficult to interpret" is not a satisfactory conclusion here. If the reader is going to be convinced of the very important argument being made in this section, a better effort needs to be made at understanding how the baseline simulations' Ni change with different ice nucleating particle concentrations. I'm also quite concerned that homogenous ice nucleation hasn't been addressed at all.*

The reviewer is right. What we meant here is that the mean $N_i$ profiles are not growing with $N_{IFN}$ as for instance we end up with very similar $N_i$ profiles but for $N_{IFN}$ =10 and 0.01 dm$^{-3}$ in Fig 14b. The many terms involved in the ice phase budget (see the profiles in Figs 9a and 13a) explain the difficulty of a bulk analysis. However the important result here is that the number concentration of nucleated IFN follows the initial IFN concentrations as shown in Fig. 14 so the IFN nucleation efficiency is independent of the initial $N_{IFN}$

*Sec. 3.5. L308-310: This statement is unclear. The authors say the nucleated IFN evolve in close proportion to initially available IFN but then the authors are also saying that the IFN do not depend on the IFN concentrations as expected?*

No what we wrote is that the nucleation efficiency of the IFN does not depend on the initial IFN concentration. We clarify this point in the revised text.

"In Fig. 14c, the IFN profiles are rescaled (multiplication by an appropriate numbers of powers of ten) to be comparable. Here this is equivalent to computing an IFN nucleation efficiency. The important result here is that the number of nucleated IFN evolves in close proportion to the initially available IFN concentrations, meaning that the nucleating properties of the IFN do not depend on the initial IFN concentration as expected."

*Sec. 3.5. L314-322: The conclusions here are struggling to be properly understood and interpreted due to the fact that not enough information about LIMA or the baseline simulation have been give.*

We rewrote this part more carefully:
"The last plot (Fig. 14d) reproduces the normalized differences of $N_i$ profiles between twin simulations performed with CIBU and without CIBU. Even if simulations made with the same initial concentration $N_{IFN}$, diverge because of additional non-linear effects (vertical transport, enhanced or reduced cloud ice sink processes), the figure gives a flavour of the bulk sensitivity of CIBU to the IFN. The enhancement ratio due to CIBU remains low (less than 1 for $N_{IFN}$ ~ 100 dm$^{-3}$) but it can reach a factor of 20 at 9,000 m height in the case of moderate IFN concentration i.e. for $N_{IFN}$ ~ 1 dm$^{-3}$. The behaviour of LIMA can be explained in the sense that increasing $N_{IFN}$ too much leads to smaller pristine crystals that need a longer time to grow because the conversion to the next category of snow-aggregates is size-dependent (see Harrington et al. 1995 and Vie et al., 2016). On the other hand, a low concentration of IFN initiates fewer snow-aggregate and thus less graupel particles, so the whole CIBU efficiency is also reduced. Consequently, this study confirms the essential role of CIBU to compensate for IFN deficit when cloud ice concentrations are building up."

*Sec. 4. L329-330: I can't agree with this statement. As I've already noted, no justification to this has been given.*

We reword the sentences at L328-330 as follows:
"The number of ice fragments that results from a single shock, $N_{sg}$, is a key parameter which is only estimated from very few past experiments (Vardiman, 1978). A merit of this study is to provide an upper bound to the value of $N_{sg}$ because of the sensitivity of $N_{sg}$ to the simulated precipitation. We found that taking $N_{sg}$>10 reduces significantly the precipitation at the ground. This is not acceptable since most of the cloud schemes (running without CIBU process) are tuned for quantitative precipitation forecasts."

*Sec. 4. L359-360: A quantitative conclusion about the sensitivity of the simulations to different realizations of CIBU (due to changes in observationally constrained parameters) hasn't really been reached.*

We reformulate our last conclusion that a complete set of secondary ice production including Hallett-Mossop, CIBU and the raindrop shattering, is of great interest to simulate very high crystal concentrations. To this end it is worth to check if such situations in real clouds are not the result of a cooperative action between the many secondary sources of ice crystals.

"So the next step in the LIMA scheme is to introduce the shattering of the raindrops during freezing as proposed by Lawson et al. (2015) and to compare with CIBU, because the basic ingredients, raindrops and small ice crystals, leading to a different ice multiplication processes are not the same. Then, the final task is to check that microphysics schemes with all known sources of small ice crystals, nucleation and secondary ice production, are able to cooperate and to reproduce observed ice concentrations which can reach very high values (units of $cm^{-3}$) in deep convective clouds but without convincing explanation yet. Quantitative cloud data gathered in the tropics during HAIC/HIWC (High Altitude Ice Crystals/ High Ice water Content) field project (Leroy et al., 2015; Ladino et al., 2017) could be a starting point to evaluate high resolution cloud simulations with high ice contents."

**References**

Lawson, R.P., S. Woods, and H. Morrison, 2015: The microphysics of ice and precipitation development in tropical cumulus clouds. J. Atmos. Sci., 72, 2429–2445, https://doi.org/10.1175/JAS-D-14-0274.1

Phillips, V. T. J., P. J. DeMott, and C. Andronache, 2008: An empirical parameterization of heterogeneous ice nucleation for multiple chemical species of aerosol. J. Atmos. Sci., 65, 2757–2783.

Sullivan, S. C., Hoose, C., and Nenes, A.: Investigating the contribution of secondary ice production to in-cloud ice crystal numbers, J. Geophys. Res., 122, doi:10.1002/2017JD026 546, 2017.

Yano, J.-I. and Phillips, V. T.: Ice–Ice Collisions: An Ice Multiplication Process in Atmospheric Clouds, J. Atmos. Sci., 68, 322–333, doi:10.1175/2010JAS3607.1, 2011.

---

## Referee Report (RR1)

**Review of "A representation of the collisional ice break-up process in the two-moment microphysics scheme LIMA v1.0 of Meso-NH"**

**Major comments**

Thank you for including a description of the LIMA scheme and the STERAO case study. These help the coherence of the manuscript. Many, but not all, of my questions have been addressed in the author response, and I have no objection to its publication in *GMD*. I am still not sure how the parameterization addresses the discrepancy between ice crystal and INP numbers found *at mixed-phase conditions*, given the highest simulated CIBU contributions at cirrus altitudes. Allowing the process to occur over a wider range of altitudes than in the real atmosphere will certainly affect the results through the vertical latent heating profile and the impact of that heating on dynamics. Some discussion of these considerations could be incorporated.

In line with this kind of discussion, a visualization of the "*upward transport in the convective cells*" of ice crystals formed by CIBU would also be appreciated, since from Figure 9, it seems rather that there is a sedimentation loss from these altitudes. The manuscript could still do with some proofreading because the wording is hard to understand in places.

**Specific comments**

Line 27 – "*The CIBU process was overlooked in cloud physics. So to our knowledge a contribution of CIBU is never accounted for in the vast majority of the currently used microphysics schemes.*"
This is still poorly worded. Can you simply say: "In contrast to the Hallett-Mossop process, the majority of microphysics schemes do not include the CIBU process."

Line 29-30 – "*Yet, even without absolutely incontestable clues, still missing even in recently published cloud data records*"
I would remove this, as it is superfluous.

Line 41 – It does not make sense to motivate the work by a discrepancy between IWC and INP number. It is a discrepancy between *ice crystal number concentration* and INP number.

Lines 63-64 – It is not clear what an "*asymmetric collision*" is. I would still prefer "mass loss" to "*erosion*".

Line 72 – Remove one "*ice*" from ice number concentration.

Line 84 – "Collisions" is a preferable term to "*shocks*" that are generally electrostatic phenomena (and the latter happens due to ice during lightning formation so the potential for confusion is particularly high).

Lines 113-114 – I am still not clear from the author response how both $D_{s,max}$ and $D_{g,min}$ are chosen based on a single criterion for relative terminal velocity. If it is just a matter of choosing round numbers because there are no other constraints, this should be stated explicitly.

Line 138 – Unless I missed it, you do not mention which nucleation scheme is used. This should be included to know if the nucleation tendencies in Figure 13 should be on the high or low side.

Line 156 – I would still explicitly state "*In a 2-moment* bulk *scheme*."

Lines 196-198 – "*by a multicellular storm*" Please add "over land" here. The STERAO case be "*very classical*" but not all readers will necessarily be familiar with it. I would also say "three 3 K-buoyant bubbles along the horizontal wind direction" if this is what is meant in line 198.

Line 226 – I do not think "*disruptive process*" is a clear description. I would just say "From these simulations, inclusion of CIBU can strongly modify surface precipitation when $N_{sg}$ > 10.0 fragments per aggregate-graupel collision."

Lines 233-235 – Here again, a direct comparison of ice mass and number metrics does not make sense. Presumably you mean that higher ice crystal concentrations with larger $N_{sg}$ deplete the supersaturation that would otherwise go to snow-aggregate growth. Please say this instead.
Line 242 – Why would one expect any change in the graupel mixing ratio at all since, from Lines 178 to 179, "*the mass of the graupel is unchanged*" in this CIBU parameterization?

Line 257 – Again can you make clear why there should be a reduction in $r_g$ given that the graupel are acting as "passive colliders" in your parameterization?

Figure 9, author response – I understand that nucleation has a much more important impact on ice number than ice mixing ratio. But here and throughout, a motivation to explain ice mass seems misguided to me. Ice-ice collisional breakup was proposed to explain *discrepancies in measured ice number concentrations.*

Around Line 277, author response – I am still unclear about why ice mixing ratio and number concentration peak at different altitudes. In the author response, I am not sure what the "*limiting value $dr_i/dt$*" means. Can you clarify? There are no *min* functions in Equations 3 to 5.

Line 312-314, Figure 13 – I am curious why the Hallett-Mossop on Graupel process peaks around 5 km if the graupel mixing ratio peaks around 9 km. Is the droplet number large enough to compensate for such low graupel mixing ratios?

Line 305, author response and Lines 340-341 – If the INP number is high enough to deplete supersaturation, you have no homogeneous nucleation. I would imagine that is why you see a decrease in $N_i$ concentration with increasing IFN in Figure 14b.

---

## Referee Report (RR2)

Review number 2 of "A representation of the collisional ice A representation of the collisional ice break-up process in the two-moment microphysics scheme LIMA v1.0 of Meso-NH"

In their revised manuscript, Hoaraue et al. have addressed some of my concerns but not all of them. I appreciate that more details about the model setup and microphysics schemes have been added. However, I still see no discussion on the limitations of the experimental setup, no justification for why  $0.1 < N_{sg} < 10$  is a physically plausible range, and no justification for the authors' conclusions that more work needs to be done by the measurement community to further constrain this range. Additionally, I did not find that the authors made a real effort to improve the language and readability of the manuscripts. As it stands, I still can't recommend publication in GMD.

Major concerns

**The importance of realizing that results are specific to the experimental setup.**

I appreciate the efforts of the authors to investigate a range of  $N_{sg}$  values. However, the results haven't been shown to be robust (e.g. generalizable to some degree to more deep convective cases). That is because the experiments weren't conducted for different cases, and perturbations to the initial conditions and other details of the microphysics scheme haven't been carried out. I understand that the authors may not wish to add experiments at this point, but there at least needs to be an emphasis on this being a limitation. The authors did conduct sensitivity tests to the initial concentration of ice freezing nuclei, these may be enough to establish some trend in the impact of CIBU on LIMA, but the authors do not give that part of the study the attention needed to do so.

**The authors' conclusion that a range of plausible $N_{\text{sg}}$ has been realized has not been justified.**

This in part follows from the preceding critique. Only one case has been simulated, very few changes to said case have been carried out which makes it difficult to conclude that this range can be generalized. In addition, I am still not convinced that a conclusion can be drawn based on how small or large the induced perturbation to the storm dynamics and microphysics. The authors may have a good understanding of this, but they still haven't communicated it well. Please revise this point. Write a very clear paragraph or even section explaining to the reader why a perturbation of a particular magnitude must not be exceeded when CIBU is introduced.

**The conclusion that more measurements are needed to constrain the $N_{sg}$ range.**

As the authors note, it is extremely important for a study such as this to guide future measurements. There is some discussion of this in the conclusions, but it's unclear. Please write a clear paragraph or section indicating the kinds of measurements needed based on the results.

**The sensitivity studies are poorly discussed.**

I understand the desire to write a short paper. We should always strive to write manuscripts in the least wordy way possible. However, this should not come at the expense of poor elaboration on the results of the experiments. It becomes especially frustrating when the reader reaches the interesting section of sensitivity to initial ice nucleating concentrations and is met with a very limited interpretation of what is happening.

**The language remains a limitation.**

Unfortunately, many statements made by the authors may struggle to be understood by a reader due to deficiencies in language. I had urged the authors to revise this aspect of the manuscript, but very little effort was made.

Line by line concerns

Sec .1. L54-55. "Huge" is not quantitative. Please replace with an actual enhancement factor.

Sec. 1. L56-57. This sentence is not clear. I'm struggling to understand what "The experiment setup used there was more appropriate to very big" means.

Sec. 1 L58-63. There is no need to clarify what the study is not. This series of sentences can be omitted, assuming the authors can clarify what the study entails in the sentences that follow.

Sec. 2.1 L88-89. Since the authors haven't introduced what the categories are at this point, they should not expect the reader to understand what "small aggregates covering pristine ice" and "large graupel particles" are. Start by explaining what the categories are, then clearly state which categories are considered for collisional breakup and what size restrictions are applied.

Sec. 2.1 L95. "Symbolic" is not necessary here.

Sec. 2.1 L99. "Simplest" is not necessary here. The writers should say "an expression for alpha which \*" where \* would state what the assumptions behind the expression are.

Sec. 2.1 L125-140. I still don't understand this explanation of  $N_{\text{sg}}$  based on previous work and how it ties to this study.

Sec. 3.1. This is the section where a better job can be done to explain to the reader why a plausible range of  $N_{sg}$  can be concluded.

Sec. 3.2 L267-268. "rain is mostly fed by melting of graupel particles". The authors don't show  $r_r$  production rates from autoconversion vs. melting. Thus, this statement isn't justified. Consider rewording to something more suggestive.

Sec. 3.2 L266. Avoid using "clearly".

Sec. 3.3 L276-279. This sentence is not clear. You are stating what the main processes are but simultaneously talking about how AGGS and CFRZ are changing? Please reword.

Sec. 3.4 L300-319. This is too dense. Please expand this explanation.

Sec. 3.4 L306-307. Please clarify that the  $N_i$  achieved when not considering CIBU is not the actual concentration of ice nucleating particles, but the resultant concentration of ice. This is an important distinction.

Sec. 3.4 L320-321. "Temporal integration" is too wordy. Consider using something simpler like "time integrated" if that's what you mean here.

Sec. 3.4 L325-327. Reword this please. "In the case of water supercooling" is not clear.

Sec. 3.5 L351-353. Why is it difficult to interpret? The results seem clear here. I highly urge expanding this section in such a way to discuss the sensitivity to ice nucleating particle concentrations without CIBU first (beyond two brief sentences) then move on to the case with CIBU.

Concerns not addressed in the first round of revision

Below is a list of comments I wrote in the first round that I believe were not properly addressed.

Sec. 2.1: Please justify the choice of a temperature independent  $N_{sg}$  here. For example, Sullivan et al. (2017) use an  $N_{sg}$  that is temperature dependent.

Sec. 3.1. L199-202: This statement is unjustified. As emphasized in the preceding comment, realism of a specific  $N_{sg}$  range has not been established, therefore the writers' conclusion on the choice of  $N_0$  by Yano and Phillips (2011) being unrealistic is not justified. Also there aren't enough details about the cited study to make a meaningful comparison here.

Sec. 3.4. L280: Why is HIND more efficient here? Is it because the air becomes subsaturated with respect to liquid water? Why about homogenous ice nucleation? What are HMG and HMS?

---

## Author Response (AR2)

[revised manuscript text omitted]

**Major comments**

*Thank you for including a description of the LIMA scheme and the STERAO case study. These help the coherence of the manuscript. Many, but not all, of my questions have been addressed in the author response, and I have no objection to its publication in GMD. I am still not sure how the parameterization addresses the discrepancy between ice crystal and INP numbers found at mixed-phase conditions, given the highest simulated CIBU contributions at cirrus altitudes. Allowing the process to occur over a wider range of altitudes than in the real atmosphere will certainly affect the results through the vertical latent heating profile and the impact of that heating on dynamics. Some discussion of these considerations could be incorporated.*

Even if the manuscript is amendable, we appreciate the positive feedback of the reviewer. The discrepancy between ice crystal and INP number concentrations is not something new but it was recently exacerbated because secondary production of ice crystals is becoming an open question besides the Hallett-Mossop process just above the freezing level.

We have added a new case (more precisely the Weisman-Klemp cases with varying CAPE) in the manuscript to check how the effects of CIBU could be generalized. Through this case, it is possible to show the growing horizontal extension of the ice clouds and an increase of the mean cloud tops when the number of fragments increases. Interestingly, this suggests that convection develops more when CIBU is very strong because a large excess of small ice crystals leads to heating when supersaturated water vapour deposits on these small crystals. However we do not want to go too much farther until the parameterization of CIBU has been well constrained by data (cloud coverage, precipitation at the ground, etc.).

*In line with this kind of discussion, a visualization of the "upward transport in the convective cells" of ice crystals formed by CIBU would also be appreciated, since from Figure 9, it seems rather that there is a sedimentation loss from these altitudes. The manuscript could still do with some proofreading because the wording is hard to understand in places.*

The ice crystals formed by CIBU would be difficult to isolate because they are mixed with those produced by nucleation and by Hallett-Mossop. For instance, a major sink of cloud ice, the transfer to the snow-aggregate category, depends on the deposition rate of big crystals of the total cloud ice crystal size distribution. However it is clearly beyond our purpose to propose, first, a parameterisation of CIBU and then to illustrate some consequences of the inclusion of CIBU in 3D cloud resolved simulations.

**Specific comments**

*Line 27 – "The CIBU process was overlooked in cloud physics. So to our knowledge a contribution of CIBU is never accounted for in the vast majority of the currently used microphysics schemes." This is still poorly worded. Can you simply say: "In contrast to the Hallett-Mossop process, the majority of microphysics schemes do not include the CIBU process."*

The final wording is now:

"However, intriguingly and in contrast to the Hallett-Mossop (hereafter H-M) ice multiplication mechanism[1] (Hallett and Mossop, 1974), the vast majority of microphysics schemes do not include the CIBU process."

*Line 29-30 – "Yet, even without absolutely incontestable clues, still missing even in recently published cloud data records" I would remove this, as it is superfluous.*

We agree.

*Line 41 – It does not make sense to motivate the work by a discrepancy between IWC and INP number. It is a discrepancy between ice crystal number concentration and INP number.*

This was a mistake, 'ice water contents' has been replaced by 'ice water concentrations'.

*Lines 63-64 – It is not clear what an "asymmetric collision" is. I would still prefer "mass loss" to "erosion".*

"asymmetric" has been removed. Also we have replaced "erosion" by "mass loss".

*Line 72 – Remove one "ice" from ice number concentration.*

Done.

*Line 84 – "Collisions" is a preferable term to "shocks" that are generally electrostatic phenomena (and the latter happens due to ice during lightning formation so the potential for confusion is particularly high).*

Done.

*Lines 113-114 – I am still not clear from the author response how both D s,max and D g,min are chosen based on a single criterion for relative terminal velocity. If it is just a matter of choosing round numbers because there are no other constraints, this should be stated explicitly.*

It is implicitly true that we are taking round numbers for $D_{s,max}$ and $D_{g,min}$. This is now explicitly stated in the text.

"The choice of round numbers for $D_{smax}$ and $D_{gmin}$ is above all dictated by …"

*Line 138 – Unless I missed it, you do not mention which nucleation scheme is used. This should be included to know if the nucleation tendencies in Figure 13 should be on the high or low side.*

The heterogeneous ice nucleation scheme is based on Phillips et al. (2008 and 2013). It is adapted to the LIMA scheme as described in Vié et al. (2016) with an integration over the IFN size distribution. We consider ice nucleation by deposition, when the ice nuclei are totally insoluble, and ice nucleation by immersion, for partially soluble ice nuclei, separately. In our case, the homogeneous freezing of the cloud droplets is very low.

We have completed the original sentence (line 140-142) for clarification:

"The ice phase is more complex as we consider the nucleation by deposition on insoluble IFN (black carbon and dust) and the nucleation by immersion (glaciation of tagged droplets because they are formed on partially soluble CCN, containing an insoluble core)."

In Fig. 13, HIND (nucleation by deposition) is shown in purple.

*Line 156 – I would still explicitly state "In a 2-moment bulk scheme."*

There are spectral or bin microphysics schemes with a detailed description of the particle size distributions (PSD) and bulk schemes which assume a mathematical form of the PSD. To us, moments of the PSD imply a bulk scheme. However we have added the word "bulk" into the text.

*Lines 196-198 – "by a multicellular storm" Please add "over land" here. The STERAO case be "very classical" but not all readers will necessarily be familiar with it. I would also say "three 3 K-buoyant bubbles along the horizontal wind direction" if this is what is meant in line 198.*

We agree with the idea of giving more details about the STERAO storm. We have reformulated the description of the case:

"This case is characterized by a multicellular storm which becomes supercellular after 2 hours. The simulations were initialized with the sounding of northeastern Colorado given in Skamarock et al. (2000) and convection was triggered by three 3K-buoyant bubbles aligned along the main diagonal of the X,Y plane in the wind axis."

We have removed the words "very classical".

*Line 226 – I do not think "disruptive process" is a clear description. I would just say "From these simulations, inclusion of CIBU can strongly modify surface precipitation when N sg > 10.0 fragments per aggregate-graupel collision."*

We agree to replace the sentence by the suggestion of the reviewer.

"From these first 3D numerical experiments, inclusion of CIBU can strongly modify surface precipitation when $N_{sg} > 10.0$ fragments per aggregate-graupel collision."

*Lines 233-235 – Here again, a direct comparison of ice mass and number metrics does not make sense. Presumably you mean that higher ice crystal concentrations with larger $N_{sg}$ deplete the supersaturation that would otherwise go to snow-aggregate growth. Please say this instead.*

Yes we have followed the suggestion.

"Basically, intensifying the CIBU process by increasing $N_{sg}$ leads to higher cloud ice crystal concentrations, which deplete the supersaturation of water vapour that would otherwise contribute to the deposition growth of the snow-aggregates."

*Line 242 – Why would one expect any change in the graupel mixing ratio at all since, from Lines 178 to 179, "the mass of the graupel is unchanged" in this CIBU parameterization?*

You are right, this is confusing. So we have added a sentence.

"… a reduction of $r_g$ which clearly stands out at z=8,000 m. The decrease of $r_g$, even if graupels are passive colliders for CIBU, is the result of the decrease of $r_s$ in the growth chain of the precipitating ice. The final result …"

*Line 257 – Again can you make clear why there should be a reduction in r g given that the graupel are acting as "passive colliders" in your parameterization?*

Done, see above.

*Figure 9, author response – I understand that nucleation has a much more important impact on ice number than ice mixing ratio. But here and throughout, a motivation to explain ice mass seems misguided to me. Ice-ice collisional breakup was proposed to explain discrepancies in measured ice number concentrations.*

We fully agree but, besides changing $N_i$ with CIBU, it is necessary to examine the impact on $r_i$ as the new small ice crystals pump the excess water vapour, with some consequences for the graupels at the end of the growth chain.

*Around Line 277, author response – I am still unclear about why ice mixing ratio and number concentration peak at different altitudes. In the author response, I am not sure what the "limiting value dr i /dt" means. Can you clarify? There are no min functions in Equations 3 to 5.*

The mass growth of the cloud ice, concerning the CIBU contribution, is computed as the mean mass of the pristine ice crystals (a local value) times the local CIBU tendency, $\partial N_i/\partial t|_{CIBU}$ in Eq. 3. So $\partial r_i/\partial t|_{CIBU} = (r_i/N_i) \times \partial N_i/\partial t|_{CIBU}$. This is justified because CIBU is not associated with a characteristic or specific hump on the small ice crystal size distributions.

However, in any case, $\partial r_i/\partial t|_{CIBU}$ must be limited by the mass of colliding individual aggregates given by Eq. 5. This is why we talked of "limiting value of $\partial r_i/\partial t$" in our response. This is clearly described in the text (section 2.3).

*Line 312-314, Figure 13 – I am curious why the Hallett-Mossop on Graupel process peaks around 5 km if the graupel mixing ratio peaks around 9 km. Is the droplet number large enough to compensate for such low graupel mixing ratios?*

Hallett-Mossop needs graupel and cloud droplets (more abundant in the low levels). But, more importantly, this process is efficient in the [-3, -8] range of temperature (and reproduced by a symmetrical triangular function). This explains why Hallett-Mossop peaks around 5 km.

*Line 305, author response and Lines 340-341 – If the INP number is high enough to deplete supersaturation, you have no homogeneous nucleation. I would imagine that is why you see a decrease in N i concentration with increasing IFN in Figure 14b.*

Yes we agree that less supersaturation decreases nucleation in Phillips et al.'s (2008 and 2013) papers.

**Additional references**

Phillips, V. T., DeMott, P. J., and Andronache, C.: An empirical parameterization of heterogeneous ice nucleation for multiple chemical species of aerosol, J. Atmos. Sci., 65, 2757–2783, 2008.

Phillips, V. T., Demott, P. J., Andronache, C., Pratt, K. A., Prather, K. A., Subramanian, R., and Twohy, C.: Improvements to an empirical parameterization of heterogeneous ice nucleation and its comparison with observations, J. Atmos. Sci., 70, 378–409, 2013.

**Responses to Referee #02**

"A representation of the collisional ice break-up process in the two-moment scheme LIMA v1.0 of Meso-NH" by Hoarau et al.

*In their revised manuscript, Hoarau et al. have addressed some of my concerns but not all of them. I appreciate that more details about the model setup and microphysics schemes have been added. However, I still see no discussion on the limitations of the experimental setup, no justification for why 0.1 < Nsg < 10 is a physically plausible range, and no justification for the authors' conclusions that more work needs to be done by the measurement community to further constrain this range. Additionally, I did not find that the authors made a real effort to improve the language and readability of the manuscripts. As it stands, I still can't recommend publication in GMD.*

We thank the referee for his careful review. We hope to have satisfactorily answered to the questions and addressed all the concerns.

The purpose of the work is to show a simple implementation of the Collisional Ice Break-Up (CIBU) process and to study the consequences of this operation in a cloud resolving model. The intensity of the CIBU process depends on the $N_{sg}$ parameter, that is, the number of ice fragments produced per "snow-graupel" collision. This parameter is crucial and not well known despite a few lab data.

In our application to STERAO, we show that taking $N_{sg}$>10 leads to a measurable decrease of the surface precipitation in 3D simulated cloud systems. We add new simulations of supercell storms with a varying CAPE following the technique of Weisman and Klemp (1982). This second series of (WK) experiments confirm that the accumulated precipitation is affected by a strong CIBU efficiency, up to $N_{sg}$=50.

As a result, one can see that is very easy to increase the cloud ice concentration $N_i$ by several orders of magnitude through a secondary process of ice production like CIBU even if the proposed parameterization of CIBU seems too simplistic or not sufficiently grounded. It is less easy to increase $N_i$ while not perturbing the surface precipitation too much. So, for these reasons we suggest randomizing $N_{sg}$ to obtain high values of $N_i$ very locally.

The last WK simulations show that a rough tuning of CIBU could be achieved by examining the coverage of ice clouds connected with a convective outbreak.

We have revised our conclusions. We no longer suggest limiting the upper range of acceptable values for $N_{sg}$ to 10 but we encourage other microphysics schemes to include CIBU and to check the consequences of CIBU by simulating deep convective events. This is a useful task because adding a very large number of small ice crystals should have a profound impact on the genesis of precipitation.

This step, i.e. the transfer and the growth of cloud ice to hydrometeor categories, often differs among microphysics schemes.
The readability of the whole manuscript has been very carefully revised by a native English speaker.

*Major concerns*

**The importance of realizing that results are specific to the experimental setup.**
*I appreciate the efforts of the authors to investigate a range of Nsg values. However, the results haven't been shown to be robust (e.g. generalizable to some degree to more deep convective cases). That is because the experiments weren't conducted for different cases, and perturbations to the initial conditions and other details of the microphysics scheme haven't been carried out. I understand that the authors may not wish to add experiments at this point, but there at least needs to be an emphasis on this being a limitation. The authors did conduct sensitivity tests to the initial concentration of ice freezing nuclei, these may be enough to establish some trend in the impact of CIBU on LIMA, but the authors do not give that part of the study the attention needed to do so.*

In the latest revised version, we have added WK, a deep convective case studied by Weisman and Klemp (1984), to show that, with a varying CAPE, inclusion of CIBU with a variable strength confirms the STERAO results.
We agree that a lot of work is necessary to establish more firmly that collisional ice break up is an important process besides raindrop shattering, for the microphysics of deep convective clouds. The interaction between ice nucleation and ice break-up deserves a more specific study. For instance, a heavy secondary production of ice crystals (here, through CIBU) necessarily perturbs the supersaturation field in the clouds with a possible competition between nucleating IFN and growing ice crystals or ice fragments.

**The authors' conclusion that a range of plausible Nsg has been realized has not been justified.**
*This in part follows from the preceding critique. Only one case has been simulated, very few changes to said case have been carried out which makes it difficult to conclude that this range can be generalized. In addition, I am still not convinced that a conclusion can be drawn based on how small or large the induced perturbation to the storm dynamics and microphysics. The authors may have a good understanding of this, but they still haven't communicated it well. Please revise this point. Write a very clear paragraph or even section explaining to the reader why a perturbation of a particular magnitude must not be exceeded when CIBU is introduced.*

Another case ("WK" sounding) is simulated now and we retrieve the same conclusion of the STERAO case that the production of a high number of ice fragments is deeply perturbing the microphysics state of the clouds with less surface precipitation and an extended ice cloud cover (but when the atmosphere is not too dry in the vicinity of the cloud tops).

We attenuate our previous conclusions that $N_{sg}$ should not exceed some precise threshold. However, a possible way of doing is also to work on the concept of random process for CIBU to get high values of $N_i$, but very locally.

**The conclusion that more measurements are needed to constrain the Nsg range.**

*As the authors note, it is extremely important for a study such as this to guide future measurements. There is some discussion of this in the conclusions, but it's unclear. Please write a clear paragraph or section indicating the kinds of measurements needed based on the results.*

This is a little bit beyond the purpose of a study based on simulations. However and after the "WK" simulations, we reworked the conclusion to suggest that looking at the ice cloud cover may be a way to get a macroscopic adjustment of the $N_{sg}$ parameter. Anyway, a fundamental need is to redo laboratory experiments in chambers with controlled environment and to get in situ sampling of the ice crystals (formvar replicator) to characterize the alterations of the crystal habits.

**The sensitivity studies are poorly discussed.**

*I understand the desire to write a short paper. We should always strive to write manuscripts in the least wordy way possible. However, this should not come at the expense of poor elaboration on the results of the experiments. It becomes especially frustrating when the reader reaches the interesting section of sensitivity to initial ice nucleating concentrations and is met with a very limited interpretation of what is happening.*

We agree but model results depend also on model set-up here, the initial concentrations and vertical profiles of the CCN and IFN particles. We try to be careful when drawing general conclusions with a limited set of experiments.

The question of the initial concentration of IFN is introduced to check if CIBU is still operating in the case of very low IFN concentrations. This corresponds to a marginal functioning of the LIMA scheme. However the results show that once the ice phase is initiated by ice nucleation (IFN are always indispensable), ice multiplication is possible. The conclusion is that the availability of the IFN, down to 0.001 $dm^{-3}$, seems not a limitation for CIBU in the STERAO case.

**The language remains a limitation.**

*Unfortunately, many statements made by the authors may struggle to be understood by a reader due to deficiencies in language. I had urged the authors to revise this aspect of the manuscript, but very little effort was made.*

We agree and we apologize for that. The new version of the manuscript has been revised in deep.

*Line by line concerns*

*Sec .1. L54-55. "Huge" is not quantitative. Please replace with an actual enhancement factor.*

On the basis of original Fig. 4 of Takahashi et al. (1995), it was difficult to say more than "huge" since the number of fragments lies between approximately 50 and 600. The paper also shows a wide variability of the number of "ejected" ice particles at a given temperature. So it was not possible to be more quantitative, although we risked indicating a value of 400 fragments in the next sentence.

*Sec. 1. L56-57. This sentence is not clear. I'm struggling to understand what "The experiment setup used there was more appropriate to very big" means.*

We have summarized the laboratory experiment by noting that the colliders were "1.8 cm diameter ice sphere(s)". The speed of the grazing collisions was varied to simulate the impact velocity between large and small graupels (4 m/s corresponding to a big particle of ~4 mm).

*Sec. 1 L58-63. There is no need to clarify what the study is not. This series of sentences can be omitted, assuming the authors can clarify what the study entails in the sentences that follow.*

We thought that, because Yano and Phillips's papers (*loc. cit.*) were heavily oriented to the study of the "explosive" nature of ice multiplication by collisional break-up, we would like to be careful by saying that our goal was to implement a simple parameterization of CIBU and then to examine the perturbations brought to the cloud microphysics.
With this way of working we consider that CIBU is a common process that should operate when cloud conditions are met (presence of aggregates and graupels) and no more.

*Sec. 2.1 L88-89. Since the authors haven't introduced what the categories are at this point, they should not expect the reader to understand what "small aggregates covering pristine ice" and "large graupel particles" are. Start by*

*explaining what the categories are, then clearly state which categories are considered for collisional breakup and what size restrictions are applied.*

The reviewer is right, the first sentence of the paragraph has been reworked:

"In contrast to the work of Yano and Phillips (2011), where large and small graupel particles fuelled the CIBU process, we consider collisions involving two types of precipitating ice here: small ice particles growing by deposition and aggregation (aggregates including dendritic pristine ice crystals with a size >150 µm) and big, massive graupel particles growing by riming."

*Sec. 2.1 L95. "Symbolic" is not necessary here.*

Right. We have replaced it by "general" because, more precisely, the "$n$" represent particle size distributions. Then, at the beginning of Section 2.3, we specify that "$N$" is a total number concentration (zeroth moment of $n$).

*Sec. 2.1 L99. "Simplest" is not necessary here. The writers should say "an expression for alpha which *" where * would state what the assumptions behind the expression are.*

We have followed the suggestion. The sentence is now:
"An expression for α, which does not include thermal and mechanical energy effects, is"

*Sec. 2.1 L125-140. I still don't understand this explanation of Nsg based on previous work and how it ties to this study.*

Well, we tried to review the studies done around $N_{sg}$ estimates. This critical parameter is a constant or the realization of a random process or it is modelled as in Vardiman (1978) and Phillips et al. (2017). So our purpose was to argue for an upper limit $N_{sg\_max}$ (here 50) in order to perform simulations for $0 < N_{sg} < N_{sg\_max}$ to explore the perturbations brought to the microphysics. In any case, $N_{sg}$ is a number, not the output of a parameterization.

*Sec. 3.1. This is the section where a better job can be done to explain to the reader why a plausible range of Nsg can be concluded.*

Following the suggestion of Rev 1, a sentence has been reworked in this section. Note that, in the last revised version of the manuscript, we added WK cases showing some similarities with STERAO, i.e. a reduction of the precipitation peak value and/or a reduction of the precipitation area.

*Sec. 3.2 L267-268. "rain is mostly fed by melting of graupel particles". The authors don't show rr production rates from autoconversion vs. melting. Thus, this statement isn't justified. Consider rewording to something more suggestive.*

The decrease of the $r_r$ profiles in Fig. 8 is barely visible so we have removed the sentence.

*Sec. 3.2 L266. Avoid using "clearly".*

Done.

*Sec. 3.3 L276-279. This sentence is not clear. You are stating what the main processes are but simultaneously talking about how AGGS and CFRZ are changing? Please reword.*

We have rewritten the paragraph:
"As expected, the tendencies of $r_i$ (Fig. 9a-c) are the most affected by the CIBU process. The main processes standing out in Fig. 9a, when CIBU is not activated, are CEDS (Deposition-Sublimation), essentially a gain term and AGGS (Aggregation), the main loss of $r_i$ by aggregation at a rate of 0.5 x 10$^{-3}$ gkg$^{-1}$s$^{-1}$. The loss of $r_i$ by CFRZ (Drop Freezing by Contact) makes a moderate contribution as some raindrops are present in the glaciated part of the storm. Above $z$=10,000 m, the net loss of $r_i$ (AGGS and SEDI, the cloud ice sedimentation) is balanced by the convective vertical transport (not shown). When $N_{sg}$ = RANDOM, the $r_i$ tendencies are amplified even with a modest contribution of ~0.2 x 10$^{-3}$ gkg$^{-1}$s$^{-1}$ for CIBU itself. The growth of AGGS, which doubles at 10 km height, is caused by CIBU and by an increase in the convection because SEDI (a loss) is amplified in response to an increase of $r_i$ in the upper levels. The CFRZ contribution is also increased. The last case, with $N_{sg}$ =10 (Fig. 9c) confirms a further increase of the rates except for CFRZ, interpreted here as a lack of raindrops."

*Sec. 3.4 L300-319. This is too dense. Please expand this explanation.*

The whole paragraph has been rewritten:
"The next point examines the behaviour of the cloud ice number concentration as a function of the strength of CIBU after 4 hours of simulation. Figure 12 shows that the altitude of the $N_i$ peak value decreases when $N_{sg}$ increases. In the absence of CIBU ($N_{sg}$ = 0), the source of $N_i$ is the heterogeneous nucleation processes on insoluble IFN and on coated IFN (nucleation by immersion) which are more efficient at low temperature. Nucleation on IFN provides a mean peak value of $N_i$ = 400 kg$^{-1}$ at $z$ = 11,500 m. In contrast, the $N_{sg}$ = 10 case (here scaled by a factor 0.1 for plotting reasons) keeps the trace of an explosive production of

cloud ice concentration, $N_i$ = 7,250 $kg^{-1}$, due to CIBU. The altitude of the maximum of $N_i$ in this case ($z$ = 10,000 m) is consistent with the location of the maximum value of the $r_s$ x $r_g$ product (see Fig. 8). The "RANDOM" simulation produces $N_i$ = 1100 $kg^{-1}$ at $z$ = 11,000 m, a number concentration which is similar to that found for the $N_{sg}$ = 2 case. Table 2 reports the peak amplitude of the $N_i$ profiles as a function of $N_{sg}$ but after 3 hours of simulation, when the CIBU rate is strongly dominant. Additional cases were run to cover 0.1< $N_{sg}$ <50 with a logarithmic progression above $N_{sg}$ = 1.0. A CIBU enhancement factor, $CIBU_{ef}$, is computed as $N_i$ ($N_{sg}$) / $N_i$ ($N_{sg}$ = 0) – 1, as $N_i$ ($N_{sg}$ = 0) stands for a baseline not affected by CIBU. The results  show that the growth of $N_i$ is fast when $N_{sg}$ reaches ~5 ($CIBU_{ef}$ switches from 135 % to 913 % when $N_{sg}$ moves from 2 to 5). Taking $N_{sg}$ = 50 leads to an extremely high peak value of $N_i$."

*Sec. 3.4 L306-307. Please clarify that the Ni achieved when not considering CIBU is not the actual concentration of ice nucleating particles, but the resultant concentration of ice. This is an important distinction.*

We don't understand this remark. We consider several sources of IFN on the one hand and a single concentration of cloud ice on the other hand. The budget of the $N_{IFN}$ is independent of the budget of $N_i$ because the IFN are also state variables in LIMA. Adding or ignoring CIBU doesn't change the situation.

*Sec. 3.4 L320-321. "Temporal integration" is too wordy. Consider using something simpler like "time integrated" if that's what you mean here.*

We have adopted "time integration".

*Sec. 3.4 L325-327. Reword this please. "In the case of water supercooling" is not clear.*

In the LIMA scheme, the Hallett-Mossop process operates with the riming of the graupel (HMG) and also during the riming of the snow-aggregates category (HMS). The heavy riming of the snow-aggregate particles is a source term for the graupel, so Hallett-Mossop should operate there. The sentence has been reworded as follows:
"Here, we consider that H-M also operates for the snow-aggregates because this category of ice includes lightly rimed particles that can rime further to form graupel particles."

*Sec. 3.5 L351-353. Why is it difficult to interpret? The results seem clear here. I highly urge expanding this section in such a way to discuss the sensitivity to ice nucleating particle concentrations without CIBU first (beyond two brief sentences) then move on to the case with CIBU.*

We found it was difficult to interpret because there is no clear trend in the "no CIBU" case of Fig. 14b to understand the sensitivity of $N_i$ to $N_{IFN}$. We could expect $N_i$ to grow in proportion to $N_{IFN}$ but this is not the case. A possible reason is that we are dealing with mean vertical profiles and also, after 4 hours of model integration, the simulations may start to diverge and cease to be comparable. Also the supersaturation field of water vapour is not tracked during the simulations to see if it plays a role. It is true, however, that this is a point to investigate more in the future.

*Concerns not addressed in the first round of revision*

*Below is a list of comments I wrote in the first round that I believe were not properly addressed.*

*Sec. 2.1: Please justify the choice of a temperature independent Nsg here. For example, Sullivan et al. (2017) use an Nsg that is temperature dependent.*

Sullivan et al. (2017) were inspired by the results of Takahashi et al. (1995), in which we have little confidence for many reasons: 1/ there is a large spread of the data around -15°C and -20°C (see above) so it is questionable how a temperature-dependent formula could be adjusted (by eye, we guess), 2/ the number of fragments per collision is at least ten times more than what was carefully observed by Vardiman (1978) with natural crystals, 3/ the laboratory apparatus of Takahashi does not simulate the break-up of, for instance, radiating dendritic crystals; it is appropriate to the study of collisions between very big, artificially grown, graupels, which do not break up (it seems that it is more the protuberances on the rough surface of the graupels that are ejected and produce the "fragments").

In our work, we put forward the size distribution properties of the colliders to select the range where CIBU should operate, in order to integrate over it. This is in contrast with Sullivan et al. (2017) who were working with less realistic, monodispersed particles. Even if there is a temperature dependence of the CIBU efficiency, we believe that the most important features of CIBU are the particle size dependence and the cloud conditions leading to the occurrence of aggregates and graupels. We also think that additional laboratory experiments are truly necessary to confirm any thermal effect on CIBU.                    .

*Sec. 3.1. L199-202: This statement is unjustified. As emphasized in the preceding comment, realism of a specific Nsg range has not been established, therefore the writers'conclusion on the choice of N0 by Yano and Phillips (2011) being unrealistic is not justified. Also there aren't enough details about the cited study to make a meaningful comparison here.*

We don't claim that the choice of N0=50 by Yano and Phillips (2011) is unrealistic. Our results suggest that taking $N_{sg}$ as high as 50 leads to very high $N_i$ concentrations (well, that's fine!) but taking $N_{sg}$=50 also strongly decreases the surface precipitation in some cases and so this is a little bit annoying because one purpose of microphysics schemes is to simulate accurate surface precipitation. Up to now, after Yano and Phillips (2011) or Sullivan et al. (2017, 2018), no experiment with a complete microphysics scheme has been performed in 3D simulations to check the consequences of a huge increase in $N_i$.

*Sec. 3.4. L280: Why is HIND more efficient here? Is it because the air becomes subsaturated with respect to liquid water? Why about homogenous ice nucleation? What are HMG and HMS*

If you are curious to understand why HIND is more efficient at 14 km when $N_{sg}$=RANDOM than when $N_{sg}$ =0, we have no short and solid explanation to offer. Basically HIND is more efficient when the concentration of IFN is increased or when the supersaturation of water vapour over ice is large but, here, there is no obvious connection with the increase of $N_i$ due to CIBU. The point you raised, interaction between nucleation and CIBU, deserves more investigations with specific diagnostics but it is well beyond the topic of the study.

The homogenous ice nucleation is not a very important source of cloud ice in our simulations. This is what we found.

The meaning of HMG and HMS were given in Table 3, they are source terms for $N_i$.

---

## Author Response (AR3)

**Responses to the Topical Editor**

« A representation of the collisional ice break-up process in the two-moment scheme LIMA v1.0 of Meso-NH » by Hoarau et al.

**Major comments**

We warmly acknowledge your last review including suggestions to improve the English expression. We hope that, thanks to your effort, most of unclear formulations have been fully removed. It is absolutely true that the first versions of the draft should have deserved a more careful internal review before sending to the journal. We apologize for that.

**Specific comments**

We followed all the suggestions of replacement.

L38: "effect" replaced by "mechanism"

L69: The sentence is replaced by "The LIMA scheme inserted in host model Meso-NH (Lafore et al, 1998) is the framework of the present study."

L92: New sentence is: "A particularity of the proposed parameterization of CIBU is to impose an impact velocity of the graupel particles well above 1 ms$^{-1}$ to stay in the break-up regime of the aggregates. This is achieved by selecting the size range of the aggregates and the graupel particles to enable CIBU."

L108: Our purpose is to ensure that "... the impact velocity $V_{sg}$ should be large enough to enable CIBU." in any case. Our choice here is to restrict the CIBU mechanism to graupel particles larger than a minimum size ($D_{gmin}$) and to aggregates with a maximal size $D_{smax}$ to ensure $V_{sg} > 1$ ms$^{-1}$ (L105). As a result, integrations over the size distribution of the graupel and the aggregates become partial integrations (Eq. 4).

L112: New sentence is "The lower bound value of the aggregates, $D_{smin}$, is such that the collision efficiency with a graupel particle is approaching unity."

L134: We retained "implying".

L136: Basically we chose this way of doing because the generation of random numbers with a uniform distribution is made on the interval (0, 1) and not on (-1, +1).

L179: We removed "flexible" which means nothing there.

L210: We cut the sentence in two parts.

L231: We rearranged the second part of the sentences, "...called "RANDOM" hereafter, where $N_{sg} \in [0.1, 10]$ is generated by a random process as explained above. The perturbation caused by CIBU is noticeable in this case too; it remains weak for the precipitation field."

L256: We changed the whole sentence, "The results are those expected when comparing with Figs 3-6. The examination of the microphysics fields suggests that the "RANDOM" simulation corresponds to a mean CIBU intensity which is intermediate between $N_{sg}=1$ and $N_{sg}=10$."

L274: We corrected the sentence, "As expected, many tendencies of $r_i$ (Fig. 9a-c) are affected by the CIBU process."

L315: "... sharply rises ..."

L372: We modified the wording, "The environmental conditions of the simulations were close to those ..."

L429: Here we prefer to keep "ice thickness" instead of "ice water path" (IWP) as suggested because we distinguish the contribution of the different ice components in LIMA. The definition is the same but IWP is more used in radiative transfer than cloud physics.

L435: The new sentence is "This is in contrast with the 1 mm contour of cloud ice thickness which area increases with $N_{sg}$ as shown in Fig. 18."

L441: "comprehensive" was superfluous.

L449: We reformulate the two sentences as follows, "We found that taking $N_{sg}>10$ significantly reduces the precipitation at the ground. This is problematic since most of the cloud schemes (running without the CIBU process) are carefully verified for quantitative precipitation forecast in operational applications."
We think that CIBU effect should be moderate. Cloud physics schemes are working fairly well so a systematic degradation of their performances in deep convective cases would have been noticed. However we think that the CIBU mechanism can be invoked in very deep (tropical) storms to explain anomalously-high cloud ice contents.

In addition, we are working on the simulation of long-lasting cloud systems, like cyclones with vast cloud ice coverage, to detect any sensitivity of this cloud characteristic to CIBU.

L468: "The effects of CIBU have been confirmed by a second series of WK simulations."

[revised manuscript text omitted]

---

## Author Response (AR4)

[revised manuscript text omitted]

We acknowledge your last review in which you mostly asked us to improve the English wording.

The manuscript has been revised independently by two English native speakers: a post doc at LACy (La Réunion) working on cyclone simulations and a retired teacher in Toulouse. They are acknowledged at the end of the manuscript. Both suggested many corrections (spelling and grammar rules) that we followed. We hope that, thanks to their effort, most of unclear formulations have been fully removed and that the text is clean now after a final cross-check to remove the last typos.